# DRL-STAF: A Deep Reinforcement Learning Framework for State-aware Forecasting of Complex Multivariate Hidden Markov Process

## Abstract

Multivariate hidden Markov process forecasting remains challenging due to non-linearity, nonstationarity, hidden state transitions, and cross-sequence dependencies. Deep learning (DL) methods have shown strong predictive performance in time series forecasting but generally lack explicit state modeling and interpretable state estimation, while Hidden Markov Model (HMM) and its variants can provide explicit state representations but are limited in capturing complex nonlinear observation patterns and suffer from scalability issues. To address these limitations, we propose a **D**eep **R**einforcement **L**earning based framework for **ST**ate-**A**ware **F**orecasting of complex multivariate hidden Markov process (**DRL-STAF**), which simultaneously predicts the next-step observation and estimates the corresponding hidden state. In the proposed framework, deep learning is used as the emission function to capture complex nonlinear observation patterns, while deep reinforcement learning models state transitions, supporting flexible adaptation to diverse transition patterns without predefined structural assumptions. In particular, DRL-STAF remains effective when dealing with complex multivariate hidden Markov processes, such as coupled higher-order semi-Markov dynamics, that typically suffer from state-space explosion. Comprehensive experiments demonstrate superior predictive performance and accurate state estimation compared with HMM and its variants, standalone deep learning methods, and existing DL-HMM hybrid methods.

## 1 Introduction

Hidden Markov models (HMMs) provide a foundational framework for capturing latent regime transitions in time series (Rabiner, 1989; Sun et al., 2023) and have been widely applied to dynamic system modeling, anomaly detection, and sequential decision-making. Classical HMMs posit a discrete, first-order Markov latent process and conditionally independent emissions drawn from simple parametric families (e.g., Gaussian or Poisson), often with time-homogeneous transition probabilities. Recent advances relax these assumptions by modeling non-exponential state durations (Lin et al., 2022), introducing higher-order dependencies (Rodriguez-Fernandez et al., 2017), and incorporating dependencies across multiple sequences (Li & Zhang, 2023). This work focuses on multivariate HMMs, which assume that the hidden states of different variables interact with each other, so the next-state distribution of every variable depends on the joint latent configuration at the previous time (Bolton et al., 2018; Wang et al., 2019). These models retain explicit, interpretable latent states that help elucidate system dynamics. However, three key limitations remain: (i) parametric emissions often fail to capture complex, nonlinear observation structure in high-dimensional multivariate data; (ii) transition dynamics generally require a pre-specified (low-parameter) structure and become cumbersome to make richly time-varying or input-dependent without proliferating parameters; and (iii) multivariate HMMs suffer from a combinatorial explosion of the joint state space, rendering exact inference impractical. These challenges motivate a deep, input-driven multivariate HMM that preserves the Markov and conditional-independence axioms while parameterizing transitions and emissions with expressive neural functions and enabling scalable approximate inference.

Recently, researchers have explored combining deep learning (DL) methods with HMMs and their variants to better model observation sequences and latent state dynamics. Existing methods gen-

erally follow two main directions. One direction primarily focuses on using deep learning to enhance the modeling of emission processes or observation trajectories, while retaining HMM-style transition structures to capture latent state dynamics (Dahl et al., 2012; Ilhan et al., 2023; Bansal & Zhou, 2025). These models are typically trained through likelihood-based procedures such as forward–backward, EM, or Viterbi decoding. The other direction further parameterizes transition potentials with neural networks but still optimizes the HMM marginal likelihood and performs inference through differentiable relaxations of the forward algorithm (Tran et al., 2016; Song et al., 2023). These DL-HMM hybrids have shown promise, particularly in improving expressiveness over traditional HMMs. However, they remain fundamentally likelihood-driven and are generally designed for univariate time series, which leads to several limitations in state-aware forecasting. First, the reliance on explicit generative assumptions in likelihood-based training hinders scalability and flexibility in high-dimensional, time-varying, or cross-variable settings. Second, maximizing marginal likelihood does not necessarily reduce forecasting errors or improve state-estimation accuracy, as likelihood can increase simply through smoother transitions or enlarged emission variances rather than genuine predictive gains (Lotfi et al., 2022). Third, when strong nonlinear emission networks are embedded within approximate inference, the posterior must also account for their complexity, which can introduce inference gaps and high-variance gradients and make long-horizon or input-conditioned settings difficult to optimize (Cremer et al., 2018; Rainforth et al., 2018; Vértes & Sahani, 2018). Finally, the fundamental issue of combinatorial state space explosion in multivariate settings remains largely unaddressed.

Another critical limitation lies in the treatment of latent state estimation. Most DL-HMM hybrids rely on soft decoding, producing posterior-weighted averages across all candidate states. While this approach minimizes expected squared error, it often blurs state boundaries and underestimates volatility, which hinder interpretability and degrade performance in downstream decision tasks. In contrast, hard decoding assigns the most probable state at each time step, yielding more interpretable and state-consistent predictions. However, its non-differentiable nature makes end-to-end optimization challenging, which is why most existing models avoid it. Developing differentiable approximations of hard decoding or alternative training strategies remains an open challenge for enabling sharper and more interpretable latent state inference in deep HMM frameworks.

To address the aforementioned limitations, we propose DRL-STAF, a state-aware forecasting framework for complex multivariate hidden Markov processes. Instead of relying on likelihood-based inference, DRL-STAF integrates deep learning (DL) for flexible emission modeling with a deep reinforcement learning (DRL) agent that directly estimates discrete hidden states. In doing so, DRL-STAF dynamically adapts transition mechanisms based on historical context and observed feedback, enabling it to capture rich temporal variability and cross-variable interactions. The core idea is to preserve the probabilistic and interpretable structure of HMMs while significantly enhancing their expressiveness and adaptability through deep architectures. The main contributions of this work are summarized as follows:

- By leveraging DRL's capabilities, we propose DRL-STAF, which is, to the best of our knowledge, the first distribution-free framework for complex multivariate hidden Markov processes. In this framework, deep networks model emissions and a DRL policy directly estimates discrete hidden states without likelihood-based inference. By avoiding strong generative assumptions, DRL-STAF can flexibly capture time-varying state transitions, cross-variable interactions, and nonlinear emission patterns.

- We design a DRL-based state estimation module that enables hard decoding of latent states, yielding interpretable, state-consistent predictions. By formulating historical observations and previous state estimates as environment feedback, the module learns adaptive and nonlinear transition policies without relying on predefined transition structures, and supports fine-grained, time-varying control of latent state evolution.

- We develop a two-stage training scheme that avoids combinatorial explosion in multivariate settings. In the first stage, each variable independently trains a DRL-based estimator to infer its local hidden state, thus bypassing the need to model the full joint state space. In the second stage, we integrate these estimators via a graph-based coordination mechanism, which enables efficient joint state inference by capturing cross-variable interactions without explicitly enumerating all state combinations.

## 2 RELATED WORK

**Classical and Extended HMMs** HMMs are foundational probabilistic models for sequential data, which provides a principled framework for capturing temporal dependencies and hidden dynamics (Mor et al., 2021). The basic first-order HMM (Bansal & Zhou, 2025) assumes that the next state depends only on the current state and that emissions follow simple parametric distributions. Then, to capture more complex dynamics, several extensions have been proposed. For example, Higher-order HMM (HOHMM) (Rodriguez-Fernandez et al., 2017) allows dependencies on several past states. Hidden Semi-Markov Model (HSMM) (Lin et al., 2022) explicitly models non-exponential state durations. Coupled HMM (CHMM) (Wang et al., 2019) models interactions among hidden states of multiple sequences. These HMM variants extend the applicability of the original model. However, the prediction performance of them remains limited due to the reliance on simple emission distributions.

However, these extensions still suffer from several limitations. First, they typically rely on simple, predefined emission distributions (e.g., Gaussian), which limits their ability to represent complex, nonlinear observation patterns. Second, although they introduce more expressive latent dynamics, they generally assume fixed and manually specified transition structures, which are insufficient for modeling nonstationary or time-varying transitions. Third, in multivariate settings, explicitly modeling the joint hidden state space leads to exponential growth in complexity (i.e., $\mathcal{O}(m^N)$ for $N$ variables each with $m$ states), rendering exact inference computationally intractable.

**Inference in HMM** A wide range of statistical inference techniques have been developed for learning HMM parameters and decoding hidden states. The Expectation-Maximization (EM) algorithm is the standard approach but often converges to local optima and scales poorly with model size (You & Oechtering, 2023). Variational inference methods (Lan et al., 2023) and mean-field approximations (Celeux et al., 2003) offer improved scalability, but rely on strong independence assumptions that may reduce accuracy. Sampling-based methods such as Gibbs sampling and particle filtering (Tripuraneni et al., 2015) provide more flexible posterior inference, yet are computationally intensive and often impractical for large-scale applications. Furthermore, while soft decoding minimizes expected prediction error by averaging over posterior state distributions, it often blurs latent regime boundaries. In contrast, hard decoding assigns the most likely state at each step, yielding clearer interpretability and more accurate, state-consistent predictions (Seshadri & Sundberg, 1994). Despite these advances, the inference complexity of extended HMMs—especially in multivariate and nonlinear contexts—remains a major bottleneck.

**Deep Learning–HMM Hybrids** To overcome the limitations of parametric emissions, recent works have integrated deep learning with HMMs. For example, CD-DNN-HMM (Dahl et al., 2012) builds a deep neural network to model posterior probabilities over HMM states while retaining the HMM transition structure and likelihood-based training. Markovian RNNs (Ilhan et al., 2023) embed HMM-style latent state switching into recurrent neural architectures to better capture nonstationary transitions. DEN-HMM (Bansal & Zhou, 2025) replaces the emission distributions of HMMs with deep neural networks, allowing for more flexible observation modeling and partially enabling time-varying transitions. NHMM (Tran et al., 2016) parameterizes both transition and emission potentials with neural networks while still performing unsupervised learning via HMM marginal-likelihood maximization and forward–backward inference. NCTRL (Song et al., 2023) models a discrete nonstationary latent process together with time-delayed causal latent dynamics, using neural networks for nonlinear mixing and transitions but still relying on likelihood-based generative modeling and inference. These models demonstrate the benefit of combining DL with HMMs, especially in univariate scenarios. However, existing DL–HMM hybrids remain fundamentally likelihood-driven, limiting their adaptability to dynamic latent behaviors. Moreover, their reliance on marginal likelihood leads to objective mismatch, inference gaps, and unstable optimization when combined with deep nonlinear emissions. More importantly, these approaches remain focused on univariate sequences and do not scale naturally to multivariate hidden processes, where state space complexity poses a serious challenge.

In addition, most existing DL–HMM methods rely on soft decoding, which—while optimizing squared error—tends to blur regime boundaries and underestimate volatility. Hard decoding, though more interpretable and often more accurate, has rarely been adopted due to its non-differentiability, which complicates end-to-end training in deep learning pipelines. Finally, although deep mod-

els enhance the flexibility of emission and transition parameterization, existing DL-HMM hybrids remain tied to likelihood-based generative inference and do not fundamentally overcome the structural limitations imposed by complex latent transitions and combinatorial state spaces. To provide an intuitive comparison, Figure 1 illustrates the differences among classical and extended HMMs, DL-HMM hybrids, and the proposed DRL-STAF.

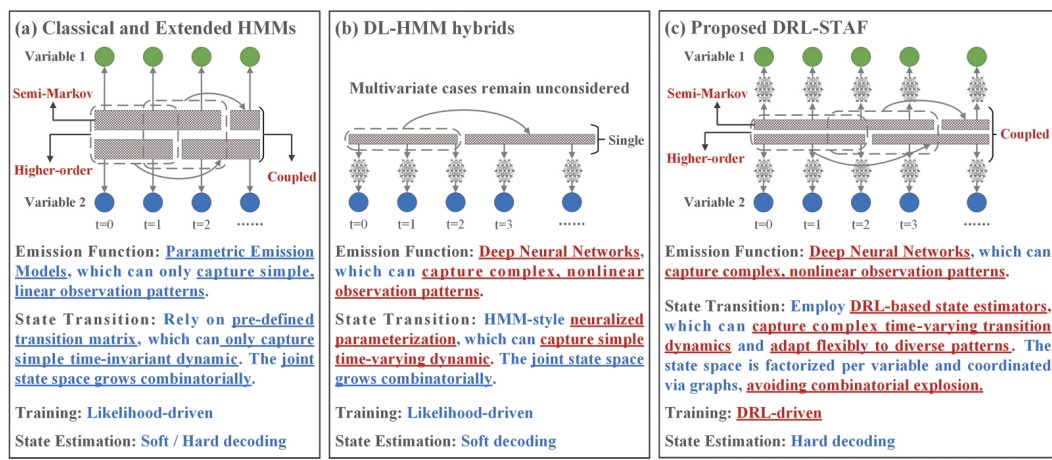

Figure 1: Comparison among classical and extended HMMs, DL-HMM hybrids, and the proposed DRL-STAF.

## 3 METHODOLOGY

### 3.1 DEEP MULTIVARIATE HIDDEN MARKOV PROCESS

We define the Deep Multivariate Hidden Markov Process (DM-HMP) as a general framework for modeling multivariate sequential data with latent state dynamics, in which both emissions and state transitions are parameterized by deep functions.

Consider $N$ observable sequences $\mathbf{X} = \{\mathbf{x}_1, \ldots, \mathbf{x}_N\}$, where the observation vector at each time step $t$ is denoted as $\mathbf{x}_t = (x_{1,t}, \ldots, x_{N,t}) \in \mathbb{R}^N$. Each variable $i$ is associated with a finite set of hidden states $S_c = \{1, \ldots, m\}$, and the hidden state of variable $i$ at time $t$ is denoted by $s_{i,t} \in S_c$. Note that hidden states are not directly observable and must be inferred.

In DM-HMP, the generative process consists of two components. First, the emission process specifies how observations are generated from hidden states. For variable $i$, the observation sequence at time $t$ is produced as

$$x_{i,t} = \mathcal{F}_{s_{i,t}}(\mathcal{H}_{i,t}) + \epsilon_{i,t}, \tag{1}$$

where $\mathcal{F}_{s_{i,t}}(\cdot)$ is a state-specific deep emission function, $\mathcal{H}_{i,t}$ denotes the input to the deep emission function of variable $i$ at time $t$, consisting of the observations and the exogenous covariates, and $\epsilon_{i,t}$ is random noise. Different hidden states correspond to distinct emission functions, which characterize heterogeneous and nonlinear observation patterns.

Second, the transition process governs the evolution of hidden states. For variable $i$, the transition probability at time $t$ is defined as

$$P(s_{i,t} \mid \mathbf{s}_{t-k:t-1}, d_{i,t}) = \mathcal{G}(\mathbf{s}_{t-k:t-1}, d_{i,t}), \tag{2}$$

where $\mathbf{s}_{t-k:t-1}$ denotes the historical states of all variables from time $t - k$ to $t - 1$, and $d_{i,t}$ represents the duration of the current state. The transition function $\mathcal{G}(\cdot)$ is parameterized by deep neural networks, allowing the hidden states to evolve with nonlinear, time-varying, and cross-variable dependent dynamics.

In multivariate state-aware forecasting, the objective is to predict the next-step observation $\mathbf{x}_{t+1} \in \mathbb{R}^N$ while simultaneously estimating the hidden states $S_{t+1} = \{s_{t+1,1}, \ldots, s_{t+1,N}\}$, which determine the corresponding emission functions and directly affect prediction accuracy. This DM-HMP formulation forms the basis of our proposed DRL-STAF framework.

## 3.2 OVERALL ARCHITECTURE

We propose a state-aware forecasting framework of DM-HMP, namely DRL-STAF. As shown in Figure 2, DRL-STAF combines deep learning (DL) for observation modeling with deep reinforcement learning (DRL) for state inference. Specifically, the state estimation module employs DRL-based estimators that perform hard decoding and capture nonlinear, time-varying state transitions. Together with the prediction module, DRL-STAF achieves accurate and state-consistent forecasting.

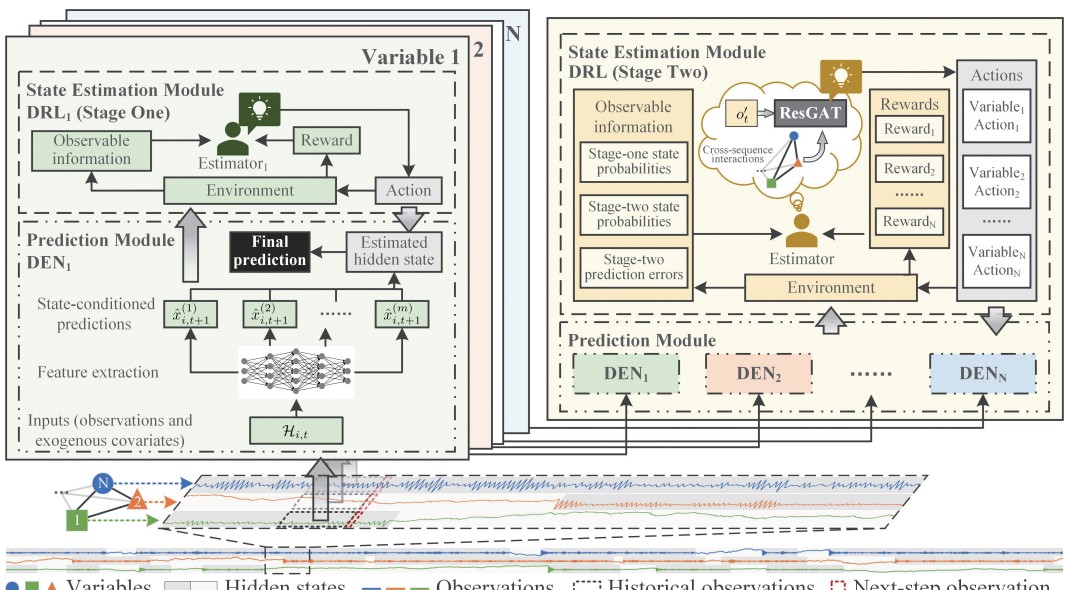

Figure 2: Overall structure of DRL-STAF.

### 3.2.1 HARD POSTERIOR STATE ESTIMATION MODULE

We formulate state estimation as a sequential decision-making problem with $m$ discrete actions, each one corresponding to a candidate hidden state. For variable $i$, the policy $\pi_{i,\theta}$ produces a categorical distribution over hidden states based on a set of observable information $o_{i,t}$, such as historical state probabilities and state-conditioned prediction errors, which can be expressed as:

$$P_{i,t+1}(s) = \pi_{i,\theta}(s|o_{i,t}). \tag{3}$$

Then, under hard decoding, the estimated hidden state $\hat{s}_{i,t+1}$ is obtained by selecting the most probable action:

$$\hat{s}_{i,t+1} = \arg\max_{s \in S_c} P_{i,t+1}(s) = \arg\max_{s \in S_c} \pi_{i,\theta}(s \mid o_{i,t}). \tag{4}$$

And the corresponding hard one-step-ahead prediction is

$$\hat{x}_{i,t+1}^{(\hat{s}_{i,t+1})} = \mathcal{F}_{\hat{s}_{i,t+1}}\left(\mathcal{H}_{i,t}\right). \tag{5}$$

To optimize the policy $\pi_{i,\theta}$, we design a DRL-based state estimator. Specifically, at each time step $t$, the agent receives observable information $o_{i,t}$ of variable $i$ and takes an action $a_{i,t}$ corresponding to the decision on hidden state $s_{i,t+1}$. Then, the hidden state changes according to the transition probability, leading to an updated environment. At this point, the quality of action $a_{i,t}$ can be evaluated by a reward $r_{i,t}$. Subsequently, the agent receives a new observable information $o_{i,t+1}$ from the updated environment. During the decision-making process, the agent aims to maximize the cumulative discounted rewards over $T_E$ steps, which is defined as $\max\left[\sum_{\beta=0}^{T_E-1} \gamma^{\beta} r_{i,t+\beta+1}\right]$, where $T_E$ is the episode length, $\beta$ denotes the look-ahead step, and $\gamma \in [0,1]$ is the discount factor reflecting the uncertainty of future rewards.

In practice, directly estimating the next-step hidden state is extremely challenging. However, given the state-conditioned predictions $\{\hat{x}_{i,t}^{(s)}\}_{s=1}^m$ from the prediction module, the corresponding mean

squared error (MSE) $\{e_{i,t}^{(s)}\}_{s=1}^m$, which is defined as $e_{i,t}^{(s)} = (\hat{x}_{i,t}^{(s)} - x_{i,t})^2$ can be obtained, and the current hidden state $s_{i,t}$ can be more reliably estimated. Therefore, assuming that hidden state transitions are infrequent, we let the hidden state remain unchanged from $t$ to $t+1$, that is, $\hat{s}_{i,t+1} \approx \hat{s}_{i,t}$. In this way, accurate estimation of the current hidden state serves as a reliable approximation for the next-step hidden state, thereby reducing the overall estimation error. Importantly, this assumption does not presume strict hidden state persistence; instead, under uncertainty it functions as a conservative default that prevents spurious rapid switching and yields more stable hidden state estimation.

A key component of the DRL-based estimator is the reward function, which directly guides the agent toward accurate state estimation. In our framework, the reward of variable $i$ at time $t$ is defined as $r_{i,t} = r_{i,t}^{\text{IR}} + \mathbf{1}_{\{t=T_E\}} r_i^{\text{ER}}$, where $r_{i,t}^{\text{IR}}$ denotes the immediate reward and $r_i^{\text{ER}}$ denotes the episodic reward. The immediate reward at each time step $t$ is designed from a local perspective, consisting of the prediction gain and the action switching penalty, and is defined as:

$$r_{i,t}^{\text{IR}} = \lambda_1 \underbrace{\left[ \alpha \left( e_{i,t+1}^{\text{base}} - e_{i,t+1}^{(a_{i,t})} \right) + (1-\alpha) \left( e_{i,t}^{\text{base}} - e_{i,t}^{(a_{i,t})} \right) \right]}_{\text{prediction gain}} + \lambda_2 \underbrace{\frac{\max\{0, \rho_c - c_{i,t}\}}{\rho_c - 1}}_{\text{action continuity penalty}}, \quad (6)$$

where $e_{i,t}^{(a_{i,t})}$ denotes the MSE of the prediction obtained by the prediction module based on action $a_{i,t}$, $e_i^{\text{base}}$ denotes the baseline MSE obtained from a predictor with a structure similar to the prediction module but without considering hidden states, $\alpha \in [0, 1]$ balances the emphasis between the accuracy of $\hat{s}_t$ and $\hat{s}_{t+1}$, $c_{i,t}$ is the length of consecutive identical actions, $\rho_c$ is the continuity threshold, $\lambda_1$ and $\lambda_2$ are the hyperparameters control the relative influence of the prediction gain term and the action continuity penalty term. Here, $e^{\text{base}} - e^{(a)}$ an advantage-style signal that measures the relative predictive gain of the selected state, encouraging the agent to focus on state-specific improvements rather than absolute errors. The action switching penalty term discourages only spurious rapid switching and serves mainly as a variance-reduction mechanism, without enforcing persistent states or restricting genuine hidden state changes.

At the end of an episode, an episodic reward is introduced from a global perspective to evaluate the overall quality of state estimation, consisting of the state separation objective and the pairwise discrepancy, which is defined as:

$$r_i^{\text{ER}} = \underbrace{-\frac{1}{m} \sum_{a_i=1}^m \bar{e}_i^{(a_i|a_i)} + \sum_{a_i=1}^m \left[ \max\{\Delta_i^{(a_i)}, 0\} + \lambda_3 \min\{\Delta_i^{(a_i)}, 0\} \right]}_{\text{state separation}} - \lambda_4 \underbrace{\frac{1}{2} \sum_{p \neq q} |\bar{e}_i^{(p|p)} - \bar{e}_i^{(q|q)}|}_{\text{pairwise discrepancy}},$$

$$(7)$$

where $\bar{e}_i^{(u|v)}$ denotes the mean of conditional MSE, i.e., $\bar{e}_i^{(u|v)} = (\sum_{t=1}^{T_E} \sum_{s=1}^m z_t^{(s|v)} e_t^{(u)})/(\sum_{t=1}^{T_E} \sum_{s=1}^m z_t^{(s|v)})$. Here, $z_t^{(s|v)} = 1$ if $s = v$, and 0 otherwise. $\Delta_i^{(a_i)} = \bar{e}_i^{(-a_i|-a_i)} - \bar{e}_i^{(a_i|a_i)}$ measures the performance advantage of the selected action over the unselected ones, where $\bar{e}_i^{(a_i|a_i)}$ denotes the mean MSE of the predictions under the selected action $a_i$, while $\bar{e}_i^{(-a_i|-a_i)}$ denotes the mean MSE under the other actions. $\frac{1}{2} \sum_{p \neq q} |\bar{e}_i^{(p|p)} - \bar{e}_i^{(q|q)}|$ measures the pairwise discrepancy in accuracy across different actions, $\lambda_3$ and $\lambda_4$ are the hyperparameters. The state separation term resembles specialization objectives in Mixture-of-Experts models. It evaluates within-state accuracy and across-state separation at the end of each episode, encouraging each state to fit its assigned samples while maintaining a clear performance margin over alternative states. The pairwise discrepancy term quantifies accuracy gaps across hidden states and thereby directs more attention to under-trained hidden states, preventing neglect and promoting more even learning across the predictors.

In summary, the immediate reward shapes step-wise state selection through prediction improvement and controlled switching behavior, while the episodic reward enforces hidden state consistency, balanced state quality, and meaningful specialization within our distribution-free RL formulation.

### 3.2.2 PREDICTION MODULE

The prediction module focuses on forecasting the next-step observation vector $x_{t+1}$. Inspired by the DEN-HMM proposed by (Bansal & Zhou, 2025), we construct an individual deep emission network

(DEN) for each variable to serve as the emission function. Specifically, each DEN is designed with $m$ output heads, where each head corresponds to a candidate hidden state. Given the inputs $\mathcal{H}_{i,t}$ of variable $i$, the $\text{DEN}_i$ outputs state-conditioned predictions $\{\hat{x}_{i,t+1}^{(s)}\}_{s=1}^m$. When the state $\hat{s}_{i,t+1}$ is estimated by the state estimation module, the final prediction can be generated as

$$\hat{x}_{i,t+1}^{(\hat{s}_{i,t+1})} = \mathcal{F}_{\hat{s}_{i,t+1}}(\mathcal{H}_{i,t}) = \sum_{s=1}^m z_{i,t+1}^{(s|\hat{s}_{i,t+1})} \hat{x}_{i,t+1}^{(s)}. \tag{8}$$

## 4 TWO-STAGE TRAINING SCHEME

The training of our framework is conducted in a two-stage scheme: in the first stage, state estimation is performed independently for each variable without considering cross-sequence interactions; in the second stage, cross-sequence interactions are incorporated to refine the state estimates obtained in the first stage and achieve more accurate estimation. Such a design decomposes the joint estimation into per-variable learning and graph-based coordination, which ensures scalability in multivariate settings and effectively avoids the combinatorial explosion of the joint state space.

### 4.1 STAGE ONE: INDEPENDENT TRAINING FOR EACH VARIABLE

For each variable, the observation sequence is split into a training set and a test set. In the DRL framework, the training process is formulated as episodic interactions. Here, each episode is assigned the same time horizon $T_E$, while the starting index is randomly sampled from the training set to introduce randomness. The episode begins with the initialization of the relevant parameters and the inputs of both modules. For the prediction module, the input of variable $i$ at step $t$ is the $\mathcal{H}_{i,t}$. Through DEN $\pi_{i,\theta_E}$, the output is the state-conditioned predictions $\{\hat{x}_{i,t}^{(s)}\}_{s=1}^m$, from which the corresponding prediction errors $\{e_{i,t}^{(s)}\}_{s=1}^m$ can be calculated. For the state estimation module in stage one, the input at step $t$ is the observable information $o_t$, which consists of $\mathcal{H}_{i,t}$, the state probabilities $p_{i,t-T_1:t-1} \in \mathbb{R}^{T_1 \times m}$ over the past $T_1$ steps, and the prediction errors $e_{i,t-T_1+1:t} \in \mathbb{R}^{T_1 \times m}$ under different states during the past $T_1$ steps. Given $o_{i,t}$, the policy network $\pi_{i,\theta_A}$ outputs a probability distribution $p_{i,t}$ of hidden states at step $t$, from which the agent samples an action $a_{i,t}$ corresponding to the estimated hidden state $\hat{s}_{i,t}$. The detailed structure of $\pi_{i,\theta_A}$ is illustrated in Appendix A (Figure 4). Based on the assumption of infrequent state transitions, the next-step state is approximated by $\hat{s}_{i,t+1} \approx \hat{s}_{i,t}$. After taking the action $a_{i,t}$, the environment returns a reward $r_{i,t}$, which evaluates the quality of the state estimation at step $t$. The observable information $o_{i,t}$ is then updated, and the procedure continues to the next step $t+1$. An episode terminates once the maximum time horizon $T_E$ is reached. At this point, the parameters of both modules are updated. The prediction module is optimized by minimizing the MSE of $\{\hat{e}_{i,t}^{(\hat{s}_{i,t})}\}_{t=1}^{T_E}$, while the state estimation module is optimized by maximizing the cumulative rewards.

It is worth noting that the training of the two modules is highly interdependent. The prediction module requires accurate state estimates as inputs, while the state estimation module relies on reliable prediction errors to compute rewards. In the early stage of training, inaccurate outputs from either module may accumulate and hinder the convergence of the framework. To address this problem, we design a sample screening strategy and introduce the soft update strategy, which aims to reduce the adverse impact of inaccurate predictions and state estimates.

The purpose of the sample screening strategy is to select high-quality training samples for updating the prediction module, so that more reliable predictions can be obtained, which in turn facilitates the effective training of the state estimation module. The key intuition is that, if the state estimation is correct, the prediction error under the selected action is likely to be the smallest among all candidate states. Thus, to quantify the reliability of the action $a_{i,t}$ taken at step $t$, a confidence score is defined as:

$$c_{i,t} = \min_{s \neq a_{i,t}} \{e_{i,t}^{(s)}\} - e_{i,t}^{(a_{i,t})}. \tag{9}$$

Here, a larger $c_{i,t}$ indicates that the selected action $a_{i,t}$ is more reliable. Based on $c_{i,t}$, we can first discard samples with negative confidence scores ($c_{i,t} < 0$), since this indicates that the selected action is likely incorrect. To avoid insufficient training of some output heads in DEN, we retain the top-$K_{sup}$ samples with the highest $c_t$ when no samples with $c_{i,t} > 0$ are available. In practice,

even when the system is actually in state $s$, the prediction accuracy of the corresponding output head $H_{i,s}$ may still be lower than that of other heads, especially when the training of $H_{i,s}$ is not yet sufficient. Thus, filtering samples solely on the criterion $c_{i,t} > 0$ may still retain low-quality samples. To further improve the reliability of training data for the prediction module, we introduce an action continuity criterion, which prioritizes samples with more continuous actions. Specifically, two thresholds $\phi_H$ and $\phi_L$ are introduced and set with $\phi_H > \phi_L$. The samples are further segmented according to consecutive actions. Only segments with a length greater than $\phi_H$ are retained, and if none are available, those with a length greater than $\phi_L$ are retained instead.

To further mitigate the impact of inaccurate state estimations on the prediction module, we adopt a soft update strategy for updating the parameters of $\text{DEN}_i$, defined as

$$\theta_E \leftarrow \tau\theta_E + (1 - \tau)\theta'_E, \tag{10}$$

where $\tau \in (0, 1)$ is the update rate.

## 4.2 STAGE TWO: CROSS-SEQUENCE DEPENDENT TRAINING

Once the prediction module and the state estimation module have been fully trained in stage one, their parameters are frozen. Then, the state probabilities generated by the state estimation module in stage one are used as inputs to the new state estimation module in stage two, while the rewards generated in stage one serve as the baseline for evaluating the quality of second-stage state estimation. Specifically, for the new state estimation module in stage two, the input at step $t$ is new observable information $o'_t$, which consists of stage-one state probabilities $P_t = [p_{1,t}, p_{2,t}, \ldots, p_{N,t}]$, the stage-two state probabilities $P'_{t-T_2:t-1} == [p'_{1,t-T_2:t-1}, p'_{2,t-T_2:t-1}, \ldots, p'_{N,t-T_2:t-1}] \in \mathbb{R}^{T_2 \times N \times m}$ over the past $T_2$ steps, and the stage-two prediction errors $E'_{t-T_2+1:t} \in \mathbb{R}^{T_2 \times N \times m}$ during the past $T_2$ steps. Given $o'_t$, the policy network $\pi'_{\theta_{A'}}$ outputs probability distributions of hidden states for all variables at step $t$. The structure of $\pi'_{\theta_{A'}}$, illustrated in Appendix A (Figure 5), mainly consists of a confidence adjustment layer and a residual graph attention (ResGAT) layer. The confidence adjustment layer rescales the state probabilities generated in stage one as $\widetilde{P}_t = \mu \cdot P_t$, where $\mu = [\mu_1, \mu_2, \ldots, \mu_N]$ denotes the confidence factors for the stage-one estimations of all variables. The ResGAT layer is designed to capture cross-sequence interactions among hidden states. For each pair of variables $(i, j)$ at step $t$, the unnormalized attention coefficient is computed as $e_{ij,t} = LeakyReLU(a^T[Wh_{i,t} || Wh_{j,t}])$, where $h_{i,t}$ and $h_{j,t}$ are the features of variables $i$ and $j$, respectively, $W$ is the weight matrix, $a$ is the attention vector, and $||$ denotes concatenation. The normalized attention weight is then obtained via $\alpha_{ij,t} = \exp(e_{ij,t})/\sum_{j' \in \mathcal{N}_i} \exp(e_{ij',t})$. After that, the aggregated feature representation for head $d$ is defined as:

$$h_{i,t}^{(d)} = \sum_{j \in \mathcal{N}_i} \alpha_{ij,t}^{(d)} W^{(d)} h_{j,t}. \tag{11}$$

The multi-head outputs can be merged following $1/D \sum_{d=1}^{D} h_{i,t}^{(d)}$ or $h_{i,t}^{Merge} = W \big|\big|_{d=1}^{D} h_{i,t}^{(d)}$, where $W$ is a projection matrix that maps the concatenated vector back to the original feature dimension. With residual connection, the updated feature representation is $h'_{i,t} = \sigma\left(h_{i,t}^{Merge} + h_{i,t}\right)$, where $\sigma(\cdot)$ is a nonlinear activation function. Finally, the stage-two state probabilities are defined as:

$$P'_t = softmax(\mu \cdot P_t + W_p \cdot H'_t), \tag{12}$$

where $W_p$ is the weighted matrix, and $H'_t = [h'_{1,t}, \ldots, h'_{N,t}]$ denotes the updated feature representations of all variables.

Based on the probability distributions, the agent samples a set of actions $[a'_{1,t}, \ldots, a'_{N,t}]$, corresponding to the estimated hidden states $[\hat{s}'_{1,t}, \ldots, \hat{s}'_{N,t}]$. After taking the actions, the environment returns a set of rewards $[r'_{1,t}, \ldots, r'_{N,t}]$. For improved guidance in stage two, we refine the reward for variable $i$ as $r_{i,t}^{\delta} = r'_{i,t} - r_{i,t}$, which captures the gain over the stage one baseline. Similar to stage one, the observable information $o'$ is updated step by step, and an episode terminates once the maximum time horizon $T_E$ is reached. At the end of each episode, only the second-stage state estimation module is updated, so as to prevent inaccurate state estimations from disrupting the already trained prediction module. The prediction module is updated again based on the state estimates from the second stage only when the number of episodes is sufficiently large and $r_{i,t}^{\delta} > 0$.

# 5 EXPERIMENTS

We extensively evaluate the proposed DRL-STAF on multivariate hidden markov process datasets with state transitions to validate both its predictive accuracy and its capability of state estimation.

**Datasets** We employ four simulated dataset and three real-world datasets in our experiments. The simulated dataset is generated based on an AR process with Coupled Higher-order Semi-Markov Model (CHOSMM) based state transitions, and the detailed parameter settings are provided in Appendix B. The real-world datasets include a server machine (SMachine) dataset, an exchange rate (Exchange) dataset, and a traffic network (Traffic) dataset, with detailed descriptions given in Appendix C.

**Baselines** We choose six well-acknowledged models as our benchmark, including (1) HMM and its variants: Parallel HMM, Parallel HSMM, Parallel HOHMM, CHMM; and (2) DL-HMM hybrids: NHMM (Tran et al., 2016), NCTRL (Song et al., 2023), Markovian-RNN (Ilhan et al., 2023), DEN-HMM (Bansal & Zhou, 2025). A detailed comparison of the parameter complexity and computational cost of all competing models is provided in Appendix E.

## 5.1 MAIN RESULTS

We evaluate the empirical results from two complementary perspectives: forecasting performance and state estimation performance. Forecasting performance is measured by Mean Average Error (MAE) and MSE, where smaller values indicate better predictive performance. State estimation performance is evaluated using accuracy, precision, recall, and F1 score, where larger values indicate more reliable state estimation. The best results are highlighted in **red** and the second-best results are underlined. The complete results can be found in Appendix D.

**Results on simulated datasets** Table 1 reports the results on simulated datasets with 3 and 10 variables. DRL-STAF consistently achieves the best performance across all evaluation metrics. In particular, DRL-STAF obtains higher accuracy and F1 scores in state estimation, while simultaneously reducing MAE and MSE for forecasting. These results demonstrate the effectiveness of simultaneously optimizing forecasting and state estimation. Furthermore, Figure 3 illustrates partial results of DRL-STAF on the dataset with 3 variables. For each variable, the upper part shows both true and predicted observations, while the lower part shows both true and estimated states, where different background colors indicate true states and solid lines denote estimated states. It is clear that DRL-STAF achieves accurate forecasting and reasonable state estimation across variables. We also evaluate on datasets with frequent transitions, where DRL-STAF remains robust while most likelihood-driven methods degrade markedly. This further shows that components motivated by the infrequent-transition assumption do not enforce strict state persistence in practice. Full results are given in Appendix D.

Table 1: Forecasting and state estimation results on simulated datasets with infrequent transitions.

| Models | 3 variables | | | | | | 10 variables | | | | | |
|---|---|---|---|---|---|---|---|---|---|---|---|---|
| | Accuracy | Precision | Recall | F1 | MAE | MSE | Accuracy | Precision | Recall | F1 | MAE | MSE |
| Parallel HMM | 74.87% | 69.44% | **100.0%** | 81.86% | 0.5861 | 0.5306 | 70.95% | 61.68% | 68.41% | 61.75% | 0.5710 | 1.0661 |
| Parallel HSMM | 60.33% | 42.40% | 66.67% | 51.79% | 0.6210 | 0.6433 | 77.09% | 48.28% | 60.00% | 52.91% | 0.6890 | 1.5272 |
| Parallel HOHMM | 74.13% | 69.19% | 98.47% | 81.18% | 0.5846 | 0.5222 | 68.39% | 63.65% | 57.82% | 57.24% | 0.5500 | 0.9784 |
| CHMM | 74.83% | 69.42% | **100.0%** | 81.84% | 0.5845 | 0.5301 | 75.16% | 61.64% | 73.87% | 64.51% | 0.6274 | 1.2249 |
| NHMM | 60.32% | 49.06% | 66.65% | 51.81% | 0.1277 | 0.0390 | 75.55% | 51.33% | 62.00% | 55.11% | 0.2621 | 0.1600 |
| NCTRL | 79.53% | 88.49% | 79.15% | 81.34% | 0.3936 | 0.3042 | 90.31% | 88.92% | 86.71% | 86.65% | 0.3431 | 0.3935 |
| Markovian-RNN | 57.70% | 44.37% | 50.28% | 46.79% | 0.2444 | 0.0935 | 68.71% | 61.18% | 66.36% | 60.93% | 0.1211 | **0.0355** |
| DEN-HMM | 60.33% | 42.40% | 66.67% | 51.79% | 0.7131 | 0.7749 | 75.89% | 53.03% | 69.30% | 58.72% | 0.7247 | 1.6617 |
| DRL-STAF | **98.17%** | **96.89%** | 99.62% | **98.22%** | **0.0889** | **0.0278** | **96.15%** | **95.44%** | **91.89%** | **93.26%** | **0.1090** | 0.0395 |

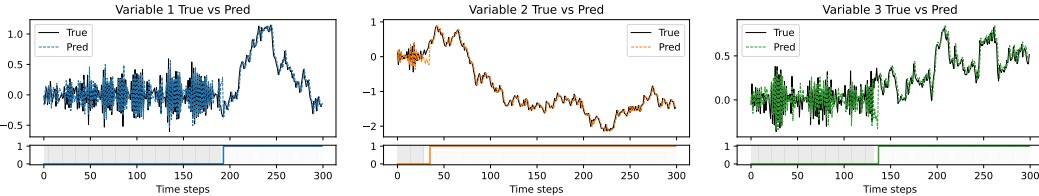

Figure 3: Partial results of DRL-STAF on the dataset with 3 variables.

**Results on real-world datasets** Since the SMachine dataset contains labels for anomalous states, we evaluate both forecasting performance and state estimation performance. In contrast, the Ex-

change dataset and the Traffic dataset do not provide state labels, so we only evaluate forecasting performance. Table 2 presents the comparison results for both datasets. On the SMachine dataset, DRL-STAF achieves best Accuracy, Precision, F1, MAE and MSE. On the Exchange dataset and the Traffic dataset, DRL-STAF achieves the lowest MAE and MSE among all methods. These results demonstrate that DRL-STAF can effectively balance forecasting and state estimation, achieving strong and consistent performance across real-world datasets.

Table 2: Forecasting results on real-world datasets.

| Models | SMachine dataset | | | | | | Exchange dataset | | Traffic dataset | |
|---|---|---|---|---|---|---|---|---|---|---|
| | Accuracy | Precision | Recall | F1 | MAE | MSE | MAE | MSE | MAE | MSE |
| Parallel HMM | 76.37% | 55.86% | **90.95%** | 69.21% | 0.0563 | 0.0056 | 8.6701 | 294.0451 | 5.5411 | 59.5501 |
| Parallel HSMM | 77.16% | 61.50% | 58.19% | 59.80% | 0.0968 | 0.0157 | 9.9193 | 366.7165 | 14.1709 | 259.8108 |
| Parallel HOHMM | 76.37% | 55.86% | **90.95%** | 69.21% | 0.0563 | 0.0056 | 8.5905 | 294.0307 | 5.5419 | 59.5004 |
| CHMM | 76.02% | 55.52% | 89.73% | 68.60% | 0.0563 | 0.0056 | 8.5851 | 293.5766 | 5.1245 | 50.2464 |
| NHMM | 70.77% | 0.00% | 0.00% | 0.00% | 0.0194 | **0.0010** | 4.4926 | 100.0099 | 1.5881 | 6.7800 |
| NCTRL | 72.54% | 54.04% | 41.66% | 42.33% | 0.0360 | 0.0023 | 1.8486 | 13.6292 | 2.5353 | 12.8844 |
| Markovian-RNN | 70.81% | 0.00% | 0.00% | 0.00% | 0.0190 | **0.0010** | 2.5361 | 28.1406 | 2.0032 | 9.9143 |
| DEN-HMM | 68.59% | 37.40% | 11.25% | 17.29% | 0.1537 | 0.0386 | 11.5619 | 470.4355 | 13.8156 | 245.5208 |
| DRL-STAF | **81.73%** | **100.00%** | 63.27% | **77.50%** | **0.0189** | **0.0010** | **1.6438** | **13.2381** | **1.5193** | **6.4610** |

## 5.2 ABLATION STUDIES

To better understand the necessity of state-aware forecasting and modeling hidden-state interactions on multivariate Hidden Markov Process datasets, we conduct ablation studies with two variants. The first is DRL-S1, which corresponds to the first-stage model without considering interactions among variables. The second is DL-F, the deep learning model that does not consider hidden states. As shown in Table 3, DL-F performs the worst across all metrics, which suggests that time series with evident state transitions cannot be effectively modeled without considering hidden states. In comparison, DRL-S1 already delivers competitive forecasting and state estimation performance, validating the effectiveness of our basic design. Building on this, DRL-STAF achieves further improvements by incorporating variable interactions in the second stage. This indicates that an increased number of variables introduces additional informative signals, and that effectively capturing and utilizing these cross-variable dependencies is critical for attaining further performance improvements in state-aware forecasting. We also conduct additional ablation studies to further demonstrate the necessity of each component in the DRL-STAF framework. Full results are provided in Appendix D.

Table 3: Comparison of ablation results on simulated and real-world datasets.

| Datasets | Simulated dataset (3 variables) | | | Simulated dataset (10 variables) | | | SMachine dataset | | | Exchange dataset | | | Traffic dataset | | |
|---|---|---|---|---|---|---|---|---|---|---|---|---|---|---|---|
| Models | DL-F | DRL-S1 | DRL-STAF | DL-F | DRL-S1 | DRL-STAF | DL-F | DRL-S1 | DRL-STAF | DL-F | DRL-S1 | DRL-STAF | DL-F | DRL-S1 | DRL-STAF |
| Accuracy | - | 97.53% | **98.17%** | - | 95.79% | **96.15%** | - | **85.80%** | 81.73% | - | - | - | - | - | - |
| Precision | - | 96.23% | **96.89%** | - | 94.09% | **95.44%** | - | 82.41% | **100.0%** | - | - | - | - | - | - |
| Recall | - | 99.22% | **99.62%** | - | **93.23%** | 91.89% | - | **65.28%** | 63.27% | - | - | - | - | - | - |
| F1 | - | 97.68% | **98.22%** | - | **93.43%** | 93.26% | - | 72.85% | **77.50%** | - | - | - | - | - | - |
| MAE | 0.2599 | 0.0956 | **0.0889** | 0.3646 | 0.1129 | **0.1090** | 0.0206 | 0.0196 | **0.0189** | 3.5285 | 1.8412 | **1.7249** | 1.8387 | 1.5686 | **1.5193** |
| MSE | 0.1807 | 0.0418 | **0.0278** | 0.3153 | 0.0490 | **0.0395** | **0.0010** | **0.0010** | **0.0010** | 58.8505 | 18.4091 | **16.4848** | 7.6680 | 6.9626 | **6.4610** |

## 6 CONCLUSION

In this paper, we propose DRL-STAF, a distribution-free framework for state-aware forecasting of complex multivariate hidden Markov process. Rather than relying on likelihood-based inference, DRL-STAF couples deep emission modeling with a DRL policy that directly estimates discrete hidden states, enabling adaptive, time-varying transition dynamics and cross-variable interactions. Furthermore, we design a two-stage training scheme and adopt hard decoding in state estimation, enabling accurate inference of hidden states and state-consistent predictions. Extensive experiments on both simulated and real-world datasets demonstrate that DRL-STAF consistently outperforms representative baselines in forecasting accuracy and state estimation reliability. For future work, we will further improve the training efficiency of DRL-STAF, extending the framework to multi-step, and explore its extension to more general HMM variants.

### ETHICS STATEMENT

This work adheres to the ICLR Code of Ethics. Our study does not involve human subjects, sensitive personal information, or applications with immediate potential for harm. The datasets used are publicly available or synthetically generated, and all experiments comply with community standards of fairness, transparency, and research integrity. We are not aware of any conflicts of interest, sponsorship issues, or ethical risks associated with this research. The use of large language models (LLMs)

was restricted solely to polishing the writing, as described in Appendix H, and did not contribute to scientific content or results.

## REPRODICIBILITY STATEMENT

We have made extensive efforts to ensure reproducibility. The proposed framework, DRL-STAF, is described in detail in the main text, while the pseudocode and the architectures of key components are provided in Appendix A. Experimental setups, dataset descriptions, and evaluation metrics are reported in Section 5, Appendix B, and Appendix C. The simulated datasets are generated with clearly specified parameters, while the real-world datasets are publicly available with references provided. Source code is available in https://anonymous.4open.science/status/DRL-STAF-E28D

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

# A PROPOSED DRL-STAF DETAILS

To provide a clearer understanding of the proposed DRL-STAF framework, we include three pseudocode algorithms in Appendix A. Algorithm 1 presents the Stage One training process, where each variable is modeled independently and the prediction module (DEN) and the state estimation module (DRL agent) are jointly trained. Since inaccurate state estimates may easily cause error accumulation, we introduce a sample screening strategy to select reliable samples for updating the DEN. This auxiliary procedure is detailed in Algorithm 2. The detailed architecture of the stage-one policy network $\pi_{\theta_A}$ is provided in Figure 4.

Algorithm 3 then presents the Stage Two training process, where cross-sequence dependencies are incorporated to further refine the state estimates obtained in Stage One. The parameters of the prediction module and Stage One policies are frozen at the beginning, and the DEN is unfrozen only when the unfreeze condition is satisfied. This ensures stable training of the Stage Two DRL agent, while still permitting further improvements once sufficient gains are observed. The structure of the stage-two policy network $\pi'\theta A'$, which integrates a confidence adjustment layer and a residual graph attention (ResGAT) layer, is illustrated in Figure 5.

---

**Algorithm 1** Stage one: Independent training for each variable

---

**Require:** Number of states $m$, episode length $T_E$, discount $\gamma$, soft rate $\tau$ and other hyper-parameters
1: **Initialization:** initialize parameters $\theta_E$ of DEN, $\theta_A$ of DRL, and target copy $\theta'_E \leftarrow \theta_E$
2: **for** epoch $= 1, 2, \ldots$ **do**
3:     **Environment init:** randomly sample a length-$T_E$ segment from sequence, and set $t \leftarrow 0$
4:     Initialize DEN input $\mathcal{H}_t$, and compute $\{e_t^{(s)}\}_{s=1}^m$ based on known $x_t$
5:     Initialize DRL input $o_t \leftarrow \{\mathcal{H}_t, \ p_{t-T_3:t-1}, \ e_{t-T_3+1:t}\}$
6:     **for** Step $= 1 \rightarrow T_E$ **do**
7:         $p_t \leftarrow \pi_{\theta_A}(o_t)$
8:         Sample action $a_t \sim p_t$ ($a_t$ corresponding to the estimated hidden state $\hat{s}_t$)
9:         Approximate next hidden state: $\hat{s}_{t+1} \approx \hat{s}_t$
10:       Update history context $\mathcal{H}_t$
11:       compute state-conditioned predictions $\{\hat{x}_{t+1}^{(s)}\}_{s=1}^m$ by DEN
12:       Obtain $\hat{x}_{t+1}^{(\hat{s}_{t+1})}$ from based on $\hat{s}_{t+1}$
13:       Observe $x_{t+1}$ and compute immediate reward $r_{i,t}^{\text{IR}}$
14:       Update $o_{t+1} \leftarrow \{\mathcal{H}_{t+1}, \ p_{t-T_3+1:t}, \ e_{t-T_3+2:t+1}\}$
15:       Compute confidence score $c_t$
16:       $t \leftarrow t + 1$
17:     **end for**
18:     Compute episodic reward $r_i^{\text{ER}}$
19:     **Sample screening:** apply Algorithm 2 to build screened dataset $\mathcal{D}_{\text{scr}}$.
20:     **Update DEN:** train on $\mathcal{D}_{\text{scr}}$, and then soft update $\theta_E \leftarrow \tau \theta_E + (1 - \tau) \theta'_E$
21:     **Update DRL:** update $\theta_A$ by maximizing cumulative discounted rewards
22: **end for**

---

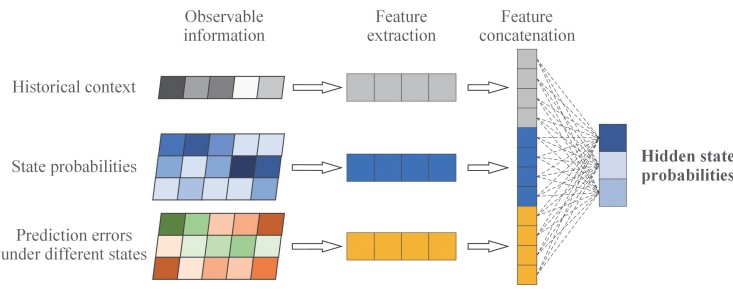

Figure 4: The detailed architecture of the stage-one policy network $\pi_{\theta_A}$.

---

**Algorithm 2** Sample screening strategy

---

**Require:** Action $a_t$ and confidence score $c_t$
1: Discard samples with negative confidence ($c_t < 0$)
2: **if** no sample remains **then**
3:      Keep top-$K_{\text{sup}}$ samples with largest $c_t$
4: **end if**
5: Segment samples by consecutive identical actions
6: **if** segments longer than $\phi_H$ exist **then**
7:      Keep those segments
8: **else if** segments longer than $\phi_L$ exist **then**
9:      Keep those segments
10: **end if**
11: Construct $\mathcal{D}_{\text{scr}}$ from retained samples

---

---

**Algorithm 3** Stage two: Cross-sequence dependent training

---

**Require:** Frozen $\{\theta_E^{(i)}\}$ and $\{\theta_A^{(i)}\}$, episode length $T_E$, discount $\gamma$, soft rate $\tau$ and other hyper-parameters
1: **Initialization:** initialize $\theta_{A'}$ of DRL in stage two, and initialize unfreeze flags = False
2: **for** epoch $= 1, 2, \ldots$ **do**
3:      **Environment init:** randomly sample a length-$T_E$ segment from sequence, and set $t \leftarrow 0$
4:      **for** Step $= 1 \rightarrow T_E$ **do**
5:          Obtain Stage one outputs: $P_t = [p_{1,t}, \ldots, p_{N,t}]$
6:          Build stage-two input $o'_t \leftarrow \{P_t, P'_{t-T_4:t-1}, E'_{t-T_4+1:t}\}$
7:          $P'_t \leftarrow \pi_{\theta'_A}(o'_t)$
8:          Sample actions $[a'_{1,t}, \ldots, a'_{N,t}] \sim P'_t$
9:          Approximate next hidden state: $\hat{s}'_{t+1} \approx \hat{s}'_t$
10:         Compute immediate rewards $[r_{1,t}^{\text{IR}'}, \ldots, r_{N,t}^{\text{IR}'}]$ and obtain relative gains $r_t^{\text{IR},\delta} \leftarrow r_t^{\text{IR}'} - r_t^{\text{IR}}$
11:         Update $o'_{t+1} \leftarrow \{P_{t+1}, P'_{t-T_4+1:t}, E'_{t-T_4+2:t+1}\}$
12:         Compute confidence score $[c_{i,t}, \ldots, c_{N,t}]$
13:         $t \leftarrow t + 1$
14:      **end for**
15:      Compute episodic reward $r_t^{\text{ER},\delta} \leftarrow r_t^{\text{ER}'} - r_t^{\text{ER}}$
16:      **for** variable $i = 1, 2, \ldots, N$ **do**
17:          **if** $\sum_{t=0}^{T_M} r_{i,t}^{\delta} > 0$ **or** unfreeze flag$_i$ = True **then**
18:             **Sample screening:** apply Algorithm 2 to build screened dataset $\mathcal{D}_{\text{scr}}^{(i)}$.
19:             **Update DEN$_i$:** train on $\mathcal{D}_{\text{scr}}^{(i)}$, and then soft update $\theta_E^{(i)} \leftarrow \tau\,\theta_E^{(i)} + (1-\tau)\,\theta_E^{(i)'}$
20:          **end if**
21:      **end for**
22:      **Update DRL in stage two:** update $\theta_{A'}$
23: **end for**

---

## B    SIMULATED DATASET DESCRIPTIONS

To evaluate the effectiveness of the proposed method, we construct simulated datasets based on Coupled Higher-Order Semi-Markov State Processes (CHOSMMs). This framework explicitly incorporates higher-order state transitions, inter-variable coupling, and semi-Markov sojourn times, while the observed series are generated through state-dependent emission functions.

### B.1    MODEL DESCRIPTION

Given historical multivariate time series $\mathbf{X} = \{\mathbf{x}_1, \ldots, \mathbf{x}_T\} \in \mathbb{R}^{T \times N}$, we assume that each variable $i \in \{1, \ldots, N\}$ is governed by a hidden state sequence $S_c = 1, 2, \ldots, m$.

Figure 5: The detailed architecture of the stage-two policy network $\pi'_{\theta_{A'}}$.

**Higher-order transitions** We adopt a second-order transition rule. The base transition probability of variable $i$ is

$$\pi_{i,t}^{\text{base}}(u) = \Pr(s_{i,t} = u | s_{i,t-2} = a, s_{i,t-1} = b) \tag{13}$$

$$= \frac{\Psi_i[a, b, u]}{\sum_{v=1}^{m} \Psi_i[a, b, v]}, \tag{14}$$

where $\Psi_i \in \mathbb{R}^{m \times m \times m}$ is the transition tensor of variable $i$.

**Inter-variable coupling** To model interactions among variables, we introduce an adjacency matrix $A \in \{0, 1\}^{N \times N}$ (with zero diagonals) and a coupling strength $\eta \geq 0$. The coupled logit for variable $i$ is

$$\ell_{i,t}(u) = \log \pi_{i,t}^{\text{base}}(u) + \eta \sum_{j=1}^{N} A_{ij} \mathbf{1}\{s_{j,t-1} = u\}, \tag{15}$$

and the final transition distribution is

$$\pi_{i,t}(u) = \frac{\exp\left(\ell_{i,t}(u)\right)}{\sum_{v=1}^{m} \exp\left(\ell_{i,t}(v)\right)}. \tag{16}$$

**Semi-Markov durations** Unlike standard HMMs, CHOSMM explicitly models the sojourn time. When variable $i$ enters state $u \in S_c$, its duration is sampled from a distribution

$$\tau \sim D_i(u), \quad \tau \geq 1, \tag{17}$$

and the state remains fixed for $\tau$ consecutive steps before the next transition is drawn.

**State-dependent emissions** Given the hidden state $s_{i,t}$, the observation $x_{i,t}$ is generated by the corresponding emission function $f_{s_{i,t}}$. In our setting, each emission function is an autoregressive model of order $P$:

$$x_{i,t} = \sum_{p=1}^{P} a_{s_{i,t},i}^{(p)} x_{i,t-p} + \varepsilon_{i,t}, \quad \varepsilon_{i,t} \sim \mathcal{N}(0, \sigma_i^2). \tag{18}$$

## B.2 EXPERIMENTAL SETUP

In the experiments, we adopt the following settings to generate simulated datasets:

- **Emission model:** autoregressive order $P = 1$, with coefficients $a_{1,i}^{(1)} = 1.0$, $a_{2,i}^{(1)} = -0.9$, $\sigma_i = 0.1$.
- **Transition patterns:** five distinct second-order transition tensors $\Psi^{(k)}$ are pre-defined (see Table 4).
- **Duration distributions:** ten candidate sojourn-time distributions are pre-defined (see Table 5).

- **Coupling:** the coupling strength is fixed as $\eta = 0.2$.

The simulator outputs $X = \{x_1, \ldots, x_T\} \in \mathbb{R}^{T \times N}$, and $S = \{s_{i,t}\} \in S_c^{T \times N}$, where $X$ is the observation matrix and $S$ is the hidden state matrix.

Table 4: Pre-defined transition patterns $\Psi^{(k)}$.

| Pattern | Description | Transition rule (history $(a, b)$) |
|---|---|---|
| 1. Inertial | Strong tendency to remain if last two states agree | If $a = b$: $[0.9, 0.1]$; else: $[0.5, 0.5]$ |
| 2. Bias-to-2 | History independent, preference for state 2 | Always $[0.1, 0.9]$ |
| 3. Back-to-1 after (1→2) | Return to 1 after (1→2), mild persistence otherwise | If $(a, b) = (1, 2)$: $[0.8, 0.2]$; else if $a = b$: $[0.7, 0.3]$; else: $[0.2, 0.8]$ |
| 4. Flip-on-equality | Flip if last two equal, otherwise keep | If $a = b$: stay 20%, flip 80%; else: stay 85%, flip 15% |
| 5. Random | Completely random transition | Always $[0.5, 0.5]$ |

Table 5: Pre-defined sojourn-time distributions.

| Index | $D_i(1)$ (state 1) | $D_i(2)$ (state 2) |
|---|---|---|
| 1 | Geometric($p = 0.01$) | 1+Poisson($\lambda = 250$) |
| 2 | Geometric($p = 0.001$) | 1+Poisson($\lambda = 20$) |
| 3 | Fixed: 200 | Geometric($p = 0.0025$) |
| 4 | 1+Poisson(100) | 1+Poisson(100) |
| 5 | Geometric($p = 0.01$) | Geometric($p = 0.005$) |
| 6 | Geometric($p = 0.01$) | 1+Poisson($\lambda = 250$) |
| 7 | Geometric($p = 0.001$) | 1+Poisson($\lambda = 20$) |
| 8 | Fixed: 200 | Geometric($p = 0.0025$) |
| 9 | 1+Poisson(100) | 1+Poisson(100) |
| 10 | Geometric($p = 0.01$) | Geometric($p = 0.005$) |

For the first simulated dataset, we set the number of variables to $N = 3$, the number of hidden states to $m = 2$, and the sequence length to $T = 5000$. Then the variables 1, 2, and 3 adopt Transition Patterns 1, 2, and 3 in Table 4, respectively. And the duration distributions of variables 1, 2, and 3 are fixed as Distributions 1, 2, and 3 in Table 5, respectively. Finally, the adjacency matrix $A$ is defined as

$$A = \begin{bmatrix} 0 & 1 & 0 \\ 1 & 0 & 1 \\ 0 & 1 & 0 \end{bmatrix}. \tag{19}$$

An illustration of the generated sequences is provided in Figure 6, where different states are highlighted with distinct background colors.

In order to increase the difficulty of the experiment, for the second simulated dataset, we set $N = 10$, $m = 2$, and $T = 10000$. Each variable independently and randomly selects one transition pattern from Table 4 and one duration distribution from Table 5. And the adjacency matrix $A$ is generated as a random off-diagonal Bernoulli(0.5) matrix. An illustration of the generated sequences is provided in Figure 7.

To evaluate the model under frequent-transition scenarios, we generated two additional fast-switching datasets by shortening the sojourn-time distributions of the first simulated dataset. The parameters for Fast-switching Dataset No. 1 are reported in Table 6, and those for Fast-switching Dataset No. 2 are reported in Table 7. An illustration of the generated sequences is provided in Figures 8 and 9.

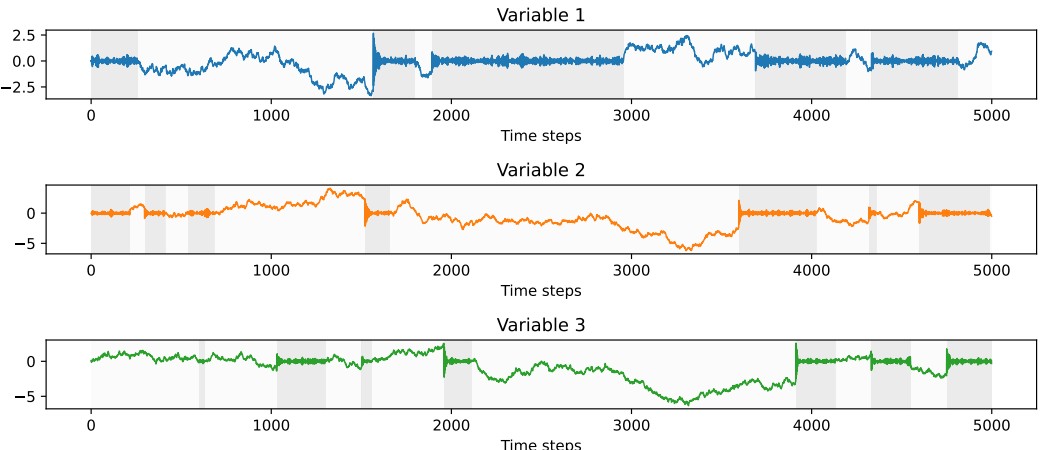

Figure 6: Simulated dataset with 3 variables.

Table 6: Pre-defined sojourn-time distributions for Fast-switching dataset No. 1.

| Index | $D_i(1)$ (state 1) | $D_i(2)$ (state 2) |
|---|---|---|
| 1 | Geometric($p = 0.3$) | 1+Poisson($\lambda = 10$) |
| 2 | Geometric($p = 0.01$) | 1+Poisson($\lambda = 2$) |
| 3 | Fixed: 20 | Geometric($p = 0.1$) |

## C  REAL-WORLD DATASET DESCRIPTIONS

The real-world datasets include a server machine (SMachine) dataset (Su et al., 2019), an exchange rate (Exchange) dataset, and a traffic network (Traffic) dataset (Cao et al., 2020).

The SMachine dataset, which can be accessed from https://github.com/NetManAIOps/OmniAnomaly, contains multivariate time series collected from 28 different machines with 38 dimensions. The training set is unlabeled, while the test set is labeled with anomaly information. Since our study focuses on state-aware multivariate time series forecasting rather than anomaly detection, we select three dimensions from the test set of machine-1 with a length of 7000, which explicitly includes anomalous states. An illustration of the selected data can be found in Figure 10, where different states are highlighted with distinct background colors.

The Exchange dataset is downloaded from https://in.investing.com. It contains daily records of USD/CNY, USD/EUR, and USD/JPY from January 2006 to February 2025, with a total length of 5000 observations. For each exchange rate, the dataset provides five indicators: opening price, highest price, lowest price, closing price, and rate of change. In the experiment, all five indicators are used as inputs, while the closing price is defined as the observation. Since this dataset does not provide explicit state labels, it is only used for evaluating forecasting performance. An illustration of the data can be found in Figure 11.

The Traffic dataset is downloaded from https://github.com/microsoft/StemGNN, which contains multivariate traffic speed records collected from highway sensor stations in California. In our experiment, we select five adjacent nodes that exhibit strong pairwise correlations, and extract a multivariate sequence with a total length of 5000 observations. Since this dataset does not provide explicit state labels, it is only used for evaluating forecasting performance. An illustration of the selected data can be found in Figure 12, where the temporal fluctuations of traffic conditions are clearly visible.

Table 7: Pre-defined sojourn-time distributions for Fast-switching dataset No. 2.

| Index | $D_i(1)$ (state 1) | $D_i(2)$ (state 2) |
|---|---|---|
| 1 | Geometric($p = 0.15$) | 1+Poisson($\lambda = 20$) |
| 2 | Geometric($p = 0.05$) | 1+Poisson($\lambda = 4$) |
| 3 | Fixed: 40 | Geometric($p = 0.05$) |

# D   FULL RESULTS

In this section, we provide supplementary results to further demonstrate the effectiveness of DRL-STAF across different settings and datasets.

## D.1   RESULTS ON SIMULATED DATASETS

Table 8 reports the complete experimental results on simulated dataset with 3 variables, including mean values and standard deviations. Figure 13 reports the results on the simulated dataset with 3 variables. DRL-STAF accurately captures the hidden state transitions and achieves stable forecasting performance under low-dimensional settings.

Table 8: Forecasting and state estimation results on simulated dataset with 3 variables.

| Models | Accuracy | Precision | Recall | F1 | MAE | MSE |
|---|---|---|---|---|---|---|
| Parallel HMM | 74.87% ± 0.80% | 69.44% ± 0.92% | 100.00% ± 0.00% | 81.86% ± 0.64% | 0.5861 ± 0.0071 | 0.5306 ± 0.0150 |
| Parallel HSMM | 60.33% ± 0.94% | 42.40% ± 1.13% | 66.67% ± 1.06% | 51.79% ± 0.92% | 0.6210 ± 0.0086 | 0.6433 ± 0.0155 |
| Parallel HOHMM | 74.13% ± 0.81% | 69.19% ± 0.96% | 98.47% ± 0.29% | 81.18% ± 0.67% | 0.5846 ± 0.0067 | 0.5222 ± 0.0141 |
| CHMM | 74.83% ± 0.79% | 69.42% ± 0.93% | 100.00% ± 0.00% | 81.84% ± 0.65% | 0.5845 ± 0.0072 | 0.5301 ± 0.0158 |
| NHMM | 60.32% ± 0.02% | 49.06% ± 13.33% | 66.65% ± 0.03% | 51.81% ± 0.05% | 0.1277 ± 0.0029 | 0.0390 ± 0.0006 |
| NCTRL | 79.53% ± 3.88% | 88.49% ± 3.90% | 79.15% ± 3.76% | 81.34% ± 2.93% | 0.3936 ± 0.0182 | 0.3042 ± 0.0261 |
| Markovian-RNN | 57.70% ± 1.23% | 44.37% ± 9.33% | 50.28% ± 15.77% | 46.79% ± 8.62% | 0.2444 ± 0.0296 | 0.0935 ± 0.0084 |
| DEN-HMM | 60.33% ± 0.00% | 42.40% ± 0.00% | 66.67% ± 0.00% | 51.79% ± 0.00% | 0.7131 ± 0.0337 | 0.7749 ± 0.0405 |
| DRL-STAF | 98.17% ± 0.07% | 96.89% ± 0.19% | 99.62% ± 0.25% | 98.22% ± 0.07% | 0.0889 ± 0.0003 | 0.0278 ± 0.0003 |

Table 9 reports the complete experimental results on simulated dataset with 10 variables, including mean values and standard deviations. Figure 14 presents the results on the simulated dataset with 10 variables. Compared to the 3-variable case, the increased dimensionality introduces more complex cross-variable dependencies. Nevertheless, DRL-STAF maintains consistent state estimation and forecasting accuracy, showing its robustness in higher-dimensional scenarios.

Table 9: Forecasting and state estimation results on simulated dataset with 10 variables.

| Models | Accuracy | Precision | Recall | F1 | MAE | MSE |
|---|---|---|---|---|---|---|
| Parallel HMM | 70.95% ± 0.30% | 61.68% ± 0.37% | 68.41% ± 0.37% | 61.75% ± 0.29% | 0.5710 ± 0.0070 | 1.0661 ± 0.0248 |
| Parallel HSMM | 77.09% ± 0.29% | 48.28% ± 0.34% | 60.00% ± 0.35% | 52.91% ± 0.27% | 0.6890 ± 0.0080 | 1.5272 ± 0.0310 |
| Parallel HOHMM | 68.39% ± 0.31% | 63.65% ± 0.38% | 57.82% ± 0.42% | 57.24% ± 0.31% | 0.5500 ± 0.0065 | 0.9784 ± 0.0223 |
| CHMM | 75.16% ± 0.35% | 61.64% ± 0.40% | 73.87% ± 0.33% | 64.51% ± 0.30% | 0.6274 ± 0.0077 | 1.2249 ± 0.0267 |
| NHMM | 75.55% ± 2.20% | 51.33% ± 2.40% | 62.00% ± 4.00% | 55.11% ± 3.32% | 0.2621 ± 0.0065 | 0.1600 ± 0.0046 |
| NCTRL | 90.31% ± 1.50% | 88.92% ± 0.69% | 86.71% ± 3.24% | 86.65% ± 1.80% | 0.3431 ± 0.0343 | 0.3935 ± 0.2922 |
| Markovian-RNN | 68.71% ± 1.27% | 61.18% ± 9.32% | 66.36% ± 15.71% | 60.93% ± 8.59% | 0.1211 ± 0.0298 | 0.0355 ± 0.0076 |
| DEN-HMM | 75.89% ± 0.00% | 53.03% ± 0.00% | 69.30% ± 0.00% | 58.72% ± 0.00% | 0.7247 ± 0.0033 | 1.6617 ± 0.0002 |
| DRL-STAF | 96.15% ± 0.15% | 95.44% ± 0.52% | 91.89% ± 0.60% | 93.26% ± 0.21% | 0.1090 ± 0.0005 | 0.0395 ± 0.0009 |

Furthermore, Tables 10 and 11 report the results on 3-variable simulated dataset with frequent transitions, where dataset No. 1 features a higher transition rate than dataset No. 2. On both fast-switching datasets, DRL-STAF continues to achieve strong forecasting accuracy and reliable state estimation, whereas most likelihood-driven methods degrade substantially under rapid regime changes. These results demonstrate that although certain components of DRL-STAF are motivated by the infrequent-transition assumption, they do not enforce strict state persistence in practice. Instead, they act as robust conservative mechanisms that prevent spurious rapid switching under uncertainty and yield stable, reliable state estimates even when the true hidden state switch frequently. Figures 15 and 16 report the results on the 3-variable simulated datasets with frequent transitions. DRL-STAF accurately captures the hidden state transitions and achieves stable forecasting performance.

Table 10: Forecasting and state estimation results on simulated dataset with 3 variables (Fast-switching No. 1).

| Models | Accuracy | Precision | Recall | F1 | MAE | MSE |
|---|---|---|---|---|---|---|
| Parallel HMM | 63.37% ± 0.92% | 45.33% ± 1.27% | 55.95% ± 1.26% | 49.75% ± 1.06% | 0.2564 ± 0.0042 | 0.1183 ± 0.0043 |
| Parallel HSMM | 63.80% ± 0.85% | 23.57% ± 1.37% | 33.33% ± 1.22% | 27.61% ± 1.13% | 0.2771 ± 0.0043 | 0.1393 ± 0.0047 |
| Parallel HOHMM | 57.87% ± 0.93% | 48.23% ± 1.35% | 47.69% ± 1.39% | 46.37% ± 1.19% | 0.2133 ± 0.0036 | 0.0847 ± 0.0034 |
| CHMM | 62.23% ± 0.93% | 58.07% ± 1.11% | 87.93% ± 0.85% | 68.17% ± 0.93% | 0.2724 ± 0.0048 | 0.1474 ± 0.0058 |
| NHMM | 62.40% ± 2.54% | 45.49% ± 21.11% | 43.42% ± 8.24% | 36.85% ± 7.68% | 0.2271 ± 0.0023 | 0.1009 ± 0.0012 |
| NCTRL | 80.93% ± 5.77% | 83.23% ± 5.94% | 78.59% ± 10.06% | 79.37% ± 6.21% | 0.2439 ± 0.0044 | 0.1137 ± 0.0021 |
| Markovian-RNN | 61.30% ± 2.20% | 51.19% ± 3.06% | 65.81% ± 5.40% | 57.24% ± 4.03% | 0.1338 ± 0.0032 | 0.0430 ± 0.0017 |
| DEN-HMM | 61.47% ± 2.33% | 29.56% ± 5.54% | 41.18% ± 9.31% | 34.40% ± 7.00% | 0.2861 ± 0.0049 | 0.1480 ± 0.0051 |
| DRL-STAF | **89.77% ± 0.11%** | **88.47% ± 1.42%** | **89.04% ± 2.28%** | **88.66% ± 0.45%** | **0.1070 ± 0.0002** | **0.0367 ± 0.0001** |

Table 11: Forecasting and state estimation results on simulated dataset with 3 variables (Fast-switching No. 2).

| Models | Accuracy | Precision | Recall | F1 | MAE | MSE |
|---|---|---|---|---|---|---|
| Parallel HMM | 62.67% ± 0.85% | 27.00% ± 1.41% | 30.62% ± 1.43% | 28.66% ± 1.24% | 0.4481 ± 0.0076 | 0.4844 ± 0.0176 |
| Parallel HSMM | 71.40% ± 0.84% | 24.73% ± 1.41% | 33.33% ± 1.37% | 28.40% ± 1.19% | 0.5744 ± 0.0086 | 0.8235 ± 0.0238 |
| Parallel HOHMM | 58.90% ± 0.91% | 60.92% ± 1.58% | 32.83% ± 1.32% | 33.90% ± 1.25% | 0.4274 ± 0.0071 | 0.4505 ± 0.0162 |
| CHMM | 62.67% ± 0.91% | 27.00% ± 1.52% | 30.62% ± 1.36% | 28.66% ± 1.25% | 0.4475 ± 0.0076 | 0.4839 ± 0.0177 |
| NHMM | 68.50% ± 4.50% | 38.50% ± 13.99% | 43.40% ± 8.24% | 36.33% ± 7.17% | 0.2497 ± 0.0070 | 0.1840 ± 0.0020 |
| NCTRL | 80.11% ± 8.48% | 70.47% ± 14.53% | 70.97% ± 15.89% | 68.43% ± 14.08% | 0.3064 ± 0.0216 | 0.2391 ± 0.0276 |
| Markovian-RNN | 63.98% ± 2.76% | 48.60% ± 4.52% | 52.32% ± 6.67% | 47.29% ± 4.73% | 0.1697 ± 0.0089 | 0.0998 ± 0.0032 |
| DEN-HMM | 66.32% ± 5.08% | 25.87% ± 1.13% | 36.13% ± 2.80% | 30.01% ± 1.61% | 0.5798 ± 0.0010 | 0.8264 ± 0.0048 |
| DRL-STAF | **93.24% ± 0.21%** | **90.20% ± 2.26%** | **93.22% ± 2.08%** | **91.52% ± 0.23%** | **0.1012 ± 0.0001** | **0.0418 ± 0.0001** |

## D.2 RESULTS ON REAL-WORLD DATASETS

Table 12 reports the complete experimental results on SMachine dataset, including mean values and standard deviations. Figure 17 illustrates the results on the SMachine dataset, where ground-truth anomaly labels are available. The model is able to identify state changes that correspond closely to anomalous behaviors, while also providing accurate forecasts of future observations.

Table 12: Forecasting and state estimation results on SMachine dataset.

| Models | Accuracy | Precision | Recall | F1 | MAE | MSE |
|---|---|---|---|---|---|---|
| Parallel HMM | 76.37% | 55.86% | **90.95%** | 69.21% | 0.0563 | 0.0056 |
| Parallel HSMM | 77.16% | 61.50% | 58.19% | 59.80% | 0.0968 | 0.0157 |
| Parallel HOHMM | 76.37% | 55.86% | **90.95%** | 69.21% | 0.0563 | 0.0056 |
| CHMM | 76.02% | 55.52% | 89.73% | 68.60% | 0.0563 | 0.0056 |
| NHMM | 70.77% ± 0.03% | 0.00% ± 0.00% | 0.00% ± 0.00% | 0.00% ± 0.00% | 0.0194 ± 0.0002 | **0.0010 ± 0.0000** |
| NCTRL | 72.54% ± 5.67% | 54.04% ± 13.66% | 41.66% ± 24.45% | 42.33% ± 20.71% | 0.0360 ± 0.0012 | 0.0023 ± 0.0001 |
| Markovian-RNN | 70.81% ± 1.67% | 0.00% ± 4.31% | 0.00% ± 2.53% | 0.00% ± 3.08% | 0.0190 ± 0.0005 | **0.0010 ± 0.0000** |
| DEN-HMM | 68.59% ± 0.01% | 37.40% ± 0.00% | 11.25% ± 0.00% | 17.29% ± 0.00% | 0.1537 ± 0.0047 | 0.0386 ± 0.0004 |
| DRL-STAF | **81.73% ± 0.10%** | **100.00% ± 0.32%** | 63.27% ± 0.18% | **77.50% ± 0.15%** | **0.0189 ± 0.0000** | **0.0010 ± 0.0000** |

Table 13 reports the complete experimental results on Exchange dataset and Traffic dataset, including mean values and standard deviations. Figures 18 and 19 show the results on the Exchange dataset and Traffic dataset, both of which do not provide ground-truth state labels. Here, the background colors indicate the hidden states estimated by DRL-STAF. The segmentation of states provides additional interpretability, as it highlights structural shifts in the time series, while the forecasting performance remains competitive compared to other baselines.

## D.3 EXTENDED ABLATION EXPERIMENTS

To further demonstrate the necessity of each component in the DRL-STAF framework, we compare the full model with several ablated variants: DRL-NASP (without the action switching penalty), DRL-NSSS (without the sample screening strategy), DRL-NASP&SSS (without both the action switching penalty and sample screening), DRL-NBL (without the baseline error term $e^{base}$), DRL-NSSE (without the state separation evaluation term), DRL-NPDE (without the pairwise discrepancy evaluation term), DRL-NER (without the episodic reward), and DRL-NSO (removing stage one

Table 13: Forecasting results on Exchange dataset and Traffic dataset.

| Models | Exchange dataset | | Traffic dataset | |
|---|---|---|---|---|
| | MAE | MSE | MAE | MSE |
| Parallel HMM | 8.6701 | 294.0451 | 5.5411 | 59.5501 |
| Parallel HSMM | 9.9193 | 366.7165 | 14.1709 | 259.8108 |
| Parallel HOHMM | 8.5905 | 294.0307 | 5.5419 | 59.5004 |
| CHMM | 8.5851 | 293.5766 | 5.1245 | 50.2464 |
| Markovian-RNN | $2.5361 \pm 0.3795$ | $28.1406 \pm 8.1495$ | $2.0032 \pm 0.0004$ | $9.9143 \pm 0.3128$ |
| DEN-HMM | $11.5619 \pm 0.1745$ | $470.4355 \pm 7.4616$ | $13.8156 \pm 0.2225$ | $245.5208 \pm 5.6578$ |
| NHMM | $4.4926 \pm 1.1645$ | $100.0099 \pm 48.8592$ | $1.5881 \pm 0.1090$ | $6.7800 \pm 0.6254$ |
| NCTRL | $1.8486 \pm 0.2596$ | $13.6292 \pm 4.1032$ | $2.5353 \pm 0.0380$ | $12.8844 \pm 0.0954$ |
| DRL-STAF | $\mathbf{1.6438 \pm 0.0029}$ | $\mathbf{13.2381 \pm 0.0464}$ | $\mathbf{1.5193 \pm 0.0023}$ | $\mathbf{6.4610 \pm 0.0156}$ |

and directly modeling cross-variable interactions). The ablation results in Table 14 highlight the necessity of each component in DRL-STAF. Removing the sample screening strategy (DRL-NSSS), the episodic reward (DRL-NER), or the training of Stage one (DRL-NSO) produces the most severe degradation, causing the model to lose reliable state estimation and forecasting accuracy. In contrast, removing other components, such as the action switching penalty (DRL-NASP), baseline error term (DRL-NBL), state separation evaluation (DRL-NSSE), or pairwise discrepancy evaluation (DRL-NPDE), primarily affects stability and boundary sharpness, resulting in reduced but still operational performance.

Table 14: Comparison of ablation results on simulated dataset with 3 variables.

| Models | Accuracy | Precision | Recall | F1 | MAE | MSE |
|---|---|---|---|---|---|---|
| DRL-STAF | $\mathbf{98.17\% \pm 0.07\%}$ | $\mathbf{96.89\% \pm 0.19\%}$ | $\mathbf{99.62\% \pm 0.25\%}$ | $\mathbf{98.22\% \pm 0.07\%}$ | $\mathbf{0.0889 \pm 0.0003}$ | $\mathbf{0.0278 \pm 0.0003}$ |
| DRL-NASP | $96.21\% \pm 0.22\%$ | $89.71\% \pm 0.34\%$ | $98.42\% \pm 0.26\%$ | $93.69\% \pm 0.30\%$ | $0.1286 \pm 0.0004$ | $0.0499 \pm 0.0022$ |
| DRL-NSSS | $68.99\% \pm 0.19\%$ | $50.38\% \pm 0.24\%$ | $66.50\% \pm 0.05\%$ | $56.71\% \pm 0.13\%$ | $0.2307 \pm 0.0008$ | $0.1572 \pm 0.0005$ |
| DRL-NASP&SSS | $58.69\% \pm 0.91\%$ | $62.79\% \pm 1.32\%$ | $58.64\% \pm 6.69\%$ | $59.78\% \pm 3.24\%$ | $0.2574 \pm 0.0004$ | $0.1800 \pm 0.0004$ |
| DRL-NBL | $95.42\% \pm 1.05\%$ | $89.75\% \pm 0.57\%$ | $95.38\% \pm 4.22\%$ | $92.33\% \pm 1.90\%$ | $0.0907 \pm 0.0012$ | $0.0287 \pm 0.0006$ |
| DRL-NSSE | $93.27\% \pm 0.36\%$ | $88.58\% \pm 0.77\%$ | $93.15\% \pm 1.60\%$ | $90.43\% \pm 0.42\%$ | $0.0965 \pm 0.0015$ | $0.0426 \pm 0.0012$ |
| DRL-NPDE | $95.46\% \pm 1.59\%$ | $94.05\% \pm 2.44\%$ | $90.06\% \pm 7.30\%$ | $91.65\% \pm 3.31\%$ | $0.0932 \pm 0.0032$ | $0.0294 \pm 0.0013$ |
| DRL-NER | $67.57\% \pm 0.00\%$ | $19.00\% \pm 0.00\%$ | $33.33\% \pm 0.00\%$ | $24.20\% \pm 0.00\%$ | $0.2099 \pm 0.0000$ | $0.1590 \pm 0.0000$ |
| DRL-NSO | $66.48\% \pm 1.53\%$ | $24.93\% \pm 8.39\%$ | $33.64\% \pm 0.43\%$ | $26.60\% \pm 3.39\%$ | $0.5015 \pm 0.0687$ | $0.9767 \pm 0.0964$ |

In addition, to verify the necessity of reinforcement learning itself, we include head-to-head comparisons against four non-RL alternatives: MoE-SOFT (soft decoding + MSE minimization), MoE-HARD (Gumbel–Softmax hard decoding + MSE minimization), MoE-ISOFT (soft decoding + DRL-aligned loss), and MoE-IHARD (Gumbel–Softmax hard decoding + DRL-aligned loss). Table 15 reports the complete experimental results on simulated dataset with 3 variables. The experimental results show that all four non-RL alternatives fail to produce meaningful state estimates and cannot achieve accurate forecasting. These results confirm that RL is essential for coupling forecasting accuracy with reliable discrete-state inference in DRL-STAF.

Table 15: Comparisons against four non-RL alternatives.

| Models | Accuracy | Precision | Recall | F1 | MAE | MSE |
|---|---|---|---|---|---|---|
| DRL-STAF | $\mathbf{98.17\% \pm 0.07\%}$ | $\mathbf{96.89\% \pm 0.19\%}$ | $\mathbf{99.62\% \pm 0.25\%}$ | $\mathbf{98.22\% \pm 0.07\%}$ | $\mathbf{0.0889 \pm 0.0003}$ | $\mathbf{0.0278 \pm 0.0003}$ |
| MoE-SOFT | $62.53\% \pm 2.71\%$ | $47.36\% \pm 7.76\%$ | $72.98\% \pm 13.27\%$ | $57.36\% \pm 9.73\%$ | $0.2687 \pm 0.0090$ | $0.1799 \pm 0.0032$ |
| MoE-ISOFT | $52.83\% \pm 1.35\%$ | $58.85\% \pm 1.45\%$ | $57.53\% \pm 6.61\%$ | $57.54\% \pm 3.26\%$ | $0.2596 \pm 0.0005$ | $0.1810 \pm 0.0006$ |
| MoE-HARD | $60.41\% \pm 0.16\%$ | $42.47\% \pm 0.15\%$ | $66.61\% \pm 0.12\%$ | $51.83\% \pm 0.07\%$ | $0.2619 \pm 0.0034$ | $0.1818 \pm 0.0019$ |
| MoE-IHARD | $51.37\% \pm 0.55\%$ | $59.16\% \pm 0.48\%$ | $51.33\% \pm 0.59\%$ | $54.63\% \pm 0.50\%$ | $0.2597 \pm 0.0002$ | $0.1809 \pm 0.0007$ |

# E   MODEL COMPLEXITY ANALYSIS

Consider $N$ variables, each associated with $m$ hidden states. In conventional multivariate HMMs, each hidden state adopts a Gaussian mixture emission model with $C$ components. For higher-order

HMMs, we denote the Markov order by $R$. For HSMMs, we use $D$ to denote the maximum duration truncation. For DL methods, we use $h$ to denote the hidden dimension of neural networks. The sequence length is written as $T$. Specially, DRL-STAF is trained on fixed-length episodes of size $L$, making its per-iteration complexity independent of $T$.

Table 16: Comparison of number of parameters for the methods we considered. "Yes" means the model explicitly encodes dependencies between different variables. "No" means variables are modeled by independent chains.

| Models | Cross-variable interactions | Parameter Complexity | Computational Complexity (per sequence of length $T$) |
|---|---|---|---|
| Parallel HMM | No | $O(N(m^2 + mC))$ | $O(NT(m^2 + mC))$ |
| Parallel HSMM | No | $O(N(m^2 + mC))$ | $O(NTm^2D)$ |
| Parallel HOHMM | No | $O(Nm^{R+1})$ | $O(NTm^{2R})$ |
| CHMM | Yes | $O(m^{2N})$ | $O(Tm^{2N})$ |
| NHMM | No | $O(N(h^2 + hm^2))$ | $O(NT(h^2 + hm^2))$ |
| NCTRL | No | $O(N(h^3 + m^2))$ | $O(NT(h^3 + m^2))$ |
| Markovian-RNN | No | $O(N(mh^2 + m^2))$ | $O(NT(mh^2 + m^2))$ |
| DEN-HMM | No | $O(N(h^2 + m^2))$ | $O(NT(h^2 + m^2))$ |
| DRL-STAF | Yes | $O(N(mh^2))$ | $O(NL(mh^2))$ |

All parameter counts and complexities reported in Table 16 are expressed using these quantities. The advantage becomes clear when comparing parameter complexities between CHMM and DRL-STAF. CHMM requires modeling the joint latent space with $O(m^{2N})$ parameters, which grows exponentially with the number of variables. In contrast, DRL-STAF only has $O(N(mh^2))$ parameters, growing linearly in $N$. Since DRL-STAF never enumerates joint state combinations, it avoids the combinatorial explosion inherent to CHMMs while still capturing cross-variable dependencies.

To make the computational cost concrete, we further measured wall-clock performance on a machine with an Intel(R) Core(TM) Ultra 5 125H CPU. Table 17 reports training time and per-step inference latency for all single-variable models, and for multivariate settings Table 18 additionally reports results under different values of $N$.

Table 17: Computational cost of single-variable models (time in seconds)

| Single-variable settings | Training | Per-step inference |
|---|---|---|
| Parallel HMM | 13.805 | 0.002 |
| Parallel HSMM | 595.557 | 14.200 |
| Parallel HOHMM | 33.753 | 0.005 |
| NHMM | 214.061 | 0.003 |
| NCTRL | 738.380 | 0.055 |
| Markovian-RNN | 2972.960 | 0.008 |
| DEN-HMM | 6465.399 | 0.001 |
| DRL-STAF (Stage one) | 5443.833 | 0.001 |

Table 18: Computational cost of multivariate models (time in seconds)

| Multivariate settings | 3 variables | | 5 variables | | 10 variables | |
|---|---|---|---|---|---|---|
| | Training | Per-step inference | Training | Per-step inference | Training | Per-step inference |
| CHMM | 77.760 | 0.014 | 158.866 | 0.030 | 156588.235 | 2.513 |
| DRL-STAF | 47468.344 | 0.006 | 74792.616 | 0.011 | 119733.858 | 0.018 |

# F  DEEP REINFORCEMENT LEARNING ALGORITHM

In DRL-STAF, the hidden-state policy is trained using a standard clipped Proximal Policy Optimization (PPO) actor-critic algorithm. The policy network $\pi_\theta(a_t \mid s_t)$ maps the current observation window and historical summaries to a categorical distribution over discrete states, while the value network $V_\phi(s_t)$ provides a scalar baseline.

During training, actions are sampled from the categorical distribution; at evaluation time, hard decoding is performed using an $\arg\max$ operation. Gradients do not flow through discrete sampling.

**Value baseline and advantage estimation.**

Given a collected trajectory $\{(s_t, a_t, r_t, s_{t+1}, \text{done}_t)\}$, we compute bootstrapped TD targets

$$\hat{V}_t^{\text{target}} = r_t + \gamma(1 - \text{done}_t)\, V_\phi(s_{t+1}), \tag{20}$$

and TD errors

$$\delta_t = \hat{V}_t^{\text{target}} - V_\phi(s_t). \tag{21}$$

We use generalized advantage estimation (GAE) to obtain low-variance advantage estimates:

$$\hat{A}_t = \text{GAE}_{\gamma,\lambda}(\delta_t). \tag{22}$$

The learned value function therefore acts as a variance-reducing baseline.

**PPO policy update.**

We store old log-probabilities $\log \pi_{\theta_{\text{old}}}(a_t \mid s_t)$, and in each PPO epoch we recompute $\log \pi_\theta(a_t \mid s_t)$ and the importance ratio

$$r_t(\theta) = \exp\left[\log \pi_\theta(a_t \mid s_t) - \log \pi_{\theta_{\text{old}}}(a_t \mid s_t)\right]. \tag{23}$$

The clipped PPO objective is

$$L_{\text{actor}}(\theta) = \mathbb{E}_t\left[\min\left(r_t(\theta)\hat{A}_t,\ \text{clip}(r_t(\theta), 1 - \varepsilon, 1 + \varepsilon)\hat{A}_t\right)\right] + \beta H(\pi_\theta(\cdot \mid s_t)), \tag{24}$$

where $H(\cdot)$ is the categorical entropy and $\beta = 0.04$ is the entropy coefficient used to encourage exploration. The value function is trained by minimizing the TD error:

$$L_{\text{critic}}(\phi) = \mathbb{E}_t\left[(V_\phi(s_t) - \hat{V}_t^{\text{target}})^2\right]. \tag{25}$$

Actor and critic parameters are optimized with Adam.

**Training stability.**

Stability is maintained through:

- The learned value-function baseline;
- GAE$(\gamma, \lambda)$ for low-variance advantage estimation;
- PPO clipping of importance ratios;
- An explicit entropy bonus.

The prediction modules are not updated through policy gradients; they are trained via supervised forecasting losses, and the policy receives only their prediction errors.

# G   PARAMETER SETTING

For reproducibility, we summarize the hyperparameter settings of all baseline models evaluated in our experiments. For Parallel HMM, Parallel HSMM, Parallel HOHMM, and CHMM, all parameters strictly follow the default configurations provided in our released implementation. For Markovian-RNN, The network architecture and training hyperparameters follow the original paper, including hidden size, recurrent structure, and optimization settings. We further performed parameter tuning within the ranges recommended by the authors to ensure fair comparison. For DEN-HMM, the emission network architecture (DEN) and all hyperparameters follow the original publication exactly. For NHMM, the model architecture mirrors the structure of the prediction module in DRL-STAF. In particular, we use the similar deep emission network design for comparison. For NCTRL, the model architecture and all training parameters are implemented following the original paper. For DRL-STAF, Table 19 provides the most key hyperparameters across different datasets.

Table 19: Key hyperparameters of DRL-STAF across different datasets.

| Hyperparameter | Simulated datasets | SMachine | Exchange |
|---|---|---|---|
| Discount factor $\gamma$ | 0.99 | 0.99 | 0.99 |
| Prediction gain weight $\lambda_1$ | 4 | 4 | 4 |
| Action switching penalty weight $\lambda_2$ | Stage-1 0.015; Stage-1 0.02 | Stage-1 0.015; Stage-1 0.02 | Stage-1 0.015; Stage-1 0.015 |
| State separation asymmetry $\lambda_3$ | 2 | 2 | 2 |
| Pairwise discrepancy weight $\lambda_4$ | 2 | 2 | 2 |
| Temporal balance $\alpha$ | 0.5 | 0.5 | 0.5 |
| Continuity threshold $\rho_c$ | 8 | 8 | 8 |
| High continuity threshold $\phi_H$ | 8 | 8 | 8 |
| Low continuity threshold $\phi_L$ | 2 | 2 | 2 |
| Soft update rate $\tau$ | 0.01 | 0.01 | 0.01 |
| History length $T_1$ | 1 | 2 | 4 |
| History length $T_2$ | 1 | 2 | 4 |
| Episode length | 2000 | 2000 | 2000 |

## H USE OF LLMs

In preparing our manuscript, we used large language models (LLMs) only to aid or polish the writing. Their use was restricted to improving grammar, readability, and style, without contributing to the methodology, theoretical results, algorithmic implementation, or experimental outcomes. The involvement of LLMs does not affect the reproducibility of our findings.

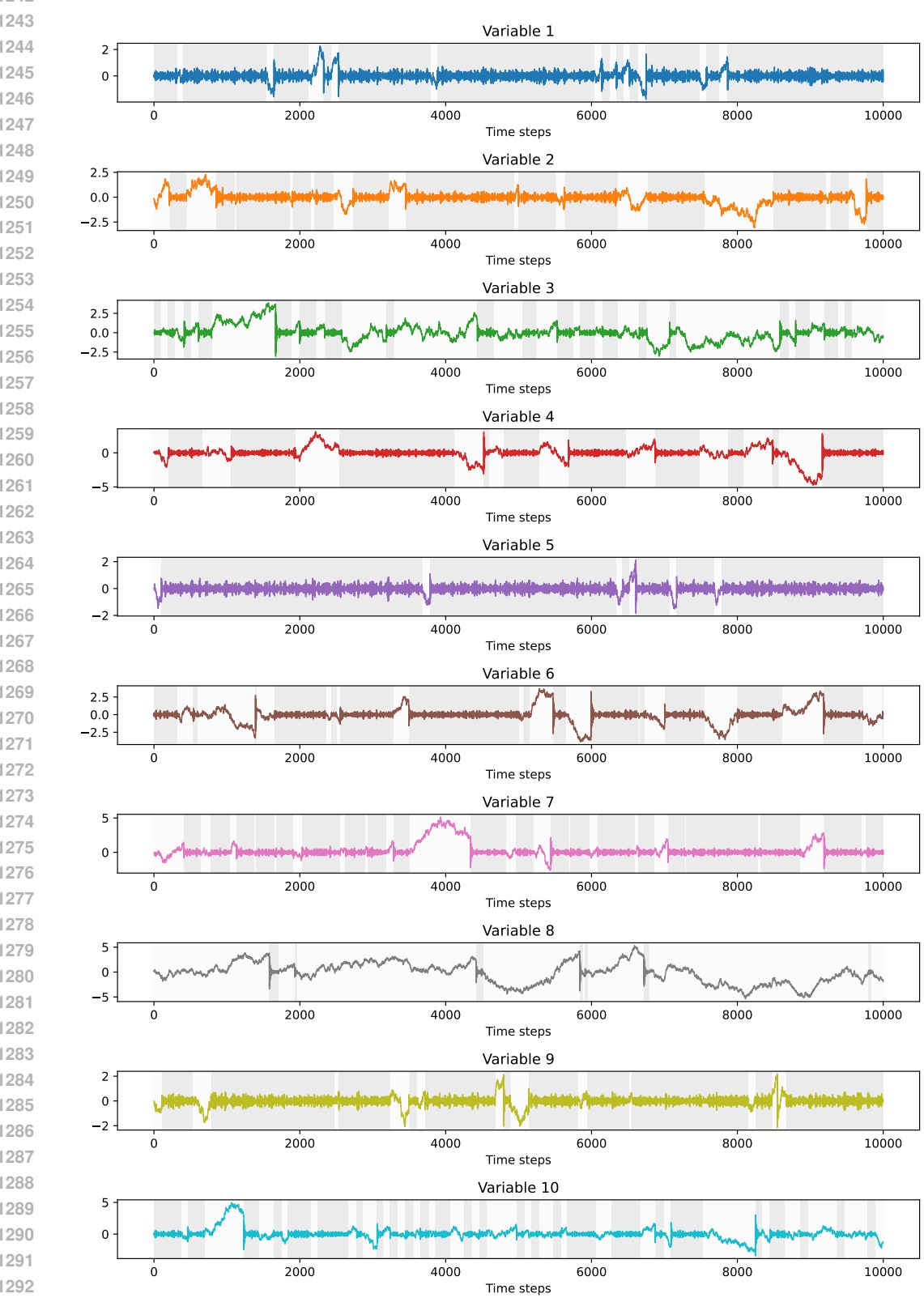

Figure 7: Simulated dataset with 10 variables.

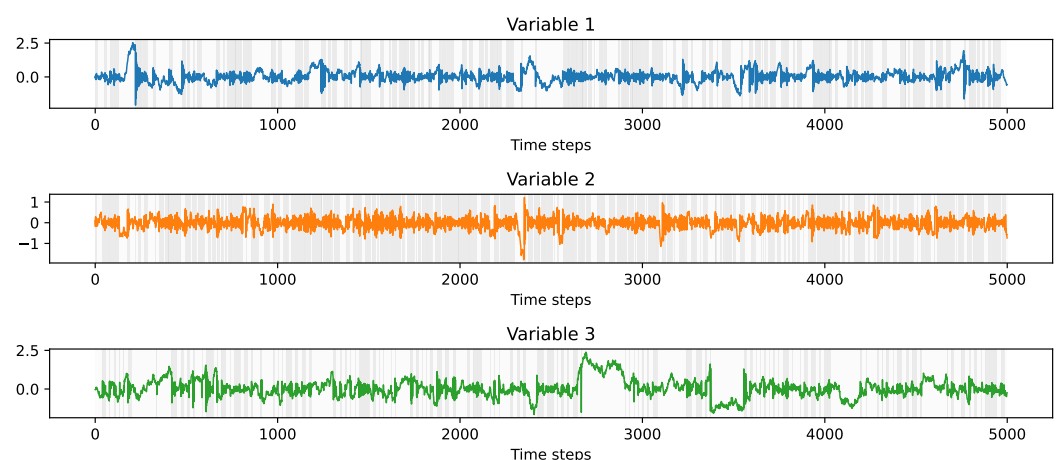

Figure 8: Simulated dataset with 3 variables (Fast-switching No. 1).

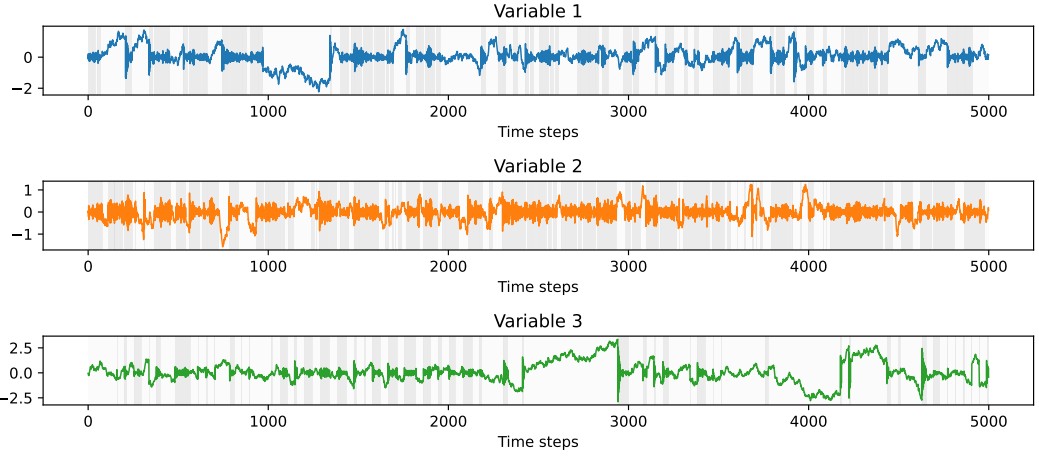

Figure 9: Simulated dataset with 3 variables (Fast-switching No. 2).

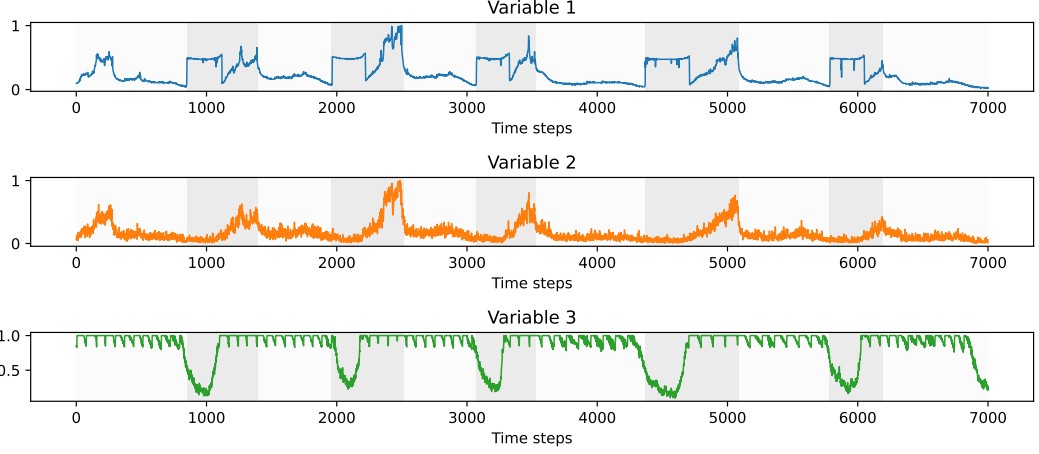

Figure 10: The server machine dataset.

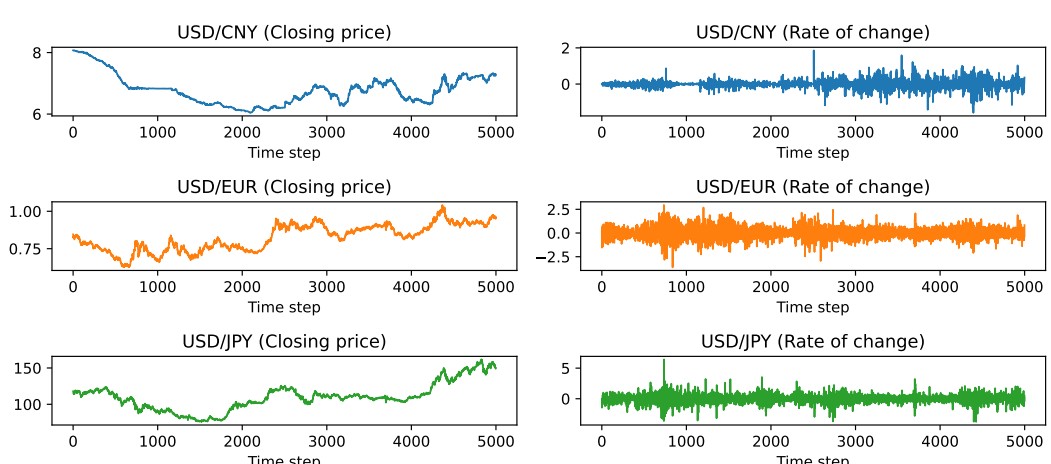

Figure 11: The exchange rate dataset.

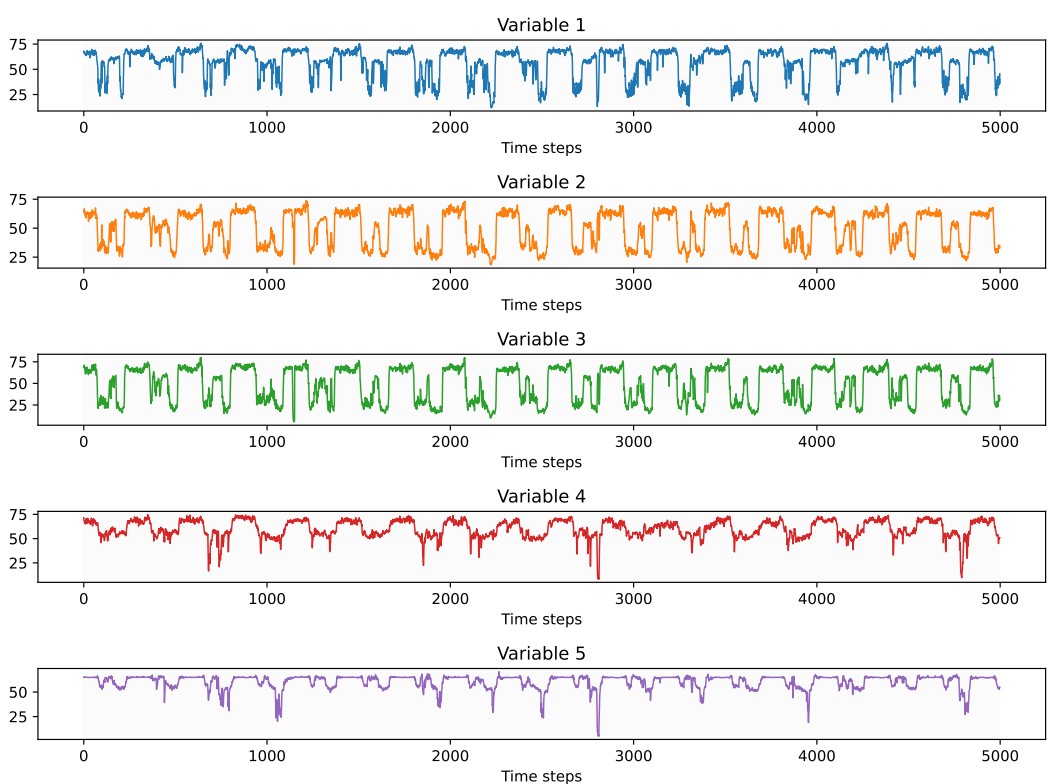

Figure 12: The traffic network dataset.

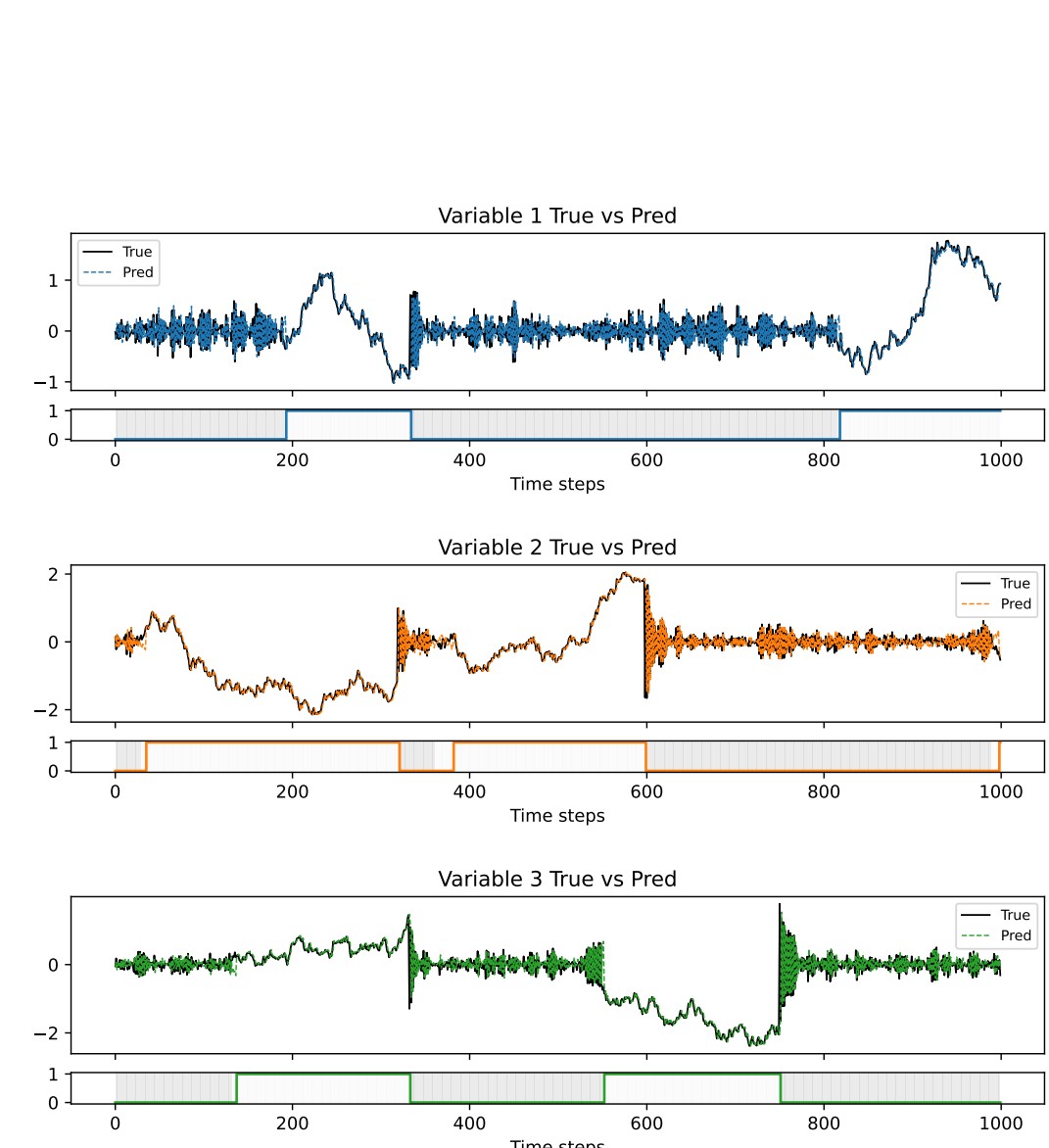

Figure 13: Results of DRL-STAF on the simulated dataset with 3 variables.

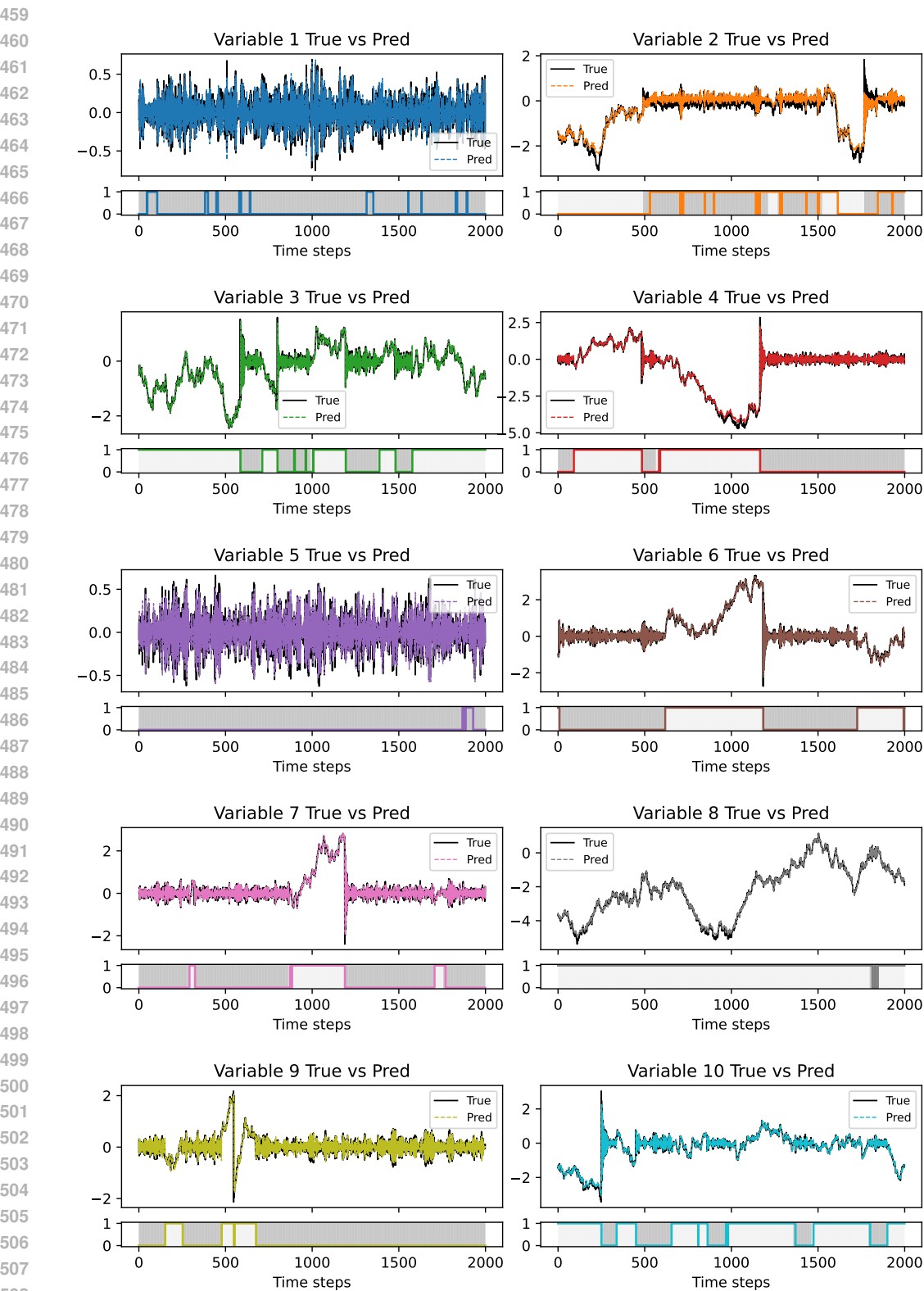

Figure 14: Results of DRL-STAF on the simulated dataset with 10 variables.

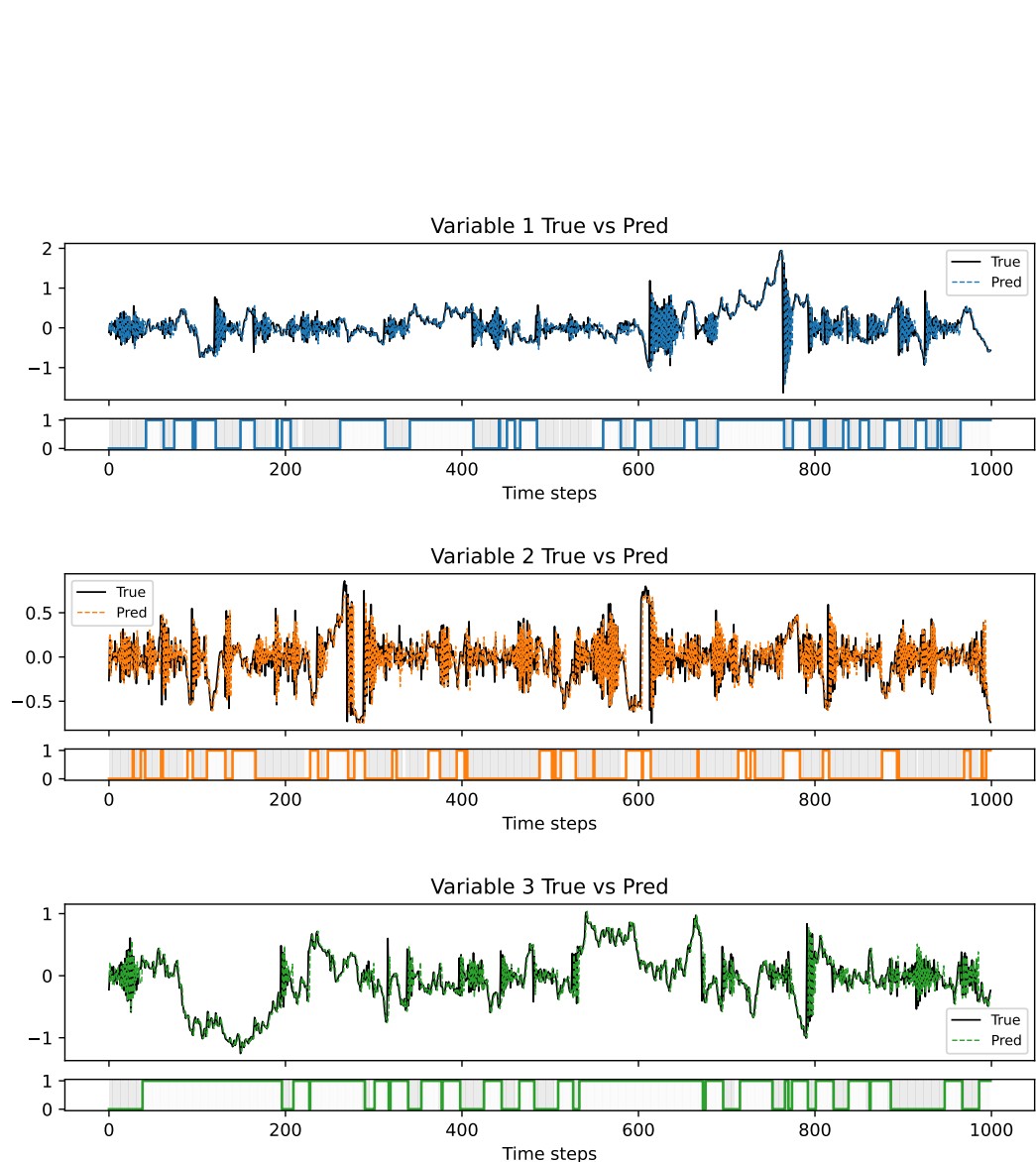

Figure 15: Results of DRL-STAF on the simulated dataset with 3 variables (Fast-switching No. 1).

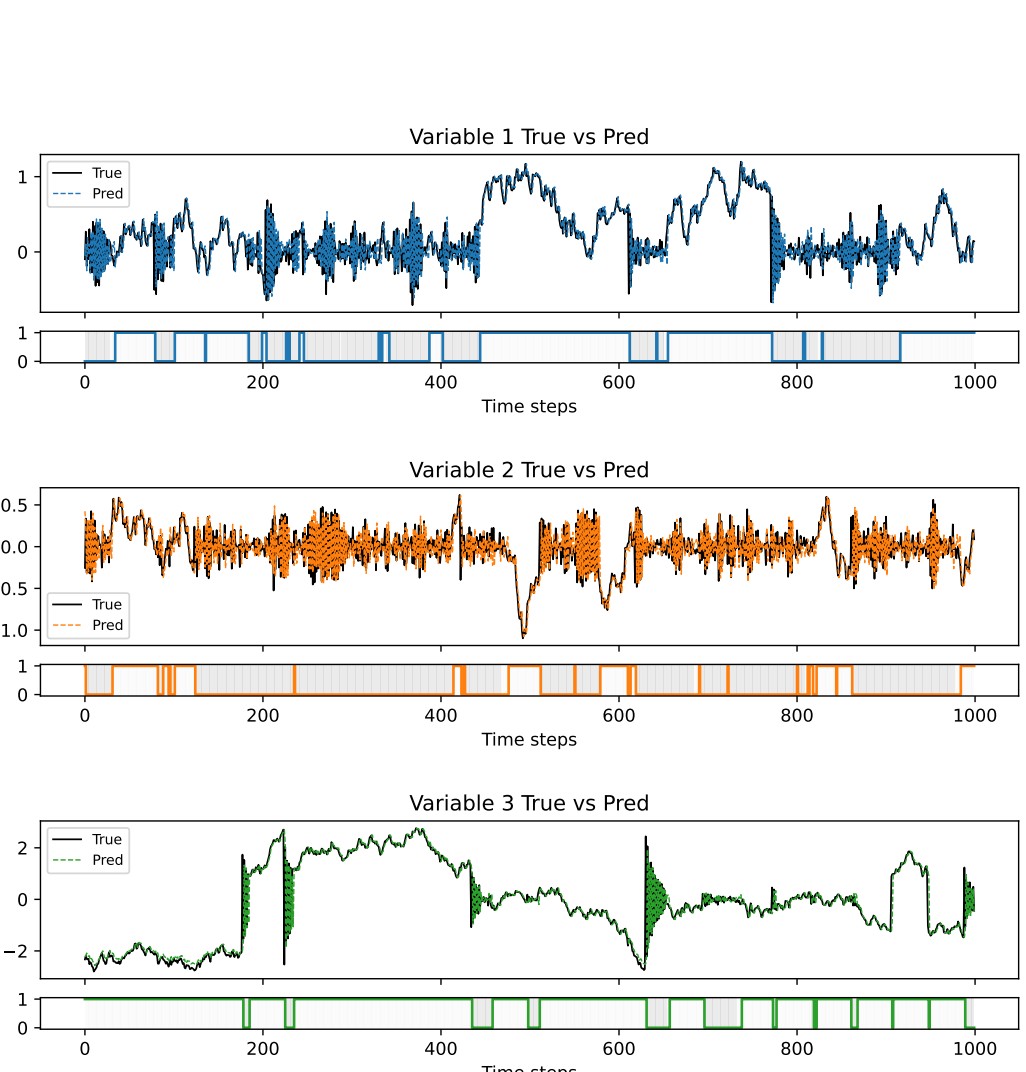

Figure 16: Results of DRL-STAF on the simulated dataset with 3 variables (Fast-switching No. 2).

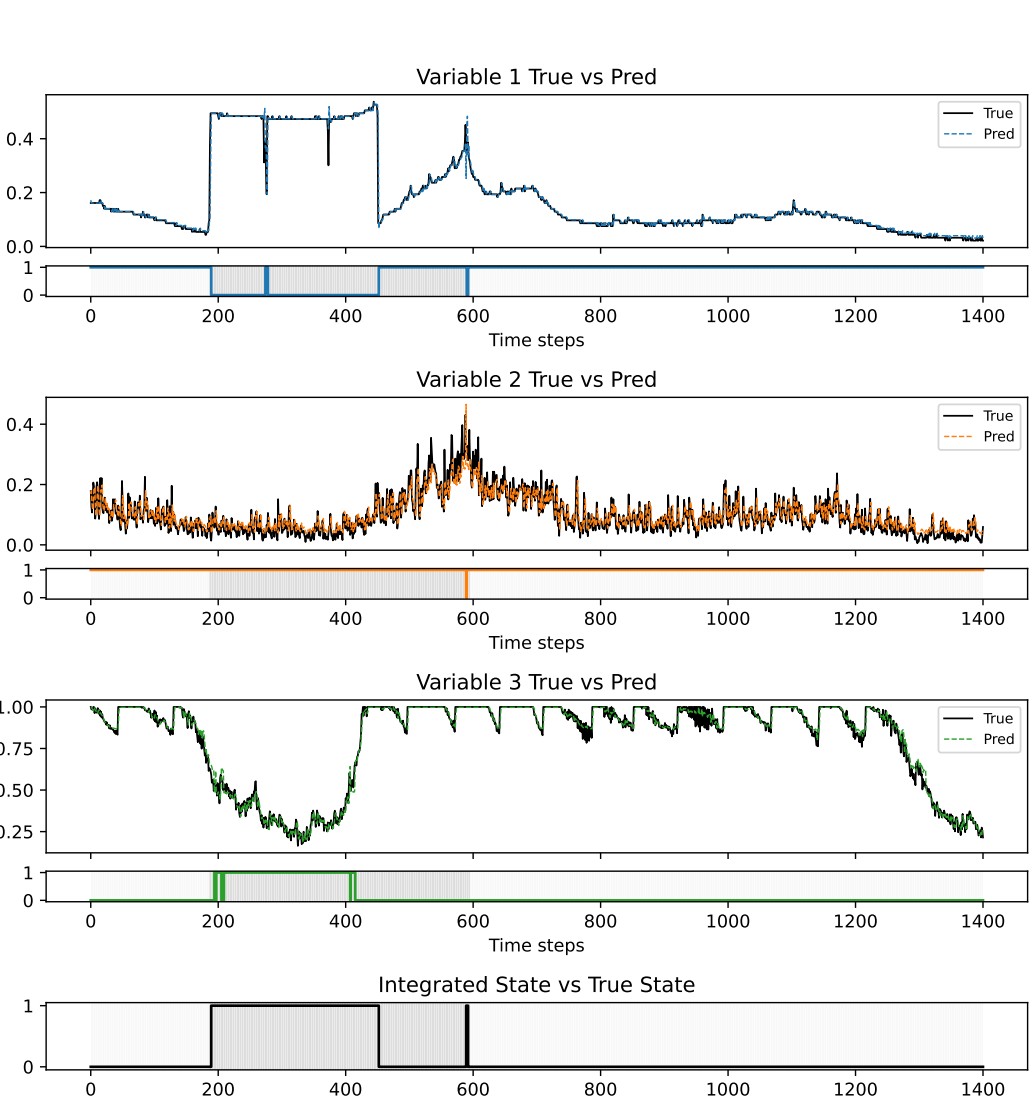

Figure 17: Results of DRL-STAF on the SMachine dataset. The integrated state represents the model's overall judgment of the hidden state, where the state is regarded as anomalous if any single variable is detected as anomalous.

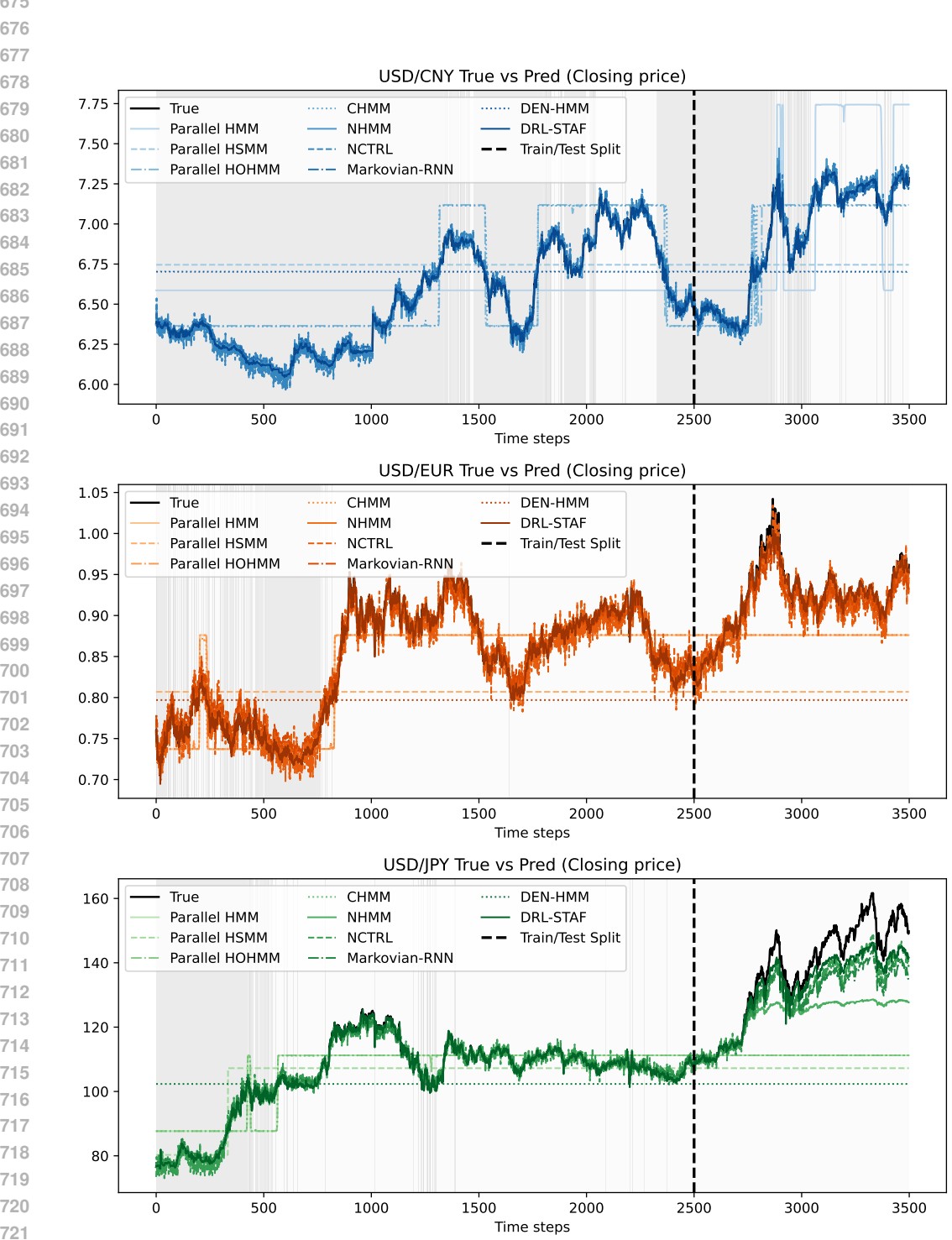

Figure 18: Results of different methods on the Exchange dataset. Background colors indicate different hidden states estimated by DRL-STAF.

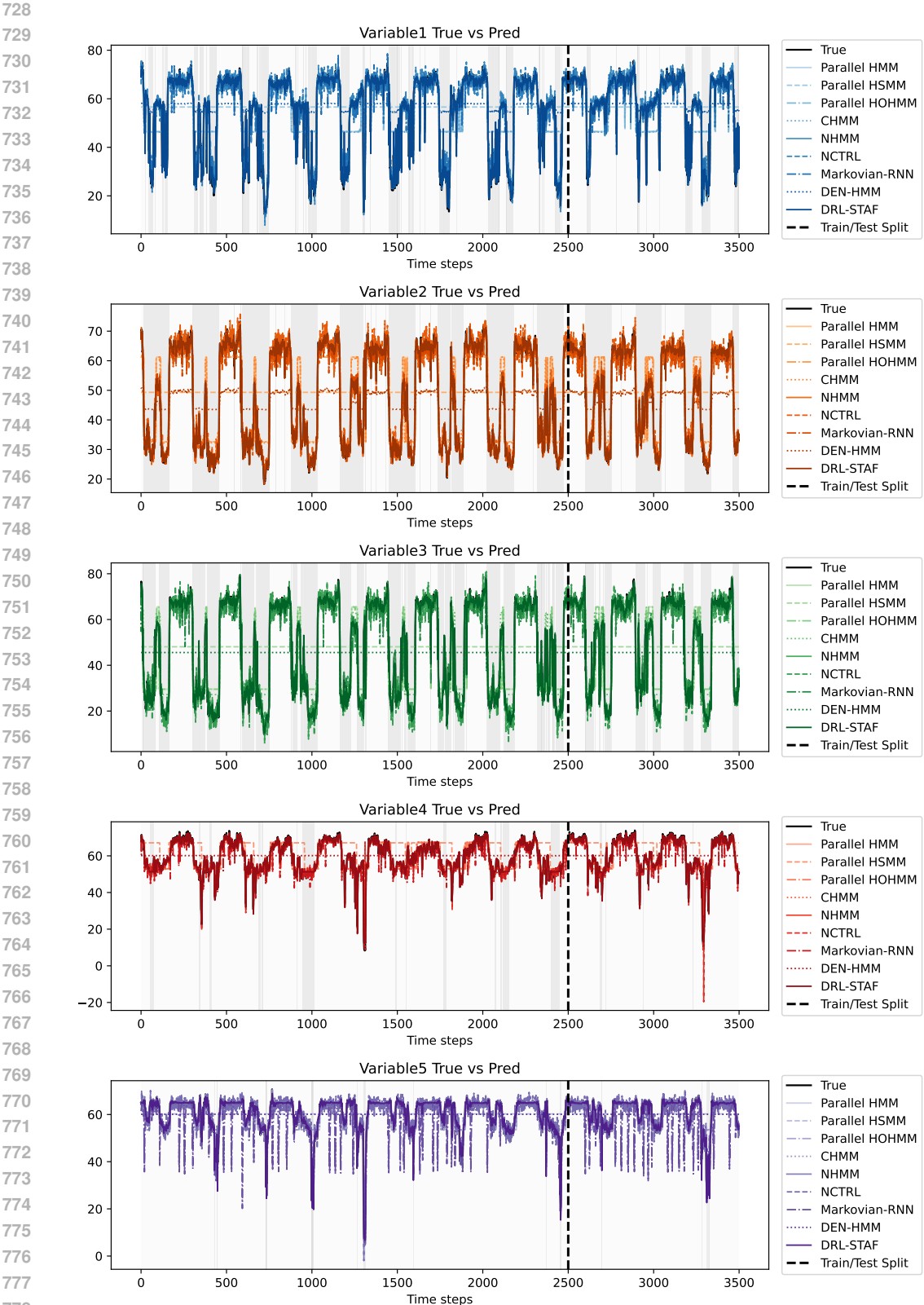

Figure 19: Results of different methods on the Traffic dataset. Background colors indicate different hidden states estimated by DRL-STAF.

