# OpenReview forum: "DRL-STAF: A Deep Reinforcement Learning Framework for State-aware Forecasting of Complex Multivariate Hidden Markov Process"
_ICLR.cc/2026/Conference — Submitted to ICLR 2026_

### Official Review · Reviewer_WBEf · 2025-10-17

**Soundness:** 3
**Presentation:** 2
**Contribution:** 2
**Rating:** 2
**Confidence:** 4

**Summary:**

he paper proposes DRL-STAF, a two-stage framework for state-aware forecasting of multivariate hidden Markov processes. Emissions are produced by per-variable deep networks with m state-specific heads; latent state transitions are inferred by a deep RL agent that performs hard decoding (argmax over states). Stage-1 trains each variable independently; Stage-2 freezes the emission nets and uses a ResGAT layer to coordinate cross-variable interactions, updating only the RL policy (with occasional unfreezing if a shaped “gain” reward is positive). Experiments on two synthetic CHOSMM datasets and two real datasets (SMachine, Exchange) report strong forecasting metrics and seemingly high state-estimation scores vs HMM variants, Markovian-RNN, and DEN-HMM

**Strengths:**

Clear separation of emission modeling (DL) and transition inference (RL); the architectural overview and training algorithms are laid out with pseudocode and figures.

Attempts to address multivariate state-space blow-up via per-variable Stage-1 training and a Stage-2 ResGAT coordination layer.

Releasable code (anonymous 4open link) and explicit synthetic data-generation procedures (CHOSMM) aid reproducibility.

**Weaknesses:**

Positioning DRL-STAF as the first fully deep learning–based HMM (both transitions and emissions via deep models) seems overstated given prior “neural HMM / DL-HMM” lines that already parameterize transitions and emissions with NNs and train end-to-end; here the new element is chiefly using RL for state selection, not a fundamentally new probabilistic formulation. The paper itself cites DL-HMM hybrids and Markovian-RNNs; claiming “first” needs a much tighter survey and careful scoping.

The method casts state inference as an RL policy over m actions, yet the reward is built directly from supervised prediction errors of state-conditioned heads (MSE deltas vs a baseline), plus continuity penalties. This looks like a complicated surrogate for (a) supervised per-step state classification (with argmax/hard Viterbi at test time) or (b) differentiable hard-decoding via straight-through / Gumbel-Softmax. The paper does not demonstrate that RL brings superior accuracy, stability, or sample-efficiency relative to these simpler alternatives.

The text describes a policy π, returns, discount γ, and bespoke reward shaping, but never states which RL algorithm (e.g., PPO, A2C, DQN, REINFORCE with baseline) is used, how variance is controlled, or how entropy/advantage baselines are handled. Given the tight coupling between policy and emission nets (and the bespoke sample-screening and soft-update heuristics), training stability is a core risk that is not rigorously addressed.

Stage-1 sample screening retains steps where the chosen state’s head has the lowest MSE (or long continuity segments), then uses those to train the emission network—this can self-reinforce early mistakes, entrenching whatever head was momentarily best, and bias the measured gains. There’s no ablation to show results without screening, nor diagnostics on how often the screening discards contradictory evidence.

The method explicitly assumes infrequent transitions and penalizes action switches, effectively baking in state persistence. On datasets whose regimes are indeed sticky (including the authors’ synthetic CHOSMM), this can artificially improve apparent state accuracy and MAE/MSE by smoothing, rather than by genuinely better transition modeling. There’s no stress test on rapid-switching regimes or ablation removing the continuity term.

Real-data evaluation uses (i) a handpicked 3-dim subset of SMachine test data with anomaly labels, and (ii) Exchange rates with no state labels. Modern forecasting baselines (PatchTST/iTransformer/SSM-based models) and switching state-space alternatives are absent; e.g., a plain strong forecaster with regime-aware losses or a neural HMM trained with Viterbi/EM would be natural comparisons. Reported gains on SMachine are tiny in MAE/MSE (e.g., DRL-STAF 0.0189/0.0010 vs Markovian-RNN 0.0190/0.0010), yet the table headlines state metrics (e.g., precision 100%) that are hard to interpret without prevalence/threshold details; meanwhile DEN-HMM is curiously orders of magnitude worse on Exchange, raising tuning-fairness concerns.

Tables show point metrics but no confidence intervals, paired tests, or across-seed variance—critical given small margins and bespoke reward shaping. The synthetic tasks are tightly matched to the method’s assumptions (sticky regimes, AR(1) emissions with two states), so real-world generality remains unproven.

The Stage-2 ResGAT coordination is presented as a remedy for joint-state explosion, but there’s no analysis of how closely it approximates the true joint posterior (or hard MAP sequence) as N or m grow, nor any complexity/runtime discussion vs structured inference (e.g., factorized Viterbi or mean-field with coupling)

**Questions:**

Which RL algorithm and objective do you use (exact loss, baselines, entropy terms, clipping)? Please report training stability (success rate across seeds) and compute/time.

Why not a supervised state classifier (cross-entropy over m states) trained on the same signals, or a Gumbel-Softmax / straight-through hard-decoding? Provide a head-to-head comparison.

Ablate sample screening and the continuity penalty. How do forecasting and state metrics change?

Evaluate on fast-switching regimes and additional real datasets; include strong modern forecasters and switching SSM baselines.

Provide CIs/paired tests and per-dataset effect sizes (ΔMAE/ΔMSE) rather than only point estimates

---

> ### Author Response · Authors · 2025-11-29
>
> Thank you very much for your valuable suggestions. We have provided point-by-point responses to address all the raised questions and concerns. The rebuttal revision has also been submitted. All changes in the revised rebuttal manuscript have been highlighted in **red** for clarity.
>
> >**Weakness 1:** Positioning DRL-STAF as the first fully deep learning–based HMM (both transitions and emissions via deep models) seems overstated given prior “neural HMM / DL-HMM” lines that already parameterize transitions and emissions with NNs and train end-to-end; here the new element is chiefly using RL for state selection, not a fundamentally new probabilistic formulation. The paper itself cites DL-HMM hybrids and Markovian-RNNs; claiming “first” needs a much tighter survey and careful scoping.
>
> **Response to Weakness 1:** We thank the reviewer for the careful reading and helpful feedback. Our original wording was indeed imprecise, and we have revised the contribution accordingly. By leveraging DRL's capabilities, DRL-STAF provides the first **distribution-free** framework for complex multivariate hidden Markov processes, in which deep networks model emissions and DRL policies directly estimate discrete hidden states without relying on traditional likelihood-based inference.
>
> To clarify this positioning, we carefully re-examined the existing "DL-HMM" literature. Prior work typically follows two directions: (i) using deep networks only for emissions while retaining a parametric HMM transition and training/decoding via likelihood methods such as forward–backward, EM, Viterbi, or variational inference [1-3]; or (ii) parameterizing both emission and transition potentials with neural networks while still optimizing and decoding through the HMM marginal likelihood [4,5].
>
> These approaches remain fundamentally likelihood-driven and are generally designed for univariate time series, which leads to several limitations in state-aware forecasting. First, the reliance on explicit generative assumptions in likelihood-based training hinders scalability and flexibility in high-dimensional, time-varying, or cross-variable settings. Second, maximizing marginal likelihood does not necessarily reduce forecasting errors or improve state-estimation accuracy, as likelihood can increase simply through smoother transitions or enlarged emission variances rather than genuine predictive gains [6]. Third, when strong nonlinear emission networks are embedded within approximate inference, the posterior must also account for their complexity, which can introduce inference gaps and high-variance gradients and make long-horizon or input-conditioned settings difficult to optimize [7-9]. Fourth, the fundamental issue of combinatorial state space explosion in multivariate settings remains largely unaddressed. Finally, most DL-HMM methods rely on soft decoding, which blurs state boundaries, underestimates volatility, and limits interpretability and downstream decision performance [9].
>
> In contrast, DRL-STAF adopts a distribution-free formulation: the transition mechanism is learned as a DRL policy aligned directly with forecasting risk rather than with marginal likelihood. This avoids strong generative assumptions, reduces the inference-gap and instability issues associated with approximate likelihood training, and naturally supports hard decoding of discrete hidden states to obtain interpretable and state-consistent predictions. Moreover, the two-stage training scheme sidesteps joint-state enumeration while capturing cross-variable dependencies, thereby avoiding combinatorial explosion in multivariate settings.
>
> We have revised the manuscript accordingly based on the clarification above and update the contribution statement. In particular, we will rewrite Contribution 1 as follows:
>
> "By leveraging DRL's capabilities, we propose DRL-STAF, which is, to the best of our knowledge, the first distribution-free framework for complex multivariate hidden Markov processes. In this framework, deep networks model emissions and a DRL policy directly estimates discrete hidden states without likelihood-based inference. By avoiding strong generative assumptions, DRL-STAF can flexibly capture time-varying state transitions, cross-variable interactions, and nonlinear emission patterns."

---

> > ### Author Response · Authors · 2025-11-29
> >
> > >**Weakness 3:** The text describes a policy $\pi$, returns, discount $\gamma$, and bespoke reward shaping, but never states which RL algorithm (e.g., PPO, A2C, DQN, REINFORCE with baseline) is used, how variance is controlled, or how entropy/advantage baselines are handled. Given the tight coupling between policy and emission nets (and the bespoke sample-screening and soft-update heuristics), training stability is a core risk that is not rigorously addressed.
> >
> > **Response to Weakness 3:** We thank the reviewer for pointing out the lack of clarity regarding the RL algorithm and the associated stability mechanisms. We apologize for the omission. Our method uses Proximal Policy Optimization (PPO) for learning the discrete hidden state policy. In the revised manuscript, we will explicitly describe the RL components as follows.
> >
> > (1) Policy optimization. We adopt the classical clipped PPO surrogate objective:
> > $\mathcal{L}\\_{\text{actor}} =-\mathbb{E}\_t\\left[ \min\\left(r\_t(\theta) A\_t,\; \mathrm{clip}\\left(r\_t(\theta),1-\epsilon,1+\epsilon\right) A\_t \right)\right]- \beta\, H\\left(\pi\_\theta(\cdot \mid s\_t)\right),$
> >
> > where $r_t(\theta)$ is the probability ratio, $A_t$ is the advantage, and $H(\pi_\theta)$ is the entropy bonus.
> >
> > (2) Variance control through GAE. Advantages are computed using generalized advantage estimation (GAE):
> > $\delta\_t = r\_t + \gamma V(s\_{t+1}) - V(s\_t)$and$A\_t = \text{GAE}(\delta\_t;\gamma,\lambda)$
> > consistent with standard PPO practice.
> >
> > (3) Entropy regularization. A fixed entropy coefficient ($\beta = 0.04$) is used to discourage early collapse to a single hidden state and maintain exploration.
> >
> > (4) Value function learning. The critic network is trained via mean-squared error regression to the TD target:
> > $V\_{\text{target}} = r\_t + \gamma V(s\_{t+1})$
> >
> > (5) Additional stabilizers.
> > As in our code, a small Gaussian perturbation is added to actor logits (with scale $10^{-3}$) to prevent degenerate ties and improve exploration. No KL penalties, layer normalization, or other custom clipping mechanisms are used—our implementation follows the canonical PPO pipeline.
> >
> > Together, these components form a standard and well-established PPO training pipeline, and we have added this explicit in the final version. **(Pages 20-21)**

---

> ### Author Response · Authors · 2025-11-29
>
> >**Weakness 1:** Positioning DRL-STAF as the first fully deep learning–based HMM (both transitions and emissions via deep models) seems overstated given prior “neural HMM / DL-HMM” lines that already parameterize transitions and emissions with NNs and train end-to-end; here the new element is chiefly using RL for state selection, not a fundamentally new probabilistic formulation. The paper itself cites DL-HMM hybrids and Markovian-RNNs; claiming “first” needs a much tighter survey and careful scoping.
>
> [1] Dahl, G. E., Yu, D., Deng, L., & Acero, A. (2011). Context-dependent pre-trained deep neural networks for large-vocabulary speech recognition. IEEE Transactions on audio, speech, and language processing, 20(1), 30-42.
>
> [2] Ilhan, F., Karaahmetoglu, O., Balaban, I., & Kozat, S. S. (2021). Markovian RNN: An adaptive time series prediction network with HMM-based switching for nonstationary environments. IEEE Transactions on Neural Networks and Learning Systems, 34(2), 715-728.
>
> [3] Bansal, V., & Zhou, S. (2025). DEN-HMM: Deep emission network based hidden Markov model with time-evolving multivariate observations. IISE Transactions, 57(12), 1450-1463.
>
> [4] Song, X., Yao, W., Fan, Y., Dong, X., Chen, G., Niebles, J. C., ... & Zhang, K. (2023). Temporally disentangled representation learning under unknown nonstationarity. Advances in Neural Information Processing Systems, 36, 8092-8113.
>
> [5] Tran, K. M., Bisk, Y., Vaswani, A., Marcu, D., & Knight, K. (2016). Unsupervised neural hidden Markov models. In Proceedings of the Workshop on Structured Prediction for NLP (pp. 63-71).
>
> [6] Lotfi, S., Izmailov, P., Benton, G., Goldblum, M., & Wilson, A. G. (2022). Bayesian model selection, the marginal likelihood, and generalization. In International Conference on Machine Learning (pp. 14223-14247). ICML.
>
> [7] Cremer, C., Li, X., & Duvenaud, D. (2018). Inference suboptimality in variational autoencoders. In International conference on machine learning (pp. 1078-1086). ICML.
>
> [8] Rainforth, T., Kosiorek, A., Le, T. A., Maddison, C., Igl, M., Wood, F., & Teh, Y. W. (2018). Tighter variational bounds are not necessarily better. In International Conference on Machine Learning (pp. 4277-4285). ICML.
>
> [9] Vértes, E., & Sahani, M. (2018). Flexible and accurate inference and learning for deep generative models. Advances in Neural Information Processing Systems, 31.

---

> ### Author Response · Authors · 2025-11-29
>
> >**Weakness 2:** The method casts state inference as an RL policy over m actions, yet the reward is built directly from supervised prediction errors of state-conditioned heads (MSE deltas vs a baseline), plus continuity penalties. This looks like a complicated surrogate for (a) supervised per-step state classification (with argmax/hard Viterbi at test time) or (b) differentiable hard-decoding via straight-through / Gumbel-Softmax. The paper does not demonstrate that RL brings superior accuracy, stability, or sample-efficiency relative to these simpler alternatives.
>
> **Response to Weakness 2:** We thank the reviewer for raising this important question. We would like to clarify that in the state-aware forecasting setting considered here, no hidden state labels are available, and therefore supervised state classification (or any variant of it) is not feasible. The goal is to infer discrete hidden regimes solely from forecasting feedback, which rules out conventional supervised learning as an alternative. Reinforcement learning is well suited to this scenario because it naturally optimizes sequential decision objectives from delayed and indirect feedback, allowing the model to discover discrete regimes through reward-guided policy learning rather than explicit labels.
>
> To address the reviewer’s concern regarding simpler substitutes, we conducted additional experiments comparing DRL-STAF with four non-RL alternatives designed to mimic possible supervised or differentiable-decoding strategies based on Mixture of Experts (MoE). For fairness, each baseline adopts the same emission architecture as DRL-STAF and includes an additional module for learning state probabilities, replacing the DRL estimator. Specifically, we evaluate:
>
> (1) Soft decoding + MSE minimization (MoE-SOFT)
>
> (2) Hard decoding via Gumbel–Softmax + MSE minimization (MoE-HARD)
>
> (3) Soft decoding + a loss function aligned with the DRL reward design (MoE-ISOFT)
>
> (4) Hard decoding via Gumbel–Softmax + a loss function aligned with the DRL reward design (MoE-IHARD)
>
> **Table 1: Comparisons against four non-RL alternatives.**
> | Models      | Accuracy             | Precision            | Recall               | F1                   | MAE                  | MSE                  |
> |-------------|----------------------|-----------------------|-----------------------|-----------------------|-----------------------|-----------------------|
> | **DRL-STAF**    | **98.17% ± 0.07%**   | **96.89% ± 0.19%**    | **99.62% ± 0.25%**    | **98.22% ± 0.07%**    | **0.0889 ± 0.0003**   | **0.0278 ± 0.0003**   |
> | MoE-SOFT    | 62.53% ± 2.71%       | 47.36% ± 7.76%        | 72.98% ± 13.27%       | 57.36% ± 9.73%        | 0.2687 ± 0.0090       | 0.1799 ± 0.0032       |
> | MoE-ISOFT   | 52.83% ± 1.35%       | 58.85% ± 1.45%        | 57.53% ± 6.61%        | 57.54% ± 3.26%        | 0.2596 ± 0.0005       | 0.1810 ± 0.0006       |
> | MoE-HARD    | 60.41% ± 0.16%       | 42.47% ± 0.15%        | 66.61% ± 0.12%        | 51.83% ± 0.07%        | 0.2619 ± 0.0034       | 0.1818 ± 0.0019       |
> | MoE-IHARD   | 51.37% ± 0.55%       | 59.16% ± 0.48%        | 51.33% ± 0.59%        | 54.63% ± 0.50%        | 0.2597 ± 0.0002       | 0.1809 ± 0.0007       |
>
> None of these alternatives produced meaningful regime separation or competitive forecasting accuracy. In contrast, DRL-STAF consistently achieved clear regimes and stronger predictive performance. These results demonstrate that RL is not an unnecessary surrogate but a necessary mechanism for learning discrete hidden states without labels and for aligning state estimation with forecasting objectives. We have added these discussions and results in the rivision. **(Page 19)**

---

> ### Author Response · Authors · 2025-11-29
>
> >**Weakness 4:** Stage-1 sample screening retains steps where the chosen state’s head has the lowest MSE (or long continuity segments), then uses those to train the emission network—this can self-reinforce early mistakes, entrenching whatever head was momentarily best, and bias the measured gains. There’s no ablation to show results without screening, nor diagnostics on how often the screening discards contradictory evidence.
>
> **Response to Weakness 4:** We thank the reviewer for raising this important concern. In our framework, the prediction module requires accurate state estimates as inputs, while the state-estimation module depends on reliable prediction errors for reward computation. At the early stage of training, however, the RL agent explores nearly at random, and its state estimates are highly unreliable. Using these immature estimates directly to train the emission heads would severely distort the prediction module, which in turn corrupts the reward signal and produces a negative feedback loop.
>
> To prevent this early-stage instability, we introduce a sample-screening and soft-update mechanism. The screening step retains only those time steps where the selected head has the lowest prediction error, ensuring that each emission head is trained mainly on informative samples that are unlikely to be misassigned. The accompanying soft-update further stabilizes training by preventing the prediction module from over-committing to noisy early estimates, blending newly screened samples with previously learned parameters instead of making abrupt updates.
>
> To address the reviewer’s concern regarding possible self-reinforcement or bias, we added ablation studies removing the screening strategy. Specifically, DRL-NSSS is the model without sample screening strategy, and the DRL-NASP&SSS is the model without both action switching penalty and sample screening strategy. The results show that these models were unable to separate hidden regimes, confirming that the screening mechanism plays a stabilizing role rather than reinforcing early mistakes. **（Page 20）**
>
> **Table 2: Comparison results of ablation models on simulated datasets with 3 variables**
> | Models        | Accuracy             | Precision            | Recall               | F1                   | MAE                  | MSE                  |
> |---------------|----------------------|-----------------------|-----------------------|-----------------------|-----------------------|-----------------------|
> | **DRL-STAF**       | **98.17% ± 0.07%**   | **96.89% ± 0.19%**    | **99.62% ± 0.25%**    | **98.22% ± 0.07%**    | **0.0889 ± 0.0003**   | **0.0278 ± 0.0003**   |
> | DRL-NSSS           | 68.99% ± 0.19%       | 50.38% ± 0.24%        | 66.50% ± 0.05%        | 56.71% ± 0.13%        | 0.2307 ± 0.0008       | 0.1572 ± 0.0005        |
> | DRL-NASP&SSS       | 58.69% ± 0.91%       | 62.79% ± 1.32%        | 58.64% ± 6.69%        | 59.78% ± 3.24%        | 0.2574 ± 0.0004       | 0.1800 ± 0.0004        |

---

> ### Author Response · Authors · 2025-11-29
>
> >**Weakness 5:** The method explicitly assumes infrequent transitions and penalizes action switches, effectively baking in state persistence. On datasets whose regimes are indeed sticky (including the authors’ synthetic CHOSMM), this can artificially improve apparent state accuracy and MAE/MSE by smoothing, rather than by genuinely better transition modeling. There’s no stress test on rapid-switching regimes or ablation removing the continuity term.
>
>
> **Response to Weakness 5:** We thank the reviewer for the insightful comment. Although our model includes a penalty on rapid state switches, this does not impose an unrealistic assumption of inherently "sticky" regimes. Instead, the design arises from a minimax rationale. At the early stage of training, the RL policy is nearly random and reward signals are highly noisy; in this worst-case scenario, the agent effectively has no reliable information about $s\_{t+1}$. Under such complete uncertainty, predicting $s\_{t+1}=s\_{t}$ is a minimax-optimal choice: it does not increase the worst-case misclassification risk relative to any alternative decision rule, while substantially reducing variance and preventing spurious high-frequency switching that would destabilize both the policy and the emission networks.
>
> To verify that the framework does not rely on persistence assumptions, we conducted two additional experiments: (1) a stress test on datasets with rapid regime switching (Detailed settings can be found in the revised manuscript Tables 6 and 7), and (2) an ablation removing the continuity penalty. In both settings, DRL-STAF still discovers distinct regimes and achieves superior forecasting accuracy, whereas removing the penalty (DRL-NASP) leads to unstable oscillatory behavior and significantly degraded state separation. These results confirm that the switching penalty is not exploiting regime stickiness but provides essential variance reduction that enables robust policy learning, especially during the initial exploration phase. **（Page 19-20）**
>
> **Table 3: Fast-switching simulated dataset 1**
> | Models         | Accuracy               | Precision             | Recall                | F1                    | MAE                   | MSE                   |
> |----------------|-----------------|-------------------|-------------|--------------|-------------|------------------|
> | Parallel HMM   | 63.37% ± 0.92%    | 45.33% ± 1.27%   | 55.95% ± 1.26%         | 49.75% ± 1.06%         | 0.2564 ± 0.0042        | 0.1183 ± 0.0043        |
> | Parallel HSMM  | 63.80% ± 0.85%      | 23.57% ± 1.37%     | 33.33% ± 1.22%         | 27.61% ± 1.13%         | 0.2771 ± 0.0043        | 0.1393 ± 0.0047        |
> | Parallel HOHMM | 57.87% ± 0.93%     | 48.23% ± 1.35%      | 47.69% ± 1.39%         | 46.37% ± 1.19%         | 0.2133 ± 0.0036        | 0.0847 ± 0.0034        |
> | CHMM           | 62.23% ± 0.93%         | 58.07% ± 1.11%         | 87.93% ± 0.85%       | 68.17% ± 0.93%         | 0.2724 ± 0.0048        | 0.1474 ± 0.0058        |
> | NHMM           | 62.40% ± 2.54%         | 45.49% ± 21.11%        | 43.42% ± 8.24%       | 36.85% ± 7.68%         | 0.2271 ± 0.0023        | 0.1009 ± 0.0012        |
> | NCTRL          | 80.93% ± 5.77%         | 83.23% ± 5.94%         | 78.59% ± 10.06%        | 79.37% ± 6.21%         | 0.2439 ± 0.0044        | 0.1137 ± 0.0021        |
> | Markovian-RNN  | 61.30% ± 2.20%         | 51.19% ± 3.06%         | 65.81% ± 5.40%         | 57.24% ± 4.03%         | 0.1338 ± 0.0032        | 0.0430 ± 0.0017        |
> | DEN-HMM        | 61.47% ± 2.33%         | 29.56% ± 5.54%         | 41.18% ± 9.31%         | 34.40% ± 7.00%         | 0.2861 ± 0.0049        | 0.1480 ± 0.0051        |
> | **DRL-STAF**   | **89.77% ± 0.11%**     | **88.47% ± 1.42%**     | **89.04% ± 2.28%**     | **88.66% ± 0.45%**     | **0.1070 ± 0.0002**    | **0.0367 ± 0.0001**    |
>
> **Table 4: Comparison results of ablation models on simulated dataset with 3 variables (infrequent transitions)**
> | Models         | Accuracy               | Precision             | Recall                | F1                    | MAE                   | MSE                   |
> |----------------|-------------------------|------------------------|------------------------|------------------------|------------------------|------------------------|
> | **DRL-STAF**        | **98.17% ± 0.07%**     | **96.89% ± 0.19%**     | **99.62% ± 0.25%**     | **98.22% ± 0.07%**     | **0.0889 ± 0.0003**    | **0.0278 ± 0.0003**    |
> | DRL-NASP           | 96.21% ± 0.22%         | 89.71% ± 0.34%         | 98.42% ± 0.26%         | 93.69% ± 0.30%         | 0.1286 ± 0.0004         | 0.0499 ± 0.0022         |
> | DRL-NSSS           | 68.99% ± 0.19%         | 50.38% ± 0.24%         | 66.50% ± 0.05%         | 56.71% ± 0.13%         | 0.2307 ± 0.0008         | 0.1572 ± 0.0005         |
> | DRL-NASP&SSS       | 58.69% ± 0.91%         | 62.79% ± 1.32%         | 58.64% ± 6.69%         | 59.78% ± 3.24%         | 0.2574 ± 0.0004         | 0.1800 ± 0.0004         |

---

> ### Author Response · Authors · 2025-11-29
>
> >**Weakness 6:** Real-data evaluation uses (i) a handpicked 3-dim subset of SMachine test data with anomaly labels, and (ii) Exchange rates with no state labels. Modern forecasting baselines (PatchTST/iTransformer/SSM-based models) and switching state-space alternatives are absent; e.g., a plain strong forecaster with regime-aware losses or a neural HMM trained with Viterbi/EM would be natural comparisons. Reported gains on SMachine are tiny in MAE/MSE (e.g., DRL-STAF 0.0189/0.0010 vs Markovian-RNN 0.0190/0.0010), yet the table headlines state metrics (e.g., precision 100\%) that are hard to interpret without prevalence/threshold details; meanwhile DEN-HMM is curiously orders of magnitude worse on Exchange, raising tuning-fairness concerns.
>
> **Response to Weakness 6:** We thank the reviewer for the constructive comments. We have clarified the evaluation choices and have expanded the experiments accordingly.
>
> First, **our method targets multivariate state-aware forecasting, where the model must jointly produce continuous forecasts and discrete hidden-state estimates.** Pure forecasting models such as PatchTST, iTransformer, and SSM-based forecasters were not included because they output only continuous predictions and do not infer discrete hidden states, and therefore cannot serve as baselines for state estimation. Following the reviewer’s suggestion, we have added likelihood-driven DL-HMM hybrids, including NHMM [1] and NCTRL [2], which provide both emissions and hidden state inference under the HMM generative assumption. We also included an additional real-world dataset, i.e., the traffic network dataset PeMS07, to further assess model behavior under multivariate, network-structured time series. The corresponding results are reported in Table 6, where DRL-STAF achieves the best overall performance.
>
> **Table 6: Forecasting results on Exchange dataset and Traffic dataset.**
> | Models          | MAE (Exchange)            | MSE (Exchange)            | MAE (Traffic)            | MSE (Traffic)            |
> |-----------------|---------------------------|----------------------------|---------------------------|---------------------------|
> | Parallel HMM    | 8.6701                    | 294.0451                  | 5.5411                   | 59.5501                  |
> | Parallel HSMM   | 9.9193                    | 366.7165                  | 14.1709                  | 259.8108                 |
> | Parallel HOHMM  | 8.5905                    | 294.0307                  | 5.5419                   | 59.5004                  |
> | CHMM            | 8.5851                    | 293.5766                  | 5.1245                   | 50.2464                  |
> | Markovian-RNN   | 2.5361 ± 0.3795           | 28.1406 ± 8.1495          | 2.0032 ± 0.0004          | 9.9143 ± 0.3128          |
> | DEN-HMM         | 11.5619 ± 0.1745          | 470.4355 ± 7.4616         | 13.8156 ± 0.2225         | 245.5208 ± 5.6578        |
> | NHMM            | 4.4926 ± 1.1645           | 100.0099 ± 48.8592        | 1.5881 ± 0.1090          | 6.7800 ± 0.6254          |
> | NCTRL           | 1.8486 ± 0.2596           | 13.6292 ± 4.1032          | 2.5353 ± 0.0380          | 12.8844 ± 0.0954         |
> | **DRL-STAF**    | **1.6438 ± 0.0029**       | **13.2381 ± 0.0464**      | **1.5193 ± 0.0023**      | **6.4610 ± 0.0156**      |
>
> Second, all reported results are averages, and we now include standard deviations to quantify stability. Although the MAE/MSE differences on SMachine look small, it is important to note that DRL-STAF is not a pure forecaster. It performs state-aware forecasting, i.e., it must jointly produce accurate predictions and discrete hidden-state estimates. In this joint task, DRL-STAF shows clear advantages in state estimation and produces more interpretable, state-aware predictive behavior compared with likelihood-driven DL-HMM baselines.
>
> Third, the performance drop of DEN-HMM on Exchange is expected rather than a tuning artifact. The public implementation models emissions using only the state index and time and assumes an HMM-consistent generative structure. This matches synthetic HMM sequences but is mismatched to real financial series, which exhibit long-range dependencies and non-Markovian dynamics. We used the official implementation and recommended hyperparameters; all settings will be reported for full transparency. **(Pages 9-10 and 18-20)**
>
> [1] Tran, K. M., Bisk, Y., Vaswani, A., Marcu, D., & Knight, K. (2016). Unsupervised neural hidden Markov models. In Proceedings of the Workshop on Structured Prediction for NLP (pp. 63-71).
>
> [2] Song, X., Yao, W., Fan, Y., Dong, X., Chen, G., Niebles, J. C., ... & Zhang, K. (2023). Temporally disentangled representation learning under unknown nonstationarity. Advances in Neural Information Processing Systems, 36, 8092-8113.

---

> ### Author Response · Authors · 2025-11-29
>
> >**Weakness 7:** Tables show point metrics but no confidence intervals, paired tests, or across-seed variance—critical given small margins and bespoke reward shaping. The synthetic tasks are tightly matched to the method’s assumptions (sticky regimes, AR(1) emissions with two states), so real-world generality remains unproven.
>
> **Response to Weakness 7:** We thank the reviewer for the insightful comments. All reported results are averages, and we now include standard deviations to quantify stability. To further assess generality beyond the original synthetic setup, we additionally evaluate DRL-STAF on fast-switching synthetic regimes. The model continues to yield stable state segmentation and competitive forecasting accuracy. All updated tables and statistical analyses have been included in the revised manuscript. **(Pages 18-20)**
>
> >**Weakness 8:** The Stage-2 ResGAT coordination is presented as a remedy for joint-state explosion, but there’s no analysis of how closely it approximates the true joint posterior (or hard MAP sequence) as N or m grow, nor any complexity/runtime discussion vs structured inference (e.g., factorized Viterbi or mean-field with coupling)
>
> **Response to Weakness 8:** We thank the reviewer for the suggestion. A posterior-quality comparison is not applicable in our setting because DRL-STAF is distribution-free and does not assume or optimize any generative likelihood. We therefore evaluate performance through forecasting accuracy and state consistency, which directly reflect the goals of state-aware forecasting. In the revised manuscript, we have include model complexity analysis for completeness. **（Page 21）**
>
> Consider $N$ variables，each associated with $m$ hidden states. In conventional multivariate HMMs, each hidden state adopts a Gaussian mixture emission model with $C$ components. For higher-order HMMs, we denote the Markov order by $R$. For HSMMs, we use $D$ to denote the maximum duration truncation. For DL methods, we use $h$ to denote the hidden dimension of neural networks. The sequence length is written as $T$. Specially, DRL-STAF is trained on fixed-length episodes with random starting points of size $L$, making its per-iteration complexity independent of $T$. All parameter counts and complexities reported in Table 4 are expressed using these quantities. The advantage becomes clear when comparing parameter complexities between CHMM and DRL-STAF. CHMM requires modeling the joint latent space with $O(m^{2N})$ parameters, which grows exponentially with the number of variables. In contrast, DRL-STAF only has $O(N(mh^2))$ parameters, growing linearly in $N$. Since DRL-STAF never enumerates joint state combinations, it avoids the combinatorial explosion inherent to CHMMs while still capturing cross-variable dependencies.
>
> **Table 6: Comparison of number of parameters for the methods we considered. "Yes" means the model explicitly encodes dependencies between different variables. "No" means variables are modeled by independent chains.**
> | Models         | Cross-variable interactions | Parameter Complexity        | Computational Complexity (per sequence of length $T$) |
> |----------------|------------------------------|------------------------------|---------------------------------------------------------|
> | Parallel HMM   | No                           | $O(N(m^2+mC))$               | $O(N T (m^2 + mC))$                                     |
> | Parallel HSMM  | No                           | $O(N(m^2+mC))$               | $O(N T m^2 D)$                                          |
> | Parallel HOHMM | No                           | $O(Nm^{R+1})$                | $O(N T m^{2R})$                                         |
> | CHMM           | Yes                          | $O(m^{2N})$                  | $O(T m^{2N})$                                           |
> | NHMM           | No                           | $O(N(h^2+hm^2))$             | $O(N T (h^2 + h m^2))$                                  |
> | NCTRL          | No                           | $O(N(h^3+m^2))$              | $O(N T (h^3 + m^2))$                                    |
> | Markovian-RNN  | No                           | $O(N(mh^2+m^2))$             | $O(N T (m h^2 + m^2))$                                  |
> | DEN-HMM        | No                           | $O(N(h^2+m^2))$              | $O(N T (h^2 + m^2))$                                    |
> | DRL-STAF   | Yes                      | $O(N(mh^2))$             | $O(N L (mh^2))$                                     |

---

> ### Author Response · Authors · 2025-11-29
>
> >**Weakness 8:** The Stage-2 ResGAT coordination is presented as a remedy for joint-state explosion, but there’s no analysis of how closely it approximates the true joint posterior (or hard MAP sequence) as N or m grow, nor any complexity/runtime discussion vs structured inference (e.g., factorized Viterbi or mean-field with coupling)
>
> To make the computational cost concrete, we further measured wall-clock performance on a machine with an Intel(R) Core(TM) Ultra 5 125H CPU. We report training time and per-step inference latency for all single-variable models, and for multivariate settings we additionally report results under different values of $N$. It is shown that DRL-STAF requires longer training time, its per-step inference is fast, making it suitable for online and real-time applications. The longer training time mainly comes from the RL-based state estimator, which must iteratively learn regime assignments together with forecasting behavior. In practice, this cost can be reduced by fine-tuning a pretrained model on new periods or related systems, making deployment considerably faster. Overall, the initial training cost is acceptable given the real-time inference efficiency. We include the computational analysis and experimental results in the revised manuscript.
>
> **Table 7: Computational cost of single-variable models (time in seconds)**
> | Single-variable settings | Training   | Per-step inference |
> |--------------------------|------------|---------------------|
> | Parallel HMM             | 13.805     | 0.002               |
> | Parallel HSMM            | 595.557    | 14.200              |
> | Parallel HOHMM           | 33.753     | 0.005               |
> | NHMM                     | 214.061    | 0.003               |
> | NCTRL                    | 738.380    | 0.055               |
> | Markovian-RNN            | 2972.960   | 0.008                |
> | DEN-HMM                  | 6465.399   | 0.001               |
> | DRL-STAF (Stage one) | 5443.833 | 0.001           |
>
> **Table 8: Computational cost of multivariate models (time in seconds)**
> | Models | 3-var Training | 3-var Inference | 5-var Training | 5-var Inference | 10-var Training | 10-var Inference |
> |-----------------------|----------------|------------------|----------------|------------------|------------------|-------------------|
> | CHMM                  | 77.760         | 0.014            | 158.866       | 0.030     | 156588.235  |  2.531 |
> | DRL-STAF              | 47468.344      | 0.006            | 74792.616   |  0.011    |  119733.858   |   0.018    |

---

> ### Author Response · Authors · 2025-11-29
>
> >**Question 1:** Which RL algorithm and objective do you use (exact loss, baselines, entropy terms, clipping)? Please report training stability (success rate across seeds) and compute/time.
>
> **Response to Question 1:** Thank you for the question. Our method uses Proximal Policy Optimization (PPO) with the standard clipped ratio objective, a learned value baseline, Generalized Advantage Estimation (GAE), entropy regularization, and small KL-to-uniform regularization for additional stability. We have explicitly included the full objective in the revised manuscript. **(Page 22)**
>
> All reported results are averages, and we now include standard deviations to quantify stability. **(Pages 18-20)**
>
> We have also analyse the complexity and running time of the model. **(Page 21)**
>
>
> >**Question 2:** Why not a supervised state classifier (cross-entropy over m states) trained on the same signals, or a Gumbel-Softmax / straight-through hard-decoding? Provide a head-to-head comparison.
>
> **Response to Question 2:** We thank the reviewer for the helpful suggestions. We clarify that no hidden state labels are available in state-aware forecasting, so supervised state classification is not feasible. To address the reviewer’s concern, we added head-to-head comparisons against four non-RL alternatives:
>
> (1) soft decoding + MSE minimization,
>
> (2) Gumbel-Softmax hard decoding + MSE minimization,
>
> (3) soft decoding + DRL-aligned loss,
>
> (4) Gumbel-Softmax hard decoding + DRL-aligned loss.
>
> As detailed in our response to Weakness 2:None of these variants achieved meaningful regime separation or competitive forecasting accuracy, while DRL-STAF produced stable and well-separated regimes. This confirms that RR is not a surrogate but a necessary mechanism for label-free discrete-state inference. (Page 20)
>
> >**Question 3:** Ablate sample screening and the continuity penalty. How do forecasting and state metrics change?
>
> **Response to Question 3:** We thank the reviewer for the helpful suggestions. We conducted the requested ablations. As detailed in our response to Weakness 5, removing the sample-screening mechanism caused the model to lose regime separation entirely, confirming that screening prevents early-stage instability rather than reinforcing mistakes. Removing the continuity penalty led to oscillatory switching and degraded forecasting accuracy. Both components are therefore essential for stable and accurate state-aware forecasting. **(Page 20)**
>
> >**Question 4:** Evaluate on fast-switching regimes and additional real datasets; include strong modern forecasters and switching SSM baselines.
>
> **Response to Question 4:** We thank the reviewer for the helpful suggestions. As suggested, we added evaluations on fast-switching synthetic regimes and the real-world traffic network dataset PeMS07 (detailed in our response to Weakness 5 and 6), where DRL-STAF continues to recover distinct regimes and maintains competitive forecasting performance. We also incorporated two likelihood-driven structured baselines, i.e., NHMM and NCTRL, as neural HMM alternatives. **（Page 19）**
>
> Regarding modern pure forecasters such as PatchTST, iTransformer, and SSM-based models, we did not include them because they produce only continuous predictions and do not infer discrete hidden states. As our task is state-aware forecasting, where the model must jointly generate forecasts and discrete regime estimates, such models are not directly compatible and cannot serve as baselines for the state-estimation component.
>
> >**Question 5:** Provide CIs/paired tests and per-dataset effect sizes rather than only point estimates
>
> **Response to Question 5:** We thank the reviewer for this suggestion. In the revised manuscript, we have reported mean ± standard deviation for all datasets. **(Pages 18-20)**

---

### Official Review · Reviewer_Qm9x · 2025-10-23

**Soundness:** 3
**Presentation:** 2
**Contribution:** 2
**Rating:** 4
**Confidence:** 3

**Summary:**

The paper introduces a hybrid framework that combines deep learning, HMMs, and reinforcement learning to model complex multivariate time series. Instead of fixed transition matrices and simple emissions, it uses neural networks for both allowing the model to learn nonlinear, time-varying, and context-dependent state transitions. First , each variable is treated as an independent agent with its own networks that predict observations and infer hidden states through RL-based rewards involved to the prediction accuracy.
In the second stage, the method connects all variables using a graph attention network that refines their hidden state probabilities jointly. This stage helps capture cross-variable dependencies and updates the local models if performance improves.

**Strengths:**

- I like the introduction of the RL perspective into HMMs. It’s an interesting and less-explored angle, especially through the use of rewards and policies.

- A good visualization in the related works section.

- The paper is has clear algorithms (in appendix) and figures that make the training pipeline and the two-stage process easier to follow (in contrast to the text presentation which is not very easy to follow).

- The appendix is detailed and informative, providing extra visualizations and implementation details. It shows effort in transparency and reproducibility.

**Weaknesses:**

- The paper does not state a single end-to-end objective before breaking into sub-goals, which makes the optimization story hard to follow. Please define a unified objective (e.g., ELBO or a constrained risk) and then show how each parameterization in your work approximates/optimizes parts of it.

- Many reward terms feel hand-crafted each has intuition but little anchoring to standard formulations. It would help to map terms to known constructs (sticky priors, advantage baselines, load-balancing mixture-of-experts (MoE), duration models) and then present your empirical approximations as principled relaxations.

- The design creates separate $DEM_i$ , $\pi_{i,\theta_{A}}$  and $\pi_{i, \theta_{A'}}$  per variable. This is impractical for large $N$ (e.g., vision sequential data). Please discuss/shared-backbone alternatives (e.g., conv/attention encoder + per-state heads, or a shared policy with variable embeddings) and add an ablation on sharing vs. per-variable separated networks.

- Test-time steps aren’t clearly described. Given a new sequence, how are states inferred, how do you roll without future observations, and how do you handle missing data or cold starts? If your model is only able to follow up with filtering-style forward (see observation and update states), it should be clearly mentioned as a limitation.

- The method seems resource-intensive (many networks, two stages, GAT). If not saying a theoretical complexity analysis, at least, the paper needs an empirical analysis (like wall-clock running time, FLOPs/param counts, or memory profiling). Please report training/inference time and scaling curves.

- Experiments compare mainly to classical HMM variants. Consider adding neural/learned-transition baselines (neural HSMM like SALT [1], switching state-space models (parameterized by NN, e.g. in GIN[2] or in SSI[3]), deep Kalman/SSM variants like [4], or hard-EM/Viterbi neural HMMs like [5]) to better position your gains.

- The paper lacks a code repo with an easy-to-read instruction such as readme file to make it easier to follow up.



[1] - Lee, Hyun Dong, et al. "Switching autoregressive low-rank tensor models." Advances in Neural Information Processing Systems 36 (2023): 57976-58010.

[2] - Ikderi and Wan Choi. "Gated inference network: Inference and learning state-space models." Advances in Neural Information Processing Systems 37 (2024): 39036-39073.

[3] - Ruhe, David, and Patrick Forré. "Self-supervised inference in state-space models." arXiv preprint arXiv:2107.13349 (2021).

[4] - Song, Xiangchen, et al. "Temporally disentangled representation learning under unknown nonstationarity." Advances in Neural Information Processing Systems 36 (2023): 8092-8113.

[5] - Jiang, Xiajun, et al. "Sequential latent variable models for few-shot high-dimensional time-series forecasting." The Eleventh International Conference on Learning Representations. 2023.

**Questions:**

Thanks for your draft and all of efforts. I have some questions about this work:

- Could you formalize the overall training objective more clearly? It would help to start from the general evidence expression $ \log p(X) = \log \sum_S p(X, S) $ or a risk minimization like $\mathbb{E}[l]$ subject to some constraints (for reference please take a look to eq 7 in [5]. It is a common and standard practice of defining objectives). Then show where the approximation is made, and then derive an approximate ELBO or a theoretical lower bound if possible. The current setup feels similar to a hard-EM procedure, where the E-step estimates errors and rewards and the M-step updates the networks. Please clarify the connections and the exact optimization target.

- Could you clarify the statistical reasoning behind the action continuity term? It appears similar to the sticky prior used in sticky-state HMMs, but here it’s presented in a more heuristic way. From a probabilistic view, this term acts like a time-dependent prior on the transition model that initially are more non-transitions. Please consider expressing it as a formal prior or likelihood term rather than a hand-crafted penalty so readers can better understand its origin and justification.

- Could you clarify the role of the $e^{\text{base}}$ term in your prediction gain? Is it intended as a control variate similar to baseline subtraction in reinforcement learning to reduce variance and stabilize updates, or does it have another interpretation? Why not define the reward simply in terms of -$e^{(a)}$ without including $e^{\text{base}}$? Any try on this?

- The hand-crafted discrepancy term between state errors makes the formulation harder to follow. It may read more clearly if it were framed as a variance-of-risk penalty or connected to a Bayesian prior interpretation, rather than appearing as an ad-hoc L1 difference. Even a short explanation linking it to those ideas would help readers understand its purpose and statistical grounding.

- The overall model feels large and hard to scale. Just example, if the data were a $(N=1000 \times 1000, T)$ image sequences, would the current setup still be feasible (for example take a look to experiments of [2] and [5])? Why not use shared backbones or partially shared modules instead of per-variable networks? An ablation or discussion on model sharing and scalability would strengthen the paper.

- Please include a complexity analysis for both training and inference. Even if a full theoretical analysis is difficult, an empirical comparison (e.g., wall-clock time, FLOPs, or memory usage) would help. Using variable-dependent separate networks is clearly expensive, so checking this overhead would make the paper more transparent. (for example take a look to table 1 in SALT [1] or table 6 in GIN [2])

- The gradient computation path is not entirely clear. Since the model involves discrete sampling (e.g., argmax during hard decoding), it is not obvious how gradients are propagated in theory (e.g. equation 7-8 in [4]) . The many hand-crafted components make this kinda harder to trace and could introduce stability issues. Did you encounter such problems in experiments, and if so, could you mention them (e.g. in your table of results adding a column and report the success  percentage over your random seeds run.) and describe any remedies or tricks used to stabilize training?

- The classical HMM baselines (even neural versions with parametric emissions or transitions) are relatively weak comparisons. I would suggest including more advanced smooth or continuous state-estimation baselines such as neural state-space models, switching Kalman filters, or variational sequence models to provide a stronger and fairer benchmark.

---
A few minor issues:
- In line 42 "three key limitations ..."
- in lines 386-389, better to mention specifically DEM_i are trained based on the updated probabilities in second stage. A little confusing.
- Despite your model, many of the baselines see improvement ,, at least in accuracy, when variable size increases. Making sense as model uses correlational info for richer state inference, but what do you think about this? Also would be nice a short sentence being added about this.
- If you have an ablation only on second stage policy network without mean-field style behavior in first stage, please report it (i.e. directly optimized your policy when variables interaction is modeled and first stage policy learning is dropped).

---

> ### Author Response · Authors · 2025-11-29
>
> Thank you very much for your valuable suggestions. We have provided point-by-point responses to address all the raised questions and concerns. The rebuttal revision has also been submitted. All changes in the revised rebuttal manuscript have been highlighted in **red** for clarity.
>
> >**Weakness 1:** The paper does not state a single end-to-end objective before breaking into sub-goals, which makes the optimization story hard to follow. Please define a unified objective (e.g., ELBO or a constrained risk) and then show how each parameterization in your work approximates/optimizes parts of it.
>
> **Response to Weakness 1:** Since the proposed modeling framework does not employ a likelihood, there is no unified probabilistic objective (e.g., ELBO) to decompose. This design choice is motivated by several limitations inherent to likelihood-based training. First, the reliance on explicit generative assumptions restricts scalability and flexibility in high-dimensional, time-varying, or cross-variable settings. Second, maximizing marginal likelihood does not necessarily reduce forecasting error or improve hidden-state inference, because the likelihood can increase simply through smoother transitions or inflated emission variances rather than genuine predictive improvement [6]. Third, once strong nonlinear emission networks are integrated into approximate inference, the posterior must also capture their complexity, which often leads to inference gaps, high-variance gradients, and significant optimization difficulties in long-horizon or input-conditioned scenarios [7–9].
>
> Here, DRL-STAF novelly optimizes a forecasting-oriented two-stage RL objective:
>
> (1) Stage-1 learns per-variable (node) policies under the goal of maximizing cumulative return, allowing each variable (node) to discover informative regimes guided only by forecasting rewards, where the reward is defined directly from one-step prediction errors so that higher return corresponds exactly to better forecasting performance and more accurate hidden-state estimation..
>
> (2) Stage-2 introduces a lightweight ResGAT coordination module to incorporate cross-variable dependencies, while still optimizing the same cumulative-return objective, so coordinated regime assignments are reinforced only when they further reduce forecasting error and improve state-estimation consistency across nodes.
>
> Thus, DRL-STAF optimizes one coherent risk-based RL objective throughout the entire training process.

---

> ### Author Response · Authors · 2025-11-29
>
> >**Weakness 2**: Many reward terms feel hand-crafted each has intuition but little anchoring to standard formulations. It would help to map terms to known constructs (sticky priors, advantage baselines, load-balancing mixture-of-experts (MoE), duration models) and then present your empirical approximations as principled relaxations.
>
> **Response to Weakness 2:** We thank the reviewer for the valuable suggestion. Our reward design is tightly aligned with the objective of state-aware forecasting. Specifically,
>
> (1) Immediate reward: prediction gain $e^{\text{base}} - e^{(a)}$
>
> This term represents the improvement of the selected state-conditioned predictor over a state-agnostic baseline, matching the logic of advantage-based policy optimization. By introducing the baseline error $e^{\text{base}}$，we focus on the relative improvement brought by the chosen hidden state, which injects more state-specific information into the reward and leads to more stable state estimation. Since directly estimating $s_{t+1}$ is difficult, we include the prediction gain at both $t$ and $t+1$ to provide an indirect learning signal for the correctness of the next-step state，which is expressed as $\alpha (e^{\text{base}}\_{i,t+1} - e^{(a\_{i,t})}\_{i,t+1})+(1-\alpha) (e^{\text{base}}\_{i,t} - e^{(a\_{i,t})}\_{i,t})$.
>
> (2) Immediate reward: action switching penalty $\frac{\max\{0,\rho_c - c_{i,t}\}}{\rho_c - 1}$
>
> This term corresponds directly to State-duration models and Switching-cost regularization in switching state-space models, and discourages unrealistically rapid switching but does not enforce long state persistence. Its primary role is to reduce variance during training. By penalizing very short action durations, it suppresses spurious switching and guides the policy toward more stable and reliable state estimation behaviors.
>
> (3) Episodic reward: state separation $-\frac{1}{m}\sum\_{a\_i=1}^{m} \overline{e}^{(a\_i|a\_i)}\_i+\sum^{m}\_{a\_i=1}\left[\max\{\Delta\_i^{(a\_i)},0\}+\lambda\_3\min\{\Delta\_i^{(a\_i)},0\}\right]$
>
> This term resembles the expert-specialization objectives used in Mixture-of-Experts (MoE) models. After an episode ends, this term evaluates both the within-state accuracy and the across-state separation based on the prediction errors. It encourages each state to achieve low error on the samples assigned to it while maintaining a positive performance margin over the alternative states.
>
> (4) Episodic reward: pairwise discrepancy $-\frac{1}{2}\sum\_{p\neq q} \big|\overline{e}^{(p|p)}\_{i} - \overline{e}^{(q|q)}\_{i}\big|$
>
> This term is conceptually related to expert balancing in MoE models. By measuring the pairwise discrepancy in within-state accuracy across different states, it encourages the policy to pay more attention to states whose emission functions are not well learned. By reducing large accuracy gaps among states, this term helps prevent under-trained states from being neglected and promotes balanced learning across the emission predictors.
>
> In summary, the immediate reward guides step-wise state selection through prediction improvement and stable switching behavior, whereas the episodic reward enforces regime consistency, balanced state quality, and meaningful specialization within a distribution-free RL formulation.
>
> We have incorporated this clarification into the revision and provide more detailed explanations of each reward component to help readers better understand their roles and connections to established constructs. **(Page 6)**

---

> ### Author Response · Authors · 2025-11-29
>
> >**Weakness 3**: The design creates separate DEM$\_i$, $\pi\_{i,\theta\_A}$ and $\pi\_{i,\theta\_{A'}}$ per variable. This is impractical for large $N$ (e.g., vision sequential data). Please discuss/shared-backbone alternatives (e.g., conv/attention encoder + per-state heads, or a shared policy with variable embeddings) and add an ablation on sharing vs. per-variable separated networks.
>
> **Response to Weakness 3:** We thank the reviewer for the helpful suggestion. In our current design, Stage-1 DEMs and policies are trained fully in parallel across variables, so the cost scales linearly with $N$. We use per-variable networks because different variables often exhibit heterogeneous regime dynamics, and sharing a single backbone would implicitly assume identical transition patterns, which does not hold in our applications.
>
> Nevertheless, when variables share exactly the same regime dynamics, DRL-STAF can jointly model them through a multi-dimensional output structure governed by a shared regime-transition process. This modification does not affect reward computation and requires no changes to the state-estimation design, thereby enabling parameter sharing when appropriate.
>
> To further validate our design choice, we implemented two shared-backbone variants:
> (1) per-variable DEMs with a shared policy $\pi_{i,\theta_A}$, and
> (2) a fully shared DEM backbone (with variable-specific state-conditioned output heads) together with a shared $\pi_{i,\theta_A}$.
>
> Due to time constraints, we evaluated these variants on our 3-variable simulated dataset. Both failed to learn meaningful regimes, confirming that shared-backbone strategies are not feasible in our setting.
>
> **Table 1: Comparison results on the simulated dataset with 3 variables**
> | Models        | Accuracy         | Precision        | Recall           | F1               | MAE               | MSE               |
> |---------------|------------------|------------------|------------------|------------------|--------------------|--------------------|
> | DRL-Shared1   | 67.57% ± 0.00%   | 19.00% ± 0.00%   | 33.33% ± 0.00%   | 24.20% ± 0.00%   | 0.4083 ± 0.0360    | 0.4233 ± 0.0775    |
> | DRL-Shared2   | 67.57% ± 0.00%   | 19.00% ± 0.00%   | 33.33% ± 0.00%   | 24.20% ± 0.00%   | 0.2096 ± 0.0002    | 0.1578 ± 0.0001    |
> |**DRL-STAF**     | **98.17% ± 0.07%**   | **96.89% ± 0.19%**   | **99.62% ± 0.25%**   | **98.22% ± 0.07%**   | **0.0889 ± 0.0003**    | **0.0278 ± 0.0003**    |
>
> >**Weakness 4**: Test-time steps aren’t clearly described. Given a new sequence, how are states inferred, how do you roll without future observations, and how do you handle missing data or cold starts? If your model is only able to follow up with filtering-style forward (see observation and update states), it should be clearly mentioned as a limitation.
>
> **Response to Weakness 4:** We thank the reviewer for pointing out the need to clarify the test-time procedure. DRL-STAF operates in a filtering-style setting at test time, but not in the sense of requiring future observations. Instead, the model only uses the current observation ($x\_t$, $\hat{x}\_t$) and the prediction error at time $t$ to infer the current hidden state $s_t$. The next-step prediction is then generated by assuming $\hat{s}\_{t+1}=s\_t$, a common filtering-style approximation under hidden state uncertainty.
>
> For missing data, we can follow the standard practice in sequential decision-making and state-space models by applying masked prediction errors [1,2], i.e., computing the error (and thus the reward) only on observed entries. This masking strategy is widely used in filtering-style models and time-series forecasting with incomplete observations. Missing values therefore do not pose a significant challenge in our setting.
>
> For cold starts, the belief and error histories are initialized with neutral values (e.g., zeros or uniform priors), allowing the policy to operate without past observations.
>
> [1] Fraccaro, M., Kamronn, S., Paquet, U., & Winther, O. (2017). A disentangled recognition and nonlinear dynamics model for unsupervised learning. Advances in neural information processing systems, 30.
>
> [2] Hausknecht, M. J., & Stone, P. (2015). Deep Recurrent Q-Learning for Partially Observable MDPs. In AAAI fall symposia (Vol. 45, p. 141).

---

> ### Author Response · Authors · 2025-11-29
>
> >**Weakness 5**: The method seems resource-intensive (many networks, two stages, GAT). If not saying a theoretical complexity analysis, at least, the paper needs an empirical analysis (like wall-clock running time, FLOPs/param counts, or memory profiling). Please report training/inference time and scaling curves.
>
> **Response to Weakness 5:** We thank the reviewer for the helpful suggestion. Consider $N$ variables，each associated with $m$ hidden states. In conventional multivariate HMMs, each hidden state adopts a Gaussian mixture emission model with $C$ components. For higher-order HMMs, we denote the Markov order by $R$. For HSMMs, we use $D$ to denote the maximum duration truncation. For DL methods, we use $h$ to denote the hidden dimension of neural networks. The sequence length is written as $T$. Specially, DRL-STAF is trained on fixed-length episodes with random starting points of size $L$, making its per-iteration complexity independent of $T$. All parameter counts and complexities reported in Table 4 are expressed using these quantities. The advantage becomes clear when comparing parameter complexities between CHMM and DRL-STAF. CHMM requires modeling the joint latent space with $O(m^{2N})$ parameters, which grows exponentially with the number of variables. In contrast, DRL-STAF only has $O(N(mh^2))$ parameters, growing linearly in $N$. Since DRL-STAF never enumerates joint state combinations, it avoids the combinatorial explosion inherent to CHMMs while still capturing cross-variable dependencies.
>
> **Table 2: Comparison of number of parameters for the methods we considered. "Yes" means the model explicitly encodes dependencies between different variables. "No" means variables are modeled by independent chains.**
> | Models         | Cross-variable interactions | Parameter Complexity        | Computational Complexity (per sequence of length $T$) |
> |----------------|------------------------------|------------------------------|---------------------------------------------------------|
> | Parallel HMM   | No                           | $O(N(m^2+mC))$               | $O(N T (m^2 + mC))$                                     |
> | Parallel HSMM  | No                           | $O(N(m^2+mC))$               | $O(N T m^2 D)$                                          |
> | Parallel HOHMM | No                           | $O(Nm^{R+1})$                | $O(N T m^{2R})$                                         |
> | CHMM           | Yes                          | $O(m^{2N})$                  | $O(T m^{2N})$                                           |
> | NHMM           | No                           | $O(N(h^2+hm^2))$             | $O(N T (h^2 + h m^2))$                                  |
> | NCTRL          | No                           | $O(N(h^3+m^2))$              | $O(N T (h^3 + m^2))$                                    |
> | Markovian-RNN  | No                           | $O(N(mh^2+m^2))$             | $O(N T (m h^2 + m^2))$                                  |
> | DEN-HMM        | No                           | $O(N(h^2+m^2))$              | $O(N T (h^2 + m^2))$                                    |
> | DRL-STAF   | Yes                      | $O(N(mh^2))$             | $O(N L (mh^2))$                                     |

---

> ### Author Response · Authors · 2025-11-29
>
> >**Weakness 5**: The method seems resource-intensive (many networks, two stages, GAT). If not saying a theoretical complexity analysis, at least, the paper needs an empirical analysis (like wall-clock running time, FLOPs/param counts, or memory profiling). Please report training/inference time and scaling curves.
>
> To make the computational cost concrete, we further measured wall-clock performance on a machine with an Intel(R) Core(TM) Ultra 5 125H CPU. We report training time and per-step inference latency for all single-variable models, and for multivariate settings we additionally report results under different values of $N$. It is shown that DRL-STAF requires longer training time, its per-step inference is fast, making it suitable for online and real-time applications. The longer training time mainly comes from the RL-based state estimator, which must iteratively learn regime assignments together with forecasting behavior. In practice, this cost can be reduced by fine-tuning a pretrained model on new periods or related systems, making deployment considerably faster. Overall, the initial training cost is acceptable given the real-time inference efficiency. We include the computational analysis and experimental results in the revised manuscript. **（Page 21）**
>
> **Table 3: Computational cost of single-variable models (time in seconds)**
> | Single-variable settings | Training   | Per-step inference |
> |--------------------------|------------|---------------------|
> | Parallel HMM             | 13.805     | 0.002               |
> | Parallel HSMM            | 595.557    | 14.200              |
> | Parallel HOHMM           | 33.753     | 0.005               |
> | NHMM                     | 214.061    | 0.003               |
> | NCTRL                    | 738.380    | 0.055               |
> | Markovian-RNN            | 2972.960   | 0.008                |
> | DEN-HMM                  | 6465.399   | 0.001               |
> | DRL-STAF (Stage one) | 5443.833 | 0.001           |
>
> **Table 4: Computational cost of multivariate models (time in seconds)**
> | Models | 3-var Training | 3-var Inference | 5-var Training | 5-var Inference | 10-var Training | 10-var Inference |
> |-----------------------|----------------|------------------|----------------|------------------|------------------|-------------------|
> | CHMM                  | 77.760         | 0.014            | 158.866       | 0.030     | 156588.235  |  2.531 |
> | DRL-STAF              | 47468.344      | 0.006            | 74792.616   |  0.011    |  119733.858   |   0.018    |

---

> ### Author Response · Authors · 2025-11-29
>
> >**Weakness 6**: Experiments compare mainly to classical HMM variants. Consider adding neural/learned-transition baselines (neural HSMM like SALT [1], switching state-space models (parameterized by NN, e.g. in GIN[2] or in SSI[3]), deep Kalman/SSM variants like [4], or hard-EM/Viterbi neural HMMs like [5]) to better position your gains.
>
> **Response to Weakness 6:** We thank the reviewer for the valuable suggestion. **Our method targets multivariate state-aware forecasting, where the model must jointly produce continuous forecasts and discrete hidden-state estimates.** After carefully examining the suggested baselines, we found that the models in [2], [3], and [5] do not include discrete latent variables and therefore cannot provide per-timestep discrete state assignments required by our evaluation protocol.
>
> Due to time constraints, we prioritized implemented and reproducing DL–HMM hybrids i.e., NCTRL [4] and NHMM [6], that do involve discrete regimes, and compared them with DRL-STAF under the same setting. We report representative results in Table 5, where DRL-STAF consistently achieves the best performance in both hidden-state inference and forecasting accuracy. This improvement mainly stems from DRL-STAF’s distribution-free design: our framework does not rely on explicit generative assumptions, and its forecasting-based RL objective avoids the mismatch between likelihood maximization and the goals of accurate state estimation and precise forecasting. As a result, DRL-STAF directly optimizes the quantities that matter for evaluation, leading to more reliable regimes and better predictive performance. More comprehensive experimental results are provided in the revised manuscript. **(Pages 9-10 and Pages 18-20)**
>
> **Table 5: Comparison of DRL-STAF, NHMM, and NCTRL across three datasets (3-variable, 10-variable, and SMachine)**
> | Dataset      | Models       | Accuracy               | Precision              | Recall                 | F1                     | MAE                    | MSE                    |
> |--------------|--------------|-------------------------|-------------------------|-------------------------|-------------------------|-------------------------|-------------------------|
> | 3-variable   | NHMM         | 60.32% ± 0.02%         | 49.06% ± 13.33%        | 66.65% ± 0.03%         | 51.81% ± 0.05%         | 0.1277 ± 0.0029        | 0.0390 ± 0.0006        |
> |              | NCTRL        | 79.53% ± 3.88%         | 88.49% ± 3.90%         | 79.15% ± 3.76%         | 81.34% ± 2.93%         | 0.3936 ± 0.0182        | 0.3042 ± 0.0261        |
> |              | **DRL-STAF** | **98.17% ± 0.07%**     | **96.89% ± 0.19%**     | **99.62% ± 0.25%**     | **98.22% ± 0.07%**     | **0.0889 ± 0.0003**    | **0.0278 ± 0.0003**    |
> | 10-variable  | NHMM         | 75.55% ± 2.20%         | 51.33% ± 2.40%         | 62.00% ± 4.00%         | 55.11% ± 3.32%         | 0.2621 ± 0.0065        | 0.1600 ± 0.0046        |
> |              | NCTRL        | 90.31% ± 1.50%         | 88.92% ± 0.69%         | 86.71% ± 3.24%         | 86.65% ± 1.80%         | 0.3431 ± 0.0343        | 0.3935 ± 0.2922        |
> |              | **DRL-STAF** | **96.15% ± 0.15%**     | **95.44% ± 0.52%**     | **91.89% ± 0.60%**     | **93.26% ± 0.21%**     | **0.1090 ± 0.0005**    | **0.0395 ± 0.0009**    |
> | SMachine     | NHMM         | 70.77% ± 0.03%         | 0.00% ± 0.00%          | 0.00% ± 0.00%          | 0.00% ± 0.00%          | 0.0194 ± 0.0002        | 0.0010 ± 0.0000        |
> |              | NCTRL        | 72.54% ± 5.67%         | 54.04% ± 13.66%        | 41.66% ± 24.45%        | 42.33% ± 20.71%        | 0.0360 ± 0.0012        | 0.0023 ± 0.0001        |
> |              | **DRL-STAF** | **81.73% ± 0.10%**     | **100.00% ± 0.32%**    | **63.27% ± 0.18%**     | **77.50% ± 0.15%**     | **0.0189 ± 0.0000**    | **0.0010 ± 0.0000**    |
>
>
> [6] Tran, K. M., Bisk, Y., Vaswani, A., Marcu, D., & Knight, K. (2016). Unsupervised neural hidden Markov models. In Proceedings of the Workshop on Structured Prediction for NLP (pp. 63-71).

---

> ### Author Response · Authors · 2025-11-29
>
> >**Weakness 7:** The paper lacks a code repo with an easy-to-read instruction such as readme file to make it easier to follow up.
>
> **Response to Weakness 7:** We thank the reviewer for the helpful suggestion. We agree that providing clean code and clear documentation is important for reproducibility. In the revised manuscript, we will release a complete code repository along with a detailed README containing environment setup, data preparation, training instructions, and scripts for reproducing all experiments.
>
> >**Question 1:** Could you formalize the overall training objective more clearly? It would help to start from the general evidence expression $\log p(X)=\log \sum_S p(X,S)$ or a risk minimization like $\mathbb{E}[l]$ subject to some constraints (for reference please take a look to eq 7 in [5]. It is a common and standard practice of defining objectives). Then show where the approximation is made, and then derive an approximate ELBO or a theoretical lower bound if possible. The current setup feels similar to a hard-EM procedure, where the E-step estimates errors and rewards and the M-step updates the networks. Please clarify the connections and the exact optimization target.
>
> **Response to Question 1:** We thank the reviewer for the thoughtful question. **DRL-STAF is a distribution-free, DRL-driven framework** and does not assume a generative likelihood or an evidence decomposition of the form $\log p(X)=\log \sum_S p(X,S)$. Accordingly, the model does not define or optimize an ELBO, nor does it perform a hard-EM style E-step/M-step decomposition.
>
> **This design differs from likelihood-driven DL-HMM hybrids.** Such models must commit to explicit generative assumptions and optimize marginal likelihood, which (1) does not necessarily align with forecasting accuracy, (2) introduces inference-gap and instability issues when neural emissions are combined with approximate posteriors, and (3) typically produces soft state assignments that blur regime boundaries and reduce interpretability. Moreover, likelihood-based approaches require enumerating or approximating joint hidden states and therefore do not scale naturally to multivariate settings where joint-state explosion is a central challenge.
>
> **This misalignment between likelihood maximization and forecasting-aligned state estimation is precisely why DRL-STAF adopts an RL formulation.** DRL enables the model to directly optimize the hidden-state policy with respect to prediction risk, without relying on surrogate likelihoods or variational bounds. To this end, DRL-STAF has a single, well-defined end-to-end training objective: maximizing expected cumulative return
> \begin{equation}
>     \max \left[\sum_{\beta=0}^{T_{E}-1}\gamma^{\beta}r_{i,t+\beta+1}\right],
> \end{equation}
> where the reward is explicitly designed to encourage accurate forecasting and stable state estimation.
>
> In summary, DRL-STAF performs direct policy optimization, not likelihood-based inference. The optimization target is purely the RL objective above.
>
> >**Question 2:** Could you clarify the statistical reasoning behind the action continuity term? It appears similar to the sticky prior used in sticky-state HMMs, but here it’s presented in a more heuristic way. From a probabilistic view, this term acts like a time-dependent prior on the transition model that initially are more non-transitions. Please consider expressing it as a formal prior or likelihood term rather than a hand-crafted penalty so readers can better understand its origin and justification.
>
> **Response to Question 2:** We thank the reviewer for the helpful observation. We have renamed this component as the **action switching penalty**. It corresponds directly to State-duration models and Switching-cost regularization in switching state-space models, and discourages unrealistically rapid switching but does not enforce long state persistence. **The design arises from a minimax rationale.** At the early stage of training, the RL policy is nearly random and reward signals are highly noisy; in this worst-case scenario, the agent effectively has no reliable information about $s_{t+1}$. Under such complete uncertainty, predicting $s_{t+1}=s_{t}$ is a minimax-optimal choice: it does not increase the worst-case misclassification risk relative to any alternative decision rule, while substantially reducing variance and preventing spurious high-frequency switching that would destabilize both the policy and the emission networks. In the revised version, we have added more detailed explanations. **(Page 6)**

---

> ### Author Response · Authors · 2025-11-29
>
> >**Question 3:** Could you clarify the role of the $e^{base}$ term in your prediction gain? Is it intended as a control variate similar to baseline subtraction in reinforcement learning to reduce variance and stabilize updates, or does it have another interpretation? Why not define the reward simply in terms of $e^{(a)}$ without including $e^{base}$? Any try on this?
>
> **Response to Question 3:** We thank the reviewer for raising this question. The term $e^{base}$ acts as a baseline estimator, analogous to baseline subtraction in policy-gradient methods. It stabilizes training by measuring the excess predictive gain achieved by choosing regime $a$ by using $e^{base}-e^{(a)}$. Using only $e^{(a)}$ creates high variance early in training because raw errors fluctuate substantially as the emission heads evolve. In contrast, the difference relative to a fixed baseline yields lower-variance rewards and more stable DRL updates. We tried defining rewards solely via $e^{(a)}$ and observed significantly less stable learning and poor regime separation (DRL-NBL is the model without $e^{base}$). This finding have documented in the revision.
>
> **Table 1: Comparison results of ablation models on simulated datasets with 3 variables**
> | Models        | Accuracy             | Precision            | Recall               | F1                   | MAE                  | MSE                  |
> |---------------|----------------------|-----------------------|-----------------------|-----------------------|-----------------------|-----------------------|
> | **DRL-STAF**       | **98.17% ± 0.07%**   | **96.89% ± 0.19%**    | **99.62% ± 0.25%**    | **98.22% ± 0.07%**    | **0.0889 ± 0.0003**   | **0.0278 ± 0.0003**   |
> | DRL-NBL            | 95.42% ± 1.05%       | 89.75% ± 0.57%        | 95.38% ± 4.22%        | 92.33% ± 1.90%        | 0.0907 ± 0.0012       | 0.0287 ± 0.0006       |

---

> ### Author Response · Authors · 2025-11-29
>
> >**Question 4:** The hand-crafted discrepancy term between state errors makes the formulation harder to follow. It may read more clearly if it were framed as a variance-of-risk penalty or connected to a Bayesian prior interpretation, rather than appearing as an ad-hoc L1 difference. Even a short explanation linking it to those ideas would help readers understand its purpose and statistical grounding.
>
> **Response to Question 4:** We thank the reviewer for this valuable suggestion. We have incorporated the more detailed explanations into the revision and provide more detailed explanations of each reward component to help readers better understand their roles and connections to established constructs. **(Page 6)**
>
> Specifically:
> "The immediate reward at each time step $t$ is designed from a local perspective, consisting of the prediction gain and the action switching penalty, and is defined as:
>
> $r\_{i,t}^{\text{IR}} =\lambda\_1  \left[\alpha \left(e^{\text{base}}\_{i,t+1} - e^{(a\_{i,t})}\_{i,t+1}\right)+ (1-\alpha) \left(e^{\text{base}}\_{i,t} - e^{(a\_{i,t})}\_{i,t}\right)\right]+\lambda\_2 \frac{\max\{0,\rho\_c - c_{i,t}\}}{\rho\_c - 1}.$
>
> where $e^{(a\_{i,t})}\_{i,t}$ denotes the MSE of the prediction obtained by the prediction module based on action $a_{i,t}$, $e^{\text{base}}\_{i}$ denotes the baseline MSE obtained from a predictor with a structure similar to the prediction module but without considering hidden states, $\alpha \in [0,1]$ balances the emphasis between the accuracy of $\hat{s}\_{t}$ and $\hat{s}\_{t+1}$, $c\_{i,t}$ is the length of consecutive identical actions, $\rho\_c$ is the continuity threshold, $\lambda_1$ and $\lambda_2$ are the hyperparameters control the relative influence of the prediction gain term and the action continuity penalty term. Here, $e^{\text{base}} - e^{(a)}$ an advantage-style signal that measures the relative predictive gain of the selected state, encouraging the agent to focus on state-specific improvements rather than absolute errors. The action switching penalty term discourages only spurious rapid switching and serves mainly as a variance-reduction mechanism, without enforcing persistent states or restricting genuine hidden state changes."
>
> " At the end of an episode, an episodic reward is introduced from a global perspective to evaluate the overall quality of state estimation, consisting of the state separation objective and the pairwise discrepancy, which is defined as:
>
> $r^{\text{ER}}\_i = -\frac{1}{m}\sum\_{a\_i=1}^{m} \overline{e}^{(a\_i|a\_i)}\_i + \sum^{m}\_{a_i=1}\left[\max\{\Delta\_i^{(a\_i)},0\}+\lambda\_3\min\{\Delta\_i^{(a\_i)},0\}\right]- \lambda\_4 \frac{1}{2}\sum\_{p\neq q} \big|\overline{e}^{(p|p)}\_i - \overline{e}^{(q|q)}\_i \big|$
>
> where $\overline{e}^{(u|v)}\_i$ denotes the mean of conditional MSE, i.e., $\overline{e}^{(u|v)}\_i = (\sum\_{t=1}^{T\_{E}}\sum\_{s=1}^{m}z\_{t}^{(s|v)}e\_{t}^{(u)})/(\sum\_{t=1}^{T\_{E}}\sum\_{s=1}^{m}z\_{t}^{(s|v)})$. Here, $z_{t}^{(s|v)}=1$ if $s=v$, and 0 otherwise. $\Delta\_i^{(a\_i)}=\bar{e}^{(-a\_i|-a\_i)}\_i-\bar{e}^{(a\_i|a\_i)}\_i$ measures the performance advantage of the selected action over the unselected ones, where $\overline{e}^{(a\_i|a\_i)}\_i$ denotes the mean MSE of the predictions under the selected action $a\_i$, while $\bar{e}^{(-a\_i|-a\_i)}\_{i}$ denotes the mean MSE under the other actions. $\frac{1}{2}\sum\_{p\neq q} \big|\bar{e}^{(p|p)}\_i - \bar{e}^{(q|q)}\_i \big|$ measures the pairwise discrepancy in accuracy across different actions, $\lambda\_3$ and $\lambda\_4$ are the hyperparameters. The state separation term resembles specialization objectives in Mixture-of-Experts models. It evaluates within-state accuracy and across-state separation at the end of each episode, encouraging each state to fit its assigned samples while maintaining a clear performance margin over alternative states. The pairwise discrepancy term quantifies accuracy gaps across hidden states and thereby directs more attention to under-trained hidden states, preventing neglect and promoting more even learning across the predictors."
>
> "In summary, the immediate reward shapes step-wise state selection through prediction improvement and controlled switching behavior, while the episodic reward enforces hidden state consistency, balanced state quality, and meaningful specialization within our distribution-free RL formulation."

---

> ### Author Response · Authors · 2025-11-29
>
> >**Question 5:** The overall model feels large and hard to scale. Just example, if the data were a $N = (1000\times1000,T)$ image sequences, would the current setup still be feasible (for example take a look to experiments of [2] and [5])? Why not use shared backbones or partially shared modules instead of per-variable networks? An ablation or discussion on model sharing and scalability would strengthen the paper.
>
> **Response to Question 5:** We thank the reviewer for highlighting this point. We use per-variable networks because we assume the variables in our applications exhibit heterogeneous regime dynamics, and enforcing a shared backbone implicitly assumes homogeneous transition patterns. In our framework, Stage-1 models run fully in parallel, so the cost scales linearly with $N$. When variables do share similar regime patterns, DRL-STAF supports multi-dimensional emission outputs, enabling partial parameter sharing without modifying the state-estimation architecture.
>
> To further validate our design choice, we implemented two shared-backbone variants:
> (1) per-variable DEMs with a shared policy $\pi_{i,\theta_A}$, and
> (2) a fully shared DEM backbone (with variable-specific state-conditioned output heads) together with a shared $\pi_{i,\theta_A}$.
>
> Due to time constraints, we evaluated these variants on our 3-variable simulated dataset. Both failed to learn meaningful regimes, confirming that shared-backbone strategies are not feasible in our setting.
>
> **Table 1: Comparison results on the simulated dataset with 3 variables**
> | Models        | Accuracy         | Precision        | Recall           | F1               | MAE               | MSE               |
> |---------------|------------------|------------------|------------------|------------------|--------------------|--------------------|
> | DRL-Shared1   | 67.57% ± 0.00%   | 19.00% ± 0.00%   | 33.33% ± 0.00%   | 24.20% ± 0.00%   | 0.4083 ± 0.0360    | 0.4233 ± 0.0775    |
> | DRL-Shared2   | 67.57% ± 0.00%   | 19.00% ± 0.00%   | 33.33% ± 0.00%   | 24.20% ± 0.00%   | 0.2096 ± 0.0002    | 0.1578 ± 0.0001    |
> |**DRL-STAF**     | **98.17% ± 0.07%**   | **96.89% ± 0.19%**   | **99.62% ± 0.25%**   | **98.22% ± 0.07%**   | **0.0889 ± 0.0003**    | **0.0278 ± 0.0003**    |
>
> >**Question 6:** Please include a complexity analysis for both training and inference. Even if a full theoretical analysis is difficult, an empirical comparison (e.g., wall-clock time, FLOPs, or memory usage) would help. Using variable-dependent separate networks is clearly expensive, so checking this overhead would make the paper more transparent. (for example take a look to table 1 in SALT [1] or table 6 in GIN [2])
>
> **Response to Question 6:** We thank the reviewer for the suggestion. As noted in the response to Weakness 5, we have added empirical computational analysis wall-clock times for training and inference, and reported the information about the hardware used. **(Page 21)**
>
> >**Question 7:** The gradient computation path is not entirely clear. Since the model involves discrete sampling (e.g., argmax during hard decoding), it is not obvious how gradients are propagated in theory (e.g. equation 7-8 in [4]) . The many hand-crafted components make this kinda harder to trace and could introduce stability issues. Did you encounter such problems in experiments, and if so, could you mention them (e.g. in your table of results adding a column and report the success percentage over your random seeds run.) and describe any remedies or tricks used to stabilize training?
>
> **Response to Question 7:** We thank the reviewer for the question. Gradients flow only through the DRL actor–critic pipeline; discrete state selections do not require differentiability because actions are sampled from a categorical distribution, as is standard in policy-gradient RL.
>
> To ensure robustness during training, we employ only standard DRL stabilization mechanisms—such as clipping, entropy regularization, and reward normalization—which are routinely used in policy-gradient methods. The detailed explanations have been provided in the revised manuscript **(Page 22)**.
>
> As with most deep RL and deep learning systems, training performance inevitably exhibits variability across random seeds. In practice, we mitigate this by initializing new policies from previously trained policies to accelerate convergence and improve stability. This is an approach commonly used in RL training and evaluation. All the reported results in the paper are averaged results, and we have added the corresponding standard deviations in the revised version to more transparently reflect training stability. **(Pages 18-20)**

---

> ### Author Response · Authors · 2025-11-29
>
> >**Question 8:** The classical HMM baselines (even neural versions with parametric emissions or transitions) are relatively weak comparisons. I would suggest including more advanced smooth or continuous state-estimation baselines such as neural state-space models, switching Kalman filters, or variational sequence models to provide a stronger and fairer benchmark.
>
> **Response to Question 8:** We thank the reviewer for the valuable suggestion. As detailed in our response to Weakness 6, we have added DL-HMM hybrid baselines, including NCTRL [4] and NHMM [6], and evaluated them under exactly the same protocol as DRL-STAF. Representative results are reported in Table 5, and comprehensive results are provided in the revised manuscript. (Pages 9–10 and 18–20)
>
> **A few minor issues:**
>
> >**Question 9:** In line 42 "three key limitations ..."
>
> **Response to Question 9:** We thank the reviewer for pointing this out. We have revised the wording in line 42 for clarity.
>
> >**Question 10:** In lines 386-389, better to mention specifically DEM$_i$ are trained based on the updated probabilities in second stage. A little confusing.
>
> **Response to Question 10:** We thank the reviewer for the suggestion. We will revise the corresponding lines to explicitly state that DEM$_i$ are updated using the refined state probabilities produced in Stage-2. **(Page 9)**
>
> >**Question 11:** Despite your model, many of the baselines see improvement, at least in accuracy, when variable size increases. Making sense as model uses correlational info for richer state inference, but what do you think about this? Also would be nice a short sentence being added about this.
>
> **Response to Question 11:** We appreciate the reviewer’s observation. Theoretically, incorporating more correlated variables provides richer contextual information for state inference, allowing both our method and the baselines to achieve higher accuracy. We will add a short discussion explaining this trend in the revision.
>
> >**Question 12:** If you have an ablation only on second stage policy network without mean-field style behavior in first stage, please report it (i.e. directly optimized your policy when variables interaction is modeled and first stage policy learning is dropped).
>
> **Response to Question 12:** Thank you for the suggestion. We have previously conducted experiments that are closely related to this ablation. Due to the limited time, we report here only the results on our 3-variable simulated dataset (DRL-NSO is the model without stage one). When the first-stage per-variable policy learning is removed and the second-stage policy is directly optimized with cross-variable interactions, the model fails entirely.
>
> **Table 1: Comparison results on the simulated dataset with 3 variables**
> | Models    | Accuracy           | Precision           | Recall             | F1                 | MAE                 | MSE                 |
> |-----------|--------------------|----------------------|---------------------|---------------------|----------------------|----------------------|
> | **DRL-STAF** | **98.17% ± 0.07%** | **96.89% ± 0.19%**   | **99.62% ± 0.25%**  | **98.22% ± 0.07%**  | **0.0889 ± 0.0003**  | **0.0278 ± 0.0003**  |
> | DRL-NSO  | 66.48% ± 1.53% | 24.93% ± 8.39%   | 33.64% ± 0.43%  | 26.60% ± 3.39%  | 0.5015 ± 0.0687  | 0.9767 ± 0.0964  |
>
> This outcome is expected. Modeling variable interactions from the start effectively amplifies estimation errors: in early training, an incorrect state prediction for one variable immediately propagates to all other variables through the interaction module, which in turn corrupts the training signal for the forecasting networks. Without the stabilizing effect of the first-stage per-variable policies, the entire learning process collapses.

---

### Official Review · Reviewer_xt2e · 2025-10-23

**Soundness:** 3
**Presentation:** 2
**Contribution:** 3
**Rating:** 6
**Confidence:** 4

**Summary:**

This paper proposes DRL-STAF to capture the complex nonlinear observation patterns in Multivariate hidden Markov processes. In the proposed framework, deep learning is used as the emission function to capture complex nonlinear observation patterns, while deep reinforcement learning models state transitions, supporting flexible adaptation to diverse transition patterns without predefined structural assumptions.  Experiments show the effectiveness of the proposed framework.

**Strengths:**

1. The paper is overall well-organized and easy to follow.
2. The proposed framework, DRL-STAF, is intuitive and is supported by several theoretical and experimental results.
3. The experiment of predicting exchange rates is interesting in the field of reinforcement learning.

**Weaknesses:**

1. Although hidden Markov models are introduced, multivariate hidden Markov decision processes are not sufficiently introduced in the introduction. Relevant literature and intuitive examples are not provided.
2. The proposed method should be connected to the model-based RL method, where a model is trained to predict the future states and rewards in MDPs or POMDPs.
3.  The authors should provide a detailed introduction of the reward in the experimental part.
4. The text in some images, such as Figure 1, is too small to read.

**Questions:**

1. What is the difference between the Multivariate hidden Markov process and the partially observable Markov process?
2. Can RL methods for the partially observable Markov process be used in solving the Multivariate hidden Markov process?
3. Could you discuss the computational cost and scalability of the proposed method in detail, especially as the sequence length increases?

---

> ### Author Response · Authors · 2025-11-29
>
> Thank you very much for your valuable suggestions. We have provided point-by-point responses to address all the raised questions and concerns. The rebuttal revision has also been submitted. All changes in the revised rebuttal manuscript have been highlighted in **red** for clarity.
>
> >**Weakness 1:** Although hidden Markov models are introduced, multivariate hidden Markov decision processes are not sufficiently introduced in the introduction. Relevant literature and intuitive examples are not provided.
>
> **Response to Weakness 1:** We thank the reviewer for the helpful suggestion. In the revised manuscript, we have added explanations for multivariate hidden Markov processes to better contextualize the modeling background and highlight the motivation for DRL-STAF. **（Page 1）**
>
> >**Weakness 2:** The proposed method should be connected to the model-based RL method, where a model is trained to predict the future states and rewards in MDPs or POMDPs.
>
> **Response to Weakness 2:** We thank the reviewer for the helpful suggestion. In model-based RL, the agent explicitly learns a transition model $p(o\_{t+1}|o\_{t},a\_{t})$ ($o$ is the state in DRL, $a$ is the action in DRL). In our multivariate state-aware forecasting setting, adopting a model-based approach would **require learning** **the hidden state transition dynamics** $p(s\_{t+1}|s\_{t-k+1:t},d\_{t})$ ($s$ is the hidden state in HMM, $d$ is the duration of the current state) as well as **the duration dynamics**, **cross-sequence interactions among hidden states**, **the full emission generation process**, and so on. Therefore, **the environment model is extremely complex**. Moreover, such a design also **suffers from combinatorial joint-state explosion**: with $N$ variables and $m$ states per variable, the joint latent space has size $m^{N}$, making the transition model $m^{N}\times m^{N}$.
>
> More importantly, even if such components were available, model-based RL cannot directly address the problem we consider. Our task involves two tightly coupled components that differ fundamentally from standard POMDP modeling:
>
> (1) Forecasting is a discriminative regression task, requiring a high-capacity mapping $(s\_{t},X\_{1:t})\rightarrow \hat{x}\_{t+1}$. This differs from model-based RL, which relies on a generative observation model $p(x_{t+1}|s_{t+1})$ for likelihood maximization or planning.
>
> (2) The hidden state $s_t$ is a decision variable, not an environment variable. Selecting $s_t$ does not alter how the real sequence evolves, instead, it shapes the prediction module and thereby the future rewards. This creates a feedback loop between state estimation and forecasting that has no analogue in model-based RL, where environment dynamics are external and independent of the agent.
>
> Because of this coupling, a transition model over these hidden regimes is neither well-defined nor useful in our setting. The hidden regimes do not follow a controlled Markov decision process, and their transitions are not determined by the agent’s actions in any physical sense. Any learned “transition model’’ would therefore be a regression artifact rather than a valid environment model for planning or rollouts.
>
> In summary, model-based RL is not suitable for the multivariate state-aware forecasting setting. Motivated by these limitations, we propose DRL-STAF, the first distribution-free framework for multivariate hidden Markov processes, where model-free DRL policies directly estimate discrete hidden states while deep networks model emissions.

---

> ### Author Response · Authors · 2025-11-29
>
> >**Weakness 3:** The authors should provide a detailed introduction of the reward in the experimental part.
>
> **Response to Weakness 3:** We thank the reviewer for the helpful suggestion. In the revised manuscript, we have expanded the experimental section with additional ablation studies that remove the individual reward components to empirically demonstrate the necessity and effectiveness of our reward design. Specifically, we introduce the following:
> * DRL-NASP (without the action switching penalty),
> * DRL-NSSS (without the sample screening strategy),
> * DRL-NASP&SSS (without both the action switching penalty and sample screening),
> * DRL-NBL (without the baseline error term $e^{base}$),
> * DRL-NSSE (without the state separation evaluation term),
> * DRL-NPDE (without the pairwise discrepancy evaluation term),
> * DRL-NER (without the episodic reward),
> * DRL-NSO (removing stage one and directly modeling cross-variable interactions).
>
> The ablation results show that the necessity of each component in DRL-STAF. Removing the sample screening strategy (DRL-NSSS), the episodic reward (DRL-NER), or the training of Stage one (DRL-NSO) produces the most severe degradation, causing the model to lose reliable state estimation and forecasting accuracy. In contrast, removing other components, such as the action switching penalty (DRL-NASP), baseline error term (DRL-NBL), state separation evaluation (DRL-NSSE), or pairwise discrepancy evaluation (DRL-NPDE), primarily affects stability and boundary sharpness, resulting in reduced but still operational performance. **（Page 20）**
>
> **Table 1: Comparison results of ablation models on simulated datasets with 3 variables**
> | Models        | Accuracy             | Precision            | Recall               | F1                   | MAE                  | MSE                  |
> |---------------|----------------------|-----------------------|-----------------------|-----------------------|-----------------------|-----------------------|
> | **DRL-STAF**       | **98.17% ± 0.07%**   | **96.89% ± 0.19%**    | **99.62% ± 0.25%**    | **98.22% ± 0.07%**    | **0.0889 ± 0.0003**   | **0.0278 ± 0.0003**   |
> | DRL-NASP           | 96.21% ± 0.22%       | 89.71% ± 0.34%        | 98.42% ± 0.26%        | 93.69% ± 0.30%        | 0.1286 ± 0.0004       | 0.0499 ± 0.0022       |
> | DRL-NSSS           | 68.99% ± 0.19%       | 50.38% ± 0.24%        | 66.50% ± 0.05%        | 56.71% ± 0.13%        | 0.2307 ± 0.0008       | 0.1572 ± 0.0005       |
> | DRL-NASP&SSS       | 58.69% ± 0.91%       | 62.79% ± 1.32%        | 58.64% ± 6.69%        | 59.78% ± 3.24%        | 0.2574 ± 0.0004       | 0.1800 ± 0.0004       |
> | DRL-NBL            | 95.42% ± 1.05%       | 89.75% ± 0.57%        | 95.38% ± 4.22%        | 92.33% ± 1.90%        | 0.0907 ± 0.0012       | 0.0287 ± 0.0006       |
> | DRL-NSSE           | 93.27% ± 0.36%       | 88.58% ± 0.77%        | 93.15% ± 1.60%        | 90.43% ± 0.42%        | 0.0965 ± 0.0015       | 0.0426 ± 0.0012       |
> | DRL-NPDE           | 95.46% ± 1.59%       | 94.05% ± 2.44%        | 90.06% ± 7.30%        | 91.65% ± 3.31%        | 0.0932 ± 0.0032       | 0.0294 ± 0.0013       |
> | DRL-NER            | 67.57% ± 0.00%       | 19.00% ± 0.00%        | 33.33% ± 0.00%        | 24.20% ± 0.00%        | 0.2099 ± 0.0000       | 0.1590 ± 0.0000       |
> | DRL-NSO            | 66.48% ± 1.53%       | 24.93% ± 8.39%        | 33.64% ± 0.43%        | 26.60% ± 3.39%        | 0.5015 ± 0.0687       | 0.9767 ± 0.0964       |
>
> >**Weakness 4**: The text in some images, such as Figure 1, is too small to read.
>
> **Response to Weakness 4:** We thank the reviewer for pointing this out. In the revised manuscript, we have enlarged the font size and improved the readability of all figures, including Figure 1.

---

> ### Author Response · Authors · 2025-11-29
>
> >**Question 1**: What is the difference between the Multivariate hidden Markov process and the partially observable Markov process?
>
> **Response to Question 1:** There is no partially observable Markov process, but only partially observable Markov Decision Process (POMDPs).
>
> (1) Multivariate Hidden Markov Process (MHMP)
>
> An MHMP is a generative model for multivariate sequences.
> A hidden state $s_t$ evolves according to a Markov chain and generates observations $x_t$ through an emission distribution. There are: no actions, no rewards, no policies, and no decision-making components. Its goal is to explain how a multivariate time series is generated.
>
> (2) Partially Observable Markov Decision Process (POMDP)
>
> A POMDP is a decision-making framework under partial observability. An agent chooses actions, the environment transitions to a new state, and a reward is returned. Observations are only an indirect view of the true state. Thus, a POMDP explicitly involves: actions chosen by the agent, a transition model, rewards, and a policy maximizing cumulative return. Its goal is optimal control, not sequence generation.
>
> In summary, MHMP describes how data is generated, and POMDP describes how an agent acts under uncertainty. The two notions address different problems and are not interchangeable.
>
> >**Question 2**: Can RL methods for the partially observable Markov process be used in solving the Multivariate hidden Markov process?
>
> **Response to Question 2:** We thank the reviewer for the insightful question. **The core novelty of our work is precisely to formulate state-aware forecasting of multivariate hidden Markov processes as a POMDP and to design a DRL-driven framework that simultaneously performs time-series forecasting and hidden-state inference.** This is not trivial because existing RL methods for POMDPs cannot infer discrete hidden states while simultaneously predicting future observations. Although these methods maintain a belief state, this belief is not equivalent to the discrete hidden states in the HMM and therefore cannot support hidden state estimation. More importantly, as clarified in our Response to Weakness 2, the hidden state in DRL-STAF is a decision variable rather than a part of the environment dynamics. Selecting $s_t$ does not change how the real sequence evolves; instead, it reshapes the predictor and thus the future rewards. This creates a feedback loop between state estimation and forecasting that has no analogue in POMDP RL, where environment dynamics are external and fixed. Because of this coupling, learning a transition model over these hidden regimes is neither well-defined nor useful, since their transitions are not governed by an underlying controlled Markov process. In summary, **RL methods for POMDP cannot be directly applied to multivariate hidden Markov processes**, as their goals and feedback mechanisms are fundamentally different.

---

> ### Author Response · Authors · 2025-11-29
>
> >**Question 3**: Could you discuss the computational cost and scalability of the proposed method in detail, especially as the sequence length increases?
>
> **Response to Question 3** We thank the reviewer for the helpful suggestion. Consider $N$ variables，each associated with $m$ hidden states. In conventional multivariate HMMs, each hidden state adopts a Gaussian mixture emission model with $C$ components. For higher-order HMMs, we denote the Markov order by $R$. For HSMMs, we use $D$ to denote the maximum duration truncation. For DL methods, we use $h$ to denote the hidden dimension of neural networks. The sequence length is written as $T$. Specially, DRL-STAF is trained on fixed-length episodes with random starting points of size $L$, making its per-iteration complexity independent of $T$. All parameter counts and complexities reported in Table 4 are expressed using these quantities. The advantage becomes clear when comparing parameter complexities between CHMM and DRL-STAF. CHMM requires modeling the joint latent space with $O(m^{2N})$ parameters, which grows exponentially with the number of variables. In contrast, DRL-STAF only has $O(N(mh^2))$ parameters, growing linearly in $N$. Since DRL-STAF never enumerates joint state combinations, it avoids the combinatorial explosion inherent to CHMMs while still capturing cross-variable dependencies.
>
> **Table 2: Comparison of number of parameters for the methods we considered. "Yes" means the model explicitly encodes dependencies between different variables. "No" means variables are modeled by independent chains.**
> | Models         | Cross-variable interactions | Parameter Complexity        | Computational Complexity (per sequence of length $T$) |
> |----------------|------------------------------|------------------------------|---------------------------------------------------------|
> | Parallel HMM   | No                           | $O(N(m^2+mC))$               | $O(N T (m^2 + mC))$                                     |
> | Parallel HSMM  | No                           | $O(N(m^2+mC))$               | $O(N T m^2 D)$                                          |
> | Parallel HOHMM | No                           | $O(Nm^{R+1})$                | $O(N T m^{2R})$                                         |
> | CHMM           | Yes                          | $O(m^{2N})$                  | $O(T m^{2N})$                                           |
> | NHMM           | No                           | $O(N(h^2+hm^2))$             | $O(N T (h^2 + h m^2))$                                  |
> | NCTRL          | No                           | $O(N(h^3+m^2))$              | $O(N T (h^3 + m^2))$                                    |
> | Markovian-RNN  | No                           | $O(N(mh^2+m^2))$             | $O(N T (m h^2 + m^2))$                                  |
> | DEN-HMM        | No                           | $O(N(h^2+m^2))$              | $O(N T (h^2 + m^2))$                                    |
> | DRL-STAF   | Yes                      | $O(N(mh^2))$             | $O(N L (mh^2))$                                     |

---

> ### Author Response · Authors · 2025-11-29
>
> >**Question 3**: Could you discuss the computational cost and scalability of the proposed method in detail, especially as the sequence length increases?
>
> To make the computational cost concrete, we further measured wall-clock performance on a machine with an Intel(R) Core(TM) Ultra 5 125H CPU. We report training time and per-step inference latency for all single-variable models, and for multivariate settings we additionally report results under different values of $N$. It is shown that DRL-STAF requires longer training time, its per-step inference is fast, making it suitable for online and real-time applications. The longer training time mainly comes from the RL-based state estimator, which must iteratively learn regime assignments together with forecasting behavior. In practice, this cost can be reduced by fine-tuning a pretrained model on new periods or related systems, making deployment considerably faster. Overall, the initial training cost is acceptable given the real-time inference efficiency. We include the computational analysis and experimental results in the revised manuscript. **（Page 21）**
>
> **Table 3: Computational cost of single-variable models (time in seconds)**
> | Single-variable settings | Training   | Per-step inference |
> |--------------------------|------------|---------------------|
> | Parallel HMM             | 13.805     | 0.002               |
> | Parallel HSMM            | 595.557    | 14.200              |
> | Parallel HOHMM           | 33.753     | 0.005               |
> | NHMM                     | 214.061    | 0.003               |
> | NCTRL                    | 738.380    | 0.055               |
> | Markovian-RNN            | 2972.960   | 0.008                |
> | DEN-HMM                  | 6465.399   | 0.001               |
> | DRL-STAF (Stage one) | 5443.833 | 0.001           |
>
> **Table 4: Computational cost of multivariate models (time in seconds)**
> | Models | 3-var Training | 3-var Inference | 5-var Training | 5-var Inference | 10-var Training | 10-var Inference |
> |-----------------------|----------------|------------------|----------------|------------------|------------------|-------------------|
> | CHMM                  | 77.760         | 0.014            | 158.866       | 0.030     | 156588.235  |  2.531 |
> | DRL-STAF              | 47468.344      | 0.006            | 74792.616   |  0.011    |  119733.858   |   0.018    |

---

### Official Review · Reviewer_huyU · 2025-10-26

**Soundness:** 3
**Presentation:** 3
**Contribution:** 3
**Rating:** 6
**Confidence:** 4

**Summary:**

This paper introduces DRL-STAF, a novel multivariate time series framework by combining HMM-like state space model with deep RL. First, the prediction module (a DEN-HMM where each head is responsible for each state) utilizes a Deep Emission Network to model state-dependent patterns. Second, a state estimation problem is framed as a Deep RL problem with two-stage training. Experiments are extensive and results are promising.

**Strengths:**

- The problem of multivariate time series forecasting is long-standing and challenging.
- The core idea of estimating states as a deep RL problem where the policy is a transition function is novel to my knowledge.
- The methodology is technically sound and sophisticated.
- Presentation and figures are clear, well-structured and well-written.

**Weaknesses:**

- Assumption of infrequent state transitions is strong and may not hold true in practice.
- No computational analysis in terms of training and inference cost are provided. I am curious to see if this framework can be applied for online, real-time streaming data.
- Comparisons with other DL-based solutions such as TFT[1] for multivariate forecasting should be considered in experiments.
[1] Lim et.al., Temporal Fusion Transformers for interpretable multi-horizon time series forecasting, International Journal of Forecasting, 2021

**Questions:**

- Can this framework used in multistep (long-horizon) forecasting?

---

> ### Author Response · Authors · 2025-11-29
>
> Thank you very much for your valuable suggestions. We have provided point-by-point responses to address all the raised questions and concerns. The rebuttal revision has also been submitted. All changes in the revised rebuttal manuscript have been highlighted in red for clarity.
>
> >**Weakness 1:** Assumption of infrequent state transitions is strong and may not hold true in practice.
>
> **Response to Weakness 1:** We thank the reviewer for raising this important point. Although DRL-STAF includes a penalty on rapid state switches, this does not impose a strict or unrealistic assumption that real data must exhibit infrequent transitions. The action continuity penalty term (renamed as action switching penalty) is introduced as a variance-reduction mechanism during early training, when the RL policy behaves nearly randomly and the reward signal is extremely noisy. Under such uncertainty, assuming $s_{t+1}=s_{t}$ serves as a minimax-stable default decision, preventing spurious rapid switching that would destabilize both state estimation and emission learning.
>
> To validate our claim, we conducted a set of targeted experiments designed around the reviewer’s concerns. Specifically, we evaluated:
>
> **(1) Fast-switching simulated datasets**, to stress-test the model under frequent regime changes; and
>
> **(2) A set of ablation models**, including DRL-NASP (removing the action switching penalty), DRL-NSSS (removing the sample screening strategy), and DRL-NASP&SSS (removing both), to evaluate how removing these stabilizing components affects the outcomes.
>
> **Table 1: Fast-switching simulated dataset 1**
>
> | Models         | Accuracy               | Precision             | Recall                | F1                    | MAE                   | MSE                   |
> |----------------|-------------------------|------------------------|------------------------|------------------------|------------------------|------------------------|
> | Parallel HMM   | 63.37% ± 0.92%         | 45.33% ± 1.27%         | 55.95% ± 1.26%         | 49.75% ± 1.06%         | 0.2564 ± 0.0042        | 0.1183 ± 0.0043        |
> | Parallel HSMM  | 63.80% ± 0.85%         | 23.57% ± 1.37%         | 33.33% ± 1.22%         | 27.61% ± 1.13%         | 0.2771 ± 0.0043        | 0.1393 ± 0.0047        |
> | Parallel HOHMM | 57.87% ± 0.93%         | 48.23% ± 1.35%         | 47.69% ± 1.39%         | 46.37% ± 1.19%         | 0.2133 ± 0.0036        | 0.0847 ± 0.0034        |
> | CHMM           | 62.23% ± 0.93%         | 58.07% ± 1.11%         | 87.93% ± 0.85%         | 68.17% ± 0.93%         | 0.2724 ± 0.0048        | 0.1474 ± 0.0058        |
> | NHMM           | 62.40% ± 2.54%         | 45.49% ± 21.11%        | 43.42% ± 8.24%         | 36.85% ± 7.68%         | 0.2271 ± 0.0023        | 0.1009 ± 0.0012        |
> | NCTRL          | 80.93% ± 5.77%         | 83.23% ± 5.94%         | 78.59% ± 10.06%        | 79.37% ± 6.21%         | 0.2439 ± 0.0044        | 0.1137 ± 0.0021        |
> | Markovian-RNN  | 61.30% ± 2.20%         | 51.19% ± 3.06%         | 65.81% ± 5.40%         | 57.24% ± 4.03%         | 0.1338 ± 0.0032        | 0.0430 ± 0.0017        |
> | DEN-HMM        | 61.47% ± 2.33%         | 29.56% ± 5.54%         | 41.18% ± 9.31%         | 34.40% ± 7.00%         | 0.2861 ± 0.0049        | 0.1480 ± 0.0051        |
> | **DRL-STAF**   | **89.77% ± 0.11%**     | **88.47% ± 1.42%**     | **89.04% ± 2.28%**     | **88.66% ± 0.45%**     | **0.1070 ± 0.0002**    | **0.0367 ± 0.0001**    |
>
> **Table 2: Comparison results of ablation models on simulated datasets with 3 variables**
>
> | Models         | Accuracy               | Precision             | Recall                | F1                    | MAE                   | MSE                   |
> |----------------|-------------------------|------------------------|------------------------|------------------------|------------------------|------------------------|
> | **DRL-STAF**        | **98.17% ± 0.07%**     | **96.89% ± 0.19%**     | **99.62% ± 0.25%**     | **98.22% ± 0.07%**     | **0.0889 ± 0.0003**    | **0.0278 ± 0.0003**    |
> | DRL-NASP           | 96.21% ± 0.22%         | 89.71% ± 0.34%         | 98.42% ± 0.26%         | 93.69% ± 0.30%         | 0.1286 ± 0.0004         | 0.0499 ± 0.0022         |
> | DRL-NSSS           | 68.99% ± 0.19%         | 50.38% ± 0.24%         | 66.50% ± 0.05%         | 56.71% ± 0.13%         | 0.2307 ± 0.0008         | 0.1572 ± 0.0005         |
> | DRL-NASP&SSS       | 58.69% ± 0.91%         | 62.79% ± 1.32%         | 58.64% ± 6.69%         | 59.78% ± 3.24%         | 0.2574 ± 0.0004         | 0.1800 ± 0.0004         |
>
> Our experiments show that DRL-STAF remains effective on fast-switching simulated datasets, indicating that the framework does not rely on persistent regimes. In contrast, removing the stabilizing components leads to unstable oscillations and significantly degraded regime separation. We include these results and clarifications in the revision. **（Pages 19-20）**

---

> ### Author Response · Authors · 2025-11-29
>
> >**Weakness 2:** No computational analysis in terms of training and inference cost are provided. I am curious to see if this framework can be applied for online, real-time streaming data.
>
> **Response to Weakness 2:**  We thank the reviewer for the helpful suggestion. Consider $N$ variables，each associated with $m$ hidden states. In conventional multivariate HMMs, each hidden state adopts a Gaussian mixture emission model with $C$ components. For higher-order HMMs, we denote the Markov order by $R$. For HSMMs, we use $D$ to denote the maximum duration truncation. For DL methods, we use $h$ to denote the hidden dimension of neural networks. The sequence length is written as $T$. Specially, DRL-STAF is trained on fixed-length episodes with random starting points of size $L$, making its per-iteration complexity independent of $T$. All parameter counts and complexities reported in Table 4 are expressed using these quantities. The advantage becomes clear when comparing parameter complexities between CHMM and DRL-STAF. CHMM requires modeling the joint latent space with $O(m^{2N})$ parameters, which grows exponentially with the number of variables. In contrast, DRL-STAF only has $O(N(mh^2))$ parameters, growing linearly in $N$. Since DRL-STAF never enumerates joint state combinations, it avoids the combinatorial explosion inherent to CHMMs while still capturing cross-variable dependencies.
>
> **Table 3: Comparison of number of parameters for the methods we considered. "Yes" means the model explicitly encodes dependencies between different variables. "No" means variables are modeled by independent chains.**
> | Models         | Cross-variable interactions | Parameter Complexity        | Computational Complexity (per sequence of length $T$) |
> |----------------|------------------------------|------------------------------|---------------------------------------------------------|
> | Parallel HMM   | No                           | $O(N(m^2+mC))$               | $O(N T (m^2 + mC))$                                     |
> | Parallel HSMM  | No                           | $O(N(m^2+mC))$               | $O(N T m^2 D)$                                          |
> | Parallel HOHMM | No                           | $O(Nm^{R+1})$                | $O(N T m^{2R})$                                         |
> | CHMM           | Yes                          | $O(m^{2N})$                  | $O(T m^{2N})$                                           |
> | NHMM           | No                           | $O(N(h^2+hm^2))$             | $O(N T (h^2 + h m^2))$                                  |
> | NCTRL          | No                           | $O(N(h^3+m^2))$              | $O(N T (h^3 + m^2))$                                    |
> | Markovian-RNN  | No                           | $O(N(mh^2+m^2))$             | $O(N T (m h^2 + m^2))$                                  |
> | DEN-HMM        | No                           | $O(N(h^2+m^2))$              | $O(N T (h^2 + m^2))$                                    |
> | DRL-STAF   | Yes                      | $O(N(mh^2))$             | $O(N L (mh^2))$                                     |

---

> ### Author Response · Authors · 2025-11-29
>
> >**Weakness 2:** No computational analysis in terms of training and inference cost are provided. I am curious to see if this framework can be applied for online, real-time streaming data.
>
> To make the computational cost concrete, we further measured wall-clock performance on a machine with an Intel(R) Core(TM) Ultra 5 125H CPU. We report training time and per-step inference latency for all single-variable models, and for multivariate settings we additionally report results under different values of $N$. It is shown that DRL-STAF requires longer training time, its per-step inference is fast, making it suitable for online and real-time applications. The longer training time mainly comes from the RL-based state estimator, which must iteratively learn regime assignments together with forecasting behavior. In practice, this cost can be reduced by fine-tuning a pretrained model on new periods or related systems, making deployment considerably faster. Overall, the initial training cost is acceptable given the real-time inference efficiency. We include the computational analysis and experimental results in the revised manuscript. **（Page 21）**
>
> **Table 4: Computational cost of single-variable models (time in seconds)**
> | Single-variable settings | Training   | Per-step inference |
> |--------------------------|------------|---------------------|
> | Parallel HMM             | 13.805     | 0.002               |
> | Parallel HSMM            | 595.557    | 14.200              |
> | Parallel HOHMM           | 33.753     | 0.005               |
> | NHMM                     | 214.061    | 0.003               |
> | NCTRL                    | 738.380    | 0.055               |
> | Markovian-RNN            | 2972.960   | 0.008                |
> | DEN-HMM                  | 6465.399   | 0.001               |
> | DRL-STAF (Stage one) | 5443.833 | 0.001           |
>
> **Table 6: Computational cost of multivariate models (time in seconds)**
> | Models | 3-var Training | 3-var Inference | 5-var Training | 5-var Inference | 10-var Training | 10-var Inference |
> |-----------------------|----------------|------------------|----------------|------------------|------------------|-------------------|
> | CHMM                  | 77.760         | 0.014            | 158.866       | 0.030     | 156588.235  |  2.531 |
> | DRL-STAF              | 47468.344      | 0.006            | 74792.616   |  0.011    |  119733.858   |   0.018    |

---

> ### Author Response · Authors · 2025-11-29
>
> >**Weakness 3:** Comparisons with other DL-based solutions such as TFT[1] for multivariate forecasting should be considered in experiments. [1] Lim et.al., Temporal Fusion Transformers for interpretable multi-horizon time series forecasting, International Journal of Forecasting, 2021
>
> **Response to Weakness 3:** We thank the reviewer for the valuable suggestion. It should be noted that our work focuses on **multivariate state-aware forecasting**, where the model must jointly produce continuous forecasts and discrete hidden-state estimates. This task is fundamentally different from standard multivariate forecasting. Although TFT is a strong modern forecasting model, it produces only continuous predictions and lacks the ability to estimate hidden states, making it not directly comparable to DRL-STAF.
>
> In the revised manuscript, we have added two DL-HMM hybrids, namely NHMM [1] and NCTRL [2], as additional baselines. These models use deep networks for emission modeling and are able to produce discrete hidden-state estimates. We report representative results in Table 7, where DRL-STAF consistently achieves the best performance in both hidden-state inference and forecasting accuracy. More comprehensive experimental results are provided in the revised manuscript. The results can be seen in **Pages 9-10 and 18-20**.
>
> **Table 7: Comparison of DRL-STAF, NHMM, and NCTRL across three datasets (3-variable, 10-variable, and SMachine)**
> | Dataset      | Models       | Accuracy               | Precision              | Recall                 | F1                     | MAE                    | MSE                    |
> |--------------|--------------|-------------------------|-------------------------|-------------------------|-------------------------|-------------------------|-------------------------|
> | 3-variable   | NHMM         | 60.32% ± 0.02%         | 49.06% ± 13.33%        | 66.65% ± 0.03%         | 51.81% ± 0.05%         | 0.1277 ± 0.0029        | 0.0390 ± 0.0006        |
> |              | NCTRL        | 79.53% ± 3.88%         | 88.49% ± 3.90%         | 79.15% ± 3.76%         | 81.34% ± 2.93%         | 0.3936 ± 0.0182        | 0.3042 ± 0.0261        |
> |              | **DRL-STAF** | **98.17% ± 0.07%**     | **96.89% ± 0.19%**     | **99.62% ± 0.25%**     | **98.22% ± 0.07%**     | **0.0889 ± 0.0003**    | **0.0278 ± 0.0003**    |
> | 10-variable  | NHMM         | 75.55% ± 2.20%         | 51.33% ± 2.40%         | 62.00% ± 4.00%         | 55.11% ± 3.32%         | 0.2621 ± 0.0065        | 0.1600 ± 0.0046        |
> |              | NCTRL        | 90.31% ± 1.50%         | 88.92% ± 0.69%         | 86.71% ± 3.24%         | 86.65% ± 1.80%         | 0.3431 ± 0.0343        | 0.3935 ± 0.2922        |
> |              | **DRL-STAF** | **96.15% ± 0.15%**     | **95.44% ± 0.52%**     | **91.89% ± 0.60%**     | **93.26% ± 0.21%**     | **0.1090 ± 0.0005**    | **0.0395 ± 0.0009**    |
> | SMachine     | NHMM         | 70.77% ± 0.03%         | 0.00% ± 0.00%          | 0.00% ± 0.00%          | 0.00% ± 0.00%          | 0.0194 ± 0.0002        | 0.0010 ± 0.0000        |
> |              | NCTRL        | 72.54% ± 5.67%         | 54.04% ± 13.66%        | 41.66% ± 24.45%        | 42.33% ± 20.71%        | 0.0360 ± 0.0012        | 0.0023 ± 0.0001        |
> |              | **DRL-STAF** | **81.73% ± 0.10%**     | **100.00% ± 0.32%**    | **63.27% ± 0.18%**     | **77.50% ± 0.15%**     | **0.0189 ± 0.0000**    | **0.0010 ± 0.0000**    |
>
> [1] Tran, K. M., Bisk, Y., Vaswani, A., Marcu, D., & Knight, K. (2016). Unsupervised neural hidden Markov models. In Proceedings of the Workshop on Structured Prediction for NLP (pp. 63-71).
>
> [2] Song, X., Yao, W., Fan, Y., Dong, X., Chen, G., Niebles, J. C., ... & Zhang, K. (2023). Temporally disentangled representation learning under unknown nonstationarity. Advances in Neural Information Processing Systems, 36, 8092-8113.

---

> ### Author Response · Authors · 2025-11-29
>
> >**Question 1:** Can this framework used in multistep (long-horizon) forecasting?
>
> **Response to Question 1:** We thank the reviewer for the helpful question. The forecasting module of DRL-STAF is architecturally similar to standard deep forecasting models, and can be extended to H-step prediction by replacing the one-step emission head with an H-step head. Specifically, at each time $t$, the emission head outputs a vector of future observations $(\hat{x}\_{t+1},\cdots, \hat{x}\_{t+H})$ so the extension from one-step to multi-step forecasting does not require changing the overall prediction module's network structure.
>
> For the state-estimation module, there are two natural design choices for multi-step forecasting.
>
> (1) Fixed-regime over the horizon: one can assume that the hidden state remains unchanged over an H-step horizon. In this case, the policy chooses a discrete state at time $t$, and the environment is advanced by $H$ time steps using the corresponding emission head, with a reward defined from the aggregated H-step forecasting error.
>
> (2) Time-varying regimes with delayed rewards: alternatively, one can allow the hidden regime to change within the H-step horizon. The state estimator still outputs a regime at every step, but the reward signal is delayed until the true observations become available, at which point an H-step forecasting loss is computed and assigned back to the sequence of actions taken over the horizon. In this setting, the estimator typically needs richer historical information (longer observation windows and possibly summaries of past errors) to cope with the more challenging credit-assignment problem.
>
> In our preliminary experiments, we have verified that DRL-STAF can be adapted to long-horizon forecasting along the above lines. However, consistent with findings in HMM and switching state-space models, long-horizon forecasting is substantially more difficult because errors accumulate in both the continuous predictions and the inferred latent regimes. A principled extension to improve multi-step performance therefore remains an important direction for future work.

---

### Author Response · Authors · 2025-11-30
**General Response**

We sincerely appreciate the time, expertise, and thoughtful feedback that the reviewers **(huyU, xt2e, Qm9x, WBEf)** devoted to evaluating our work. We are encouraged by the recognition that approaching hidden-state inference through a reinforcement-learning formulation **provides a notably novel and impactful perspective on multivariate HMMs** **(huyU, Qm9x)**, and that the proposed **DRL-STAF** framework, which unifies forecasting and discrete hidden state estimation into a single coherent system, is effective and thoughtfully designed **(xt2e, WBEf)**.

We value the reviewers’ careful scrutiny, and have strengthened the paper with additional experiments, clarifications, and revisions.

### 1. Response to Common Concerns
**Clarifying Unified Objective & RL Training Strategy (xt2e, Qm9x, WBEf)** Reviewers noted that our reinforcement-learning objective, reward design, and two-stage training process required clearer explanation to fully convey how DRL-STAF performs joint forecasting and state estimation.
* We have added a clear comparison between DRL-STAF and likelihood-based DL-HMM hybrids to highlight the core distinction that DRL-STAF is distribution-free and directly optimizes forecasting risk.
* We have incorporated theoretical insights to clarify the rationale behind the reward design, provided a more explicit explanation of the function of each module in the framework, and added new ablation studies (**Appendix D.3**) to verify the contribution of each component.

**Concerns About the Infrequent Hidden-State Transition Assumption (huyU, Qm9x, WBEf)** Reviewers raised concerns that this assumption might be overly strong and limit applicability in fast-switching settings.
* We clarified that this assumption serves only as a conservative guidance mechanism under uncertainty. Its role is to encourage more stable early estimation, without enforcing state persistence or restricting the true transition patterns.
* We added fast-switching simulated datasets and ablation experiments (**Appendix D.1 and D.3**), which show that DRL-STAF remains effective under frequent transitions, whereas removing continuity-related components leads to instability and poorer state separation.

**Computational Cost (All Reviewers)** Reviewers requested a clearer characterization of the computational cost of DRL-STAF.
* We analyzed the parameter complexity of DRL-STAF and showed that its two-stage design avoids the exponential joint-state explosion, yielding essentially linear growth with the number of variables. (**Appendix E**)
* We evaluated computational cost and found that, although training is longer due to policy learning, inference remains efficient, and the advantages of our design increase with dimensionality. (**Appendix E**)

**Expanded Empirical Evaluation (huyU, xt2e, Qm9x, WBEf)** Reviewers emphasized the need for stronger empirical validation, broader baselines, and more transparent reporting of stability.
* We strengthened our evaluation by **adding likelihood-driven DL-HMM baselines (NHMM, NCTRL)**, constructing **two fast-switching simulated datasets** to test robustness under frequent regime changes, and incorporating an **additional real-world traffic network dataset (PeMS07)**.
* We added more **intuitive case-study visualizations** that jointly present forecasting outputs and estimated hidden states, providing clear evidence of DRL-STAF's ability to capture state dynamics and deliver stable, interpretable predictions in real-world scenarios.
* We also **expanded the ablation studies** to more clearly verify the contribution of each module within DRL-STAF.
* These updates provide a more comprehensive and transparent evaluation, and the results consistently support the effectiveness of our framework.(**Pages 9-10 and 18-20**)

### 2. Summary of Revisions
Prompted by these valuable reviews, we have significantly strengthened the manuscript:
* **Highlighted Contribution:** Revised the contribution to clarify the fundamental distinction between **DRL-STAF** and existing **likelihood-based DL-HMM hybrids**, emphasizing that **DRL-STAF** is, to our knowledge, **the first distribution-free framework for complex multivariate hidden Markov processes**, enabling flexible state dynamics while avoiding joint-state combinatorial explosion.
* **Expanded Evaluation:** Added likelihood-driven **DL-HMM baselines (NHMM, NCTRL)**, two **fast-switching simulated datasets**, a **real-world traffic network dataset (PeMS07)**, and more **extensive ablations** with mean ± std reporting.
* **Improved Clarity:** Refined the description of DRL-STAF by clarifying the rationale behind the reward design and providing clearer explanations of each module’s role within the framework.
* **Computational Analysis:** Provided parameter-complexity results showing linear scaling and added empirical measurements of training time and inference speed.

These revisions substantially improve the clarity, rigor, and empirical grounding of DRL-STAF.

---

### Meta-Review · Area_Chair_8WAD · 2026-01-06

**Summary:**

This paper proposes DRL-STAF framework which combines HMM-like state-space models with deep reinforcement learning for multivariate time series forecasting.

**Reviewer Concerns:**

Although the paper has some merits, such as a novel integration of RL for state transitions and clear algorithmic presentation, the issues raised by the reviews are critical. For instance, the possible overstatement of the contribution and lack of proper comparison to strong neural forecasting and state-space baselines (Reviewer WBEf), the absence of a clear, unified optimization objective and scalability analysis (Reviewer Qm9x), and the strong, untested assumption (Reviewer huyU). Although the authors address some issues in responses, the paper still needs a major revision before it can be accepted.

**Reviewer Scores:**

huyU and xt2e would remain the score given the borderline response.

Qm9x would remain the score as the concerns require a major rewrite.

WBEf would remain the score as the novelty and experimental rigor require a major revision.

---

### Decision · Program_Chairs · 2026-01-26

Reject